# Enhanced ocean-atmosphere carbon partitioning via the carbonate counter pump during the last deglacial

Stéphanie Duchamp-Alphonse[1], Giuseppe Siani[1], Elisabeth Michel[2], Luc Beaufort[3], Yves Gally[3] & Samuel L. Jaccard [4]

Several synergistic mechanisms were likely involved in the last deglacial atmospheric $pCO_2$ rise. Leading hypotheses invoke a release of deep-ocean carbon through enhanced convection in the Southern Ocean (SO) and concomitant decreased efficiency of the global soft-tissue pump (STP). However, the temporal evolution of both the STP and the carbonate counter pump (CCP) remains unclear, thus preventing the evaluation of their contributions to the $pCO_2$ rise. Here we present sedimentary coccolith records combined with export production reconstructions from the Subantarctic Pacific to document the leverage the SO biological carbon pump (BCP) has imposed on deglacial $pCO_2$. Our data suggest a weakening of BCP during the phases of carbon outgassing, due in part to an increased CCP along with higher surface ocean fertility and elevated $[CO_{2aq}]$. We propose that reduced BCP efficiency combined with enhanced SO ventilation played a major role in propelling the Earth out of the last ice age.

[1] GEOPS, Universities of Paris Sud and Paris-Saclay, CNRS, 91405 Orsay, France. [2] LSCE/IPSL Laboratoire des Sciences du Climat et de l'Environnement, CEA-CNRS-UVSQ, 91198 Gif-sur-Yvette, France. [3] Aix Marseille Univ, CNRS, IRD, INRA, Coll France, CEREGE, Aix-en-Provence, France. [4] Institute of Geological Sciences and Oeschger Center for Climate Change Research, University of Bern, 3012 Bern, Switzerland. Correspondence and requests for materials should be addressed to S.D.-A. (email: stephanie.duchamp@u-psud.fr)

The Southern Ocean (SO) is a key part of the global over-turning circulation as it witnesses the outcropping of carbon- and nutrient-rich Circumpolar Deep Water (CDW) in the Antarctic Zone (AZ), as a result of wind-driven (Ekman) upwelling[1]. A portion of these upwelled waters flow southwards to feed the abyssal circuit to form Antarctic Bottom Water (AABW), while the remainder flows to the North to feed Subantarctic Surface Waters (SSW) that mix with warm sub-tropical waters to form Antarctic Intermediate Waters (AAIW) and Subantarctic Mode Waters (SAMW), the mid-depth oceanic circuit that supplies nutrients to the low-latitude thermocline[1,2]. The STP, that is the net downward flux of carbon associated with organic matter export, counteracts carbon evasion to the atmosphere, as a fraction of the photosynthetic biomass that fixes dissolved inorganic carbon (DIC) in the sunlit ocean is exported and remineralized in the ocean interior. However, owing to iron (Fe) limitation on phytoplankton growth[3,4], the pre-industrial SO STP was unable to fully compensate the $CO_2$ outgassing and this area represented one of the main oceanic sources of natural $CO_2$ to the atmosphere[5].

During the last ice age, the deep circuit was probably more isolated from the atmosphere due to increased sea-ice coverage[6] and increased stratification[7,8]. Meanwhile the STP, fueled by enhanced deposition of Fe-bearing dust that favored a more complete macronutrient uptake by phytoplankton, might have been more efficient[3,9,10]. Therefore, reduced rates of vertical exchange combined with a more efficient STP promoted the storage of $CO_2$ in the ocean abyss, thereby contributing to lower atmospheric $CO_2$[11,12].

The collapse of vertical $\Delta^{14}C$ and $\delta^{13}C$ gradients in the SO suggests that more vigorous deep and mid-depth circulations would have reconnected the deep carbon reservoir to the surface during Heinrich Stadial 1 (HS1, 17.5–14.7 kyr BP) and the Younger Dryas (YD, 12.8–11.5 kyr BP), thus promoting the transfer of respired carbon to the surface ocean and the atmosphere[13,14]. These observations have been corroborated by sedimentary geochemical data suggesting that surface waters of the Subantarctic Atlantic and the Eastern Equatorial Pacific (EEP), which derive from water upwelled in the SO, became a substantial source of $CO_2$ during the last deglaciation[15].

Surprisingly, little is known about deglacial export production and STP patterns in the Subantarctic Zone (SAZ) outside the Atlantic sector. The weakening of the global STP efficiency has been documented during the early deglaciation (~17.5– ~14 kyr BP) using a global compilation of $\delta^{15}N$ measurements and oxygenation proxies[16]. Millennial-scale export production reconstructions based on organic (TOC, biomarkers) and inorganic ($SiO_2$, $CaCO_3$, bioBa) proxies[9,10,17–21] provide insights into oceanic nutrient dynamics, ventilation changes, and export of particulate carbon across the deglaciation. However, these records remain sparse and mainly come from South Atlantic cores located downwind of Patagonia, the most prominent dust source region to the SO[22]. While they provide valuable case studies for testing the "Fe-hypothesis"[3], these records are not necessarily representative of the entire SO. Particularly, the comparison with South Pacific and Indian Oceans is not straightforward because hydrothermal and sedimentary Fe sources may play an important role in modulating productivity in these sectors[23].

Besides, these reconstructions neither document the contribution of specific phytoplankton groups, nor their respective leverage on the BCP strength. The sedimentary burial of biogenic opal and carbonate, often thought to reflect diatom and coccolithophore export production respectively, could be affected by changes in zooplankton abundance, grazing pressure, and/or changes in the degree of remineralization processes. More importantly, these reconstructions largely ignore the relative

contribution of the CCP, despite its fundamental role in the marine carbon cycle[24]. The production of particulate inorganic carbon (PIC) by calcifying plankton in the sunlit ocean and its eventual dissolution in the subsurface engenders a surface-to-depth alkalinity gradient, causing $CO_2$ to be released back to the atmosphere[24]. As such, the CCP acts to partially offset the air-sea partitioning of carbon associated with the STP.

Therefore, it is crucial to overcome these important short-comings by focusing on the production pattern of specific phytoplankton groups from a broad range of locations within the SAZ and evaluating their impact on the carbon cycle. Coccolithophores are relevant for addressing this outstanding issue. This single-celled phytoplankton group has unique effects on the oceanic carbon cycle in that it uses DIC for both photosynthesis and calcification and accounts for a significant proportion of the global marine export production[25]. Hence, coccolithophores contribute to both the STP and CCP and impact the strength of the BCP since modifications in calcification patterns related to changes in surface water chemistry modulate the POC:PIC ratio of sinking biogenic material with consequences for the air-sea partitioning of carbon[26–28].

Here, we explore the deglacial calcification pattern of cocco-lithophores, particularly the Noëlaerhabdaceae family, in a well-dated sediment core[14], retrieved from the Chilean margin (MD07-3088; 46.1°S, 75.7°W, 1536 m water depth), at the transition of AAIW and the Pacific deep water (PDW)[2] (Fig. 1 and Methods). This site is located within the Antarctic Circumpolar Current (ACC) under the direct influence of the northward transport of nutrient rich-SSW[2] and relatively far from the main dust sources today, but likely also in the past[22]. As such, this core represents a suitable archive to document the leverage SO upwelling and associated changes in surface water chemistry might have exerted on SAZ coccolithophore productivity. We complement these observations with planktonic foraminifera records to provide a complete representation related to the integrated PIC accumulation and compare these data with reconstructed past changes in the buried POC:PIC ratio, suggested to reflect the C-rain ratio. As such, we qualitatively document the relative contribution of STP and CCP to the deglacial rise in atmospheric $CO_2$ at a decadal timescale. Our study highlights that changes in biological export production in high southern latitudes operated synergistically with physical mechanisms thereby enhancing the transfer of carbon from the ocean to the atmosphere during the last deglacial. The reinvigoration of the SO vertical mixing contributed to the release of respired carbon and regenerated nutrients to the SAZ that promoted planktonic calcification, thereby increasing the CCP, and concomitantly weakening the BCP.

## Results and discussion

**Increase in SSW fertility and $[CO_{2aq}]$ during SO upwelling.** In Fig. 2, we present coccolith abundance and mass (Fig. 2a, c, d, i) from sediment core MD07-3088 (Methods) that are compared to $\Delta\Delta^{14}C$ and $\Delta\delta^{13}C$ reconstructions from the same core[14] (Fig. 2f, g), coccolith abundances from the SE Pacific[29] (ODP 1233, Fig. 2b; Methods and Supplementary Fig. 1), biogenic opal flux from the South Atlantic[30] (TN057-13-4PC, Fig. 2h), $\delta^{11}B$-based $\Delta p_{CO_2}$ from the EEP[15] (ODP 1238, Fig. 2j; Methods; Supplementary Fig. 1), and local summer SST estimates[14,31] (Fig. 2e), to better understand their relationships to productivity and ocean circulation patterns on a regional scale (Figs. 1 and 2). Coccolithophore calcification patterns are not affected by diagenetic alteration, with no evidence of coccolith dissolution (Methods and Supplementary Fig. 2). Noëlaerhabdaceae coccolith abundance features three distinct deglacial peaks (18.6–18.2, 17.7–16.2,

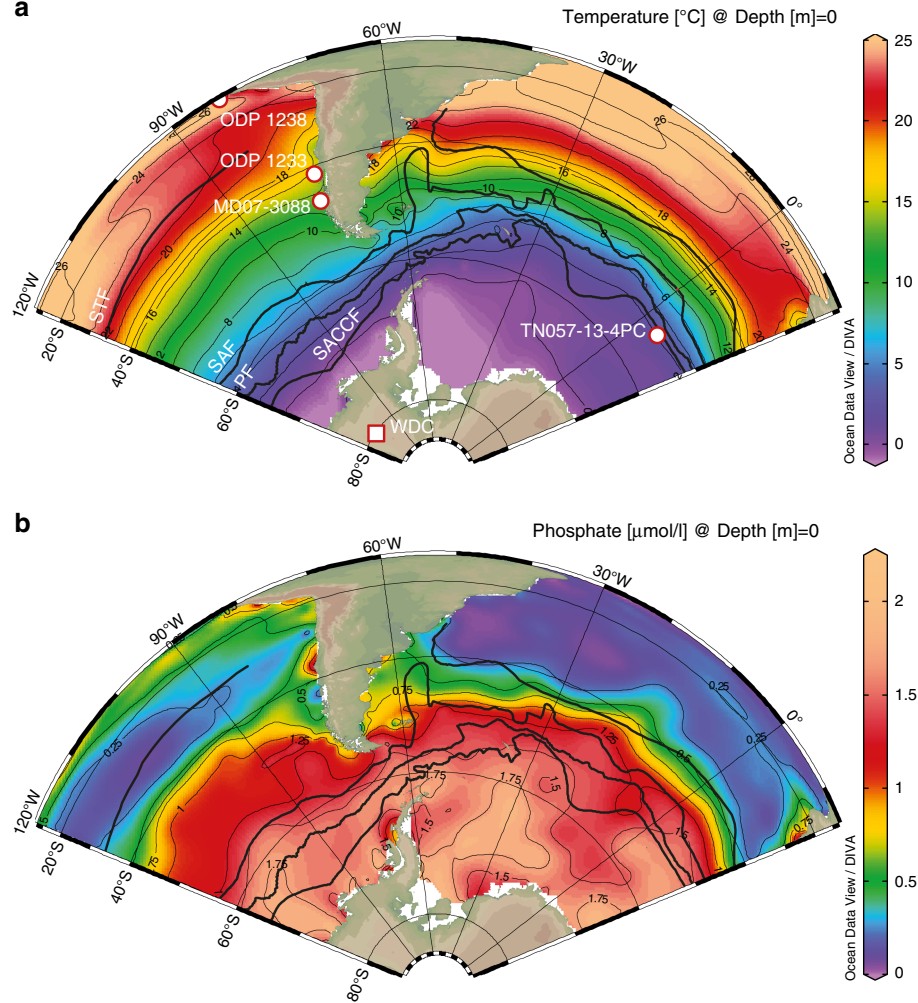

**Fig. 1** Location of sites and modern ocean surface temperature and phosphate concentrations. Temperature (**a**) and phosphate (**b**) concentration fields are plotted with the Ocean Data View (ODV) software[68] with WOA09[69]. Solid black lines represent the: Subtropical (STF), Subantarctic (SAF), Polar (PF) and Subantarctic Circumpolar Current (SACCF) Fronts[70]. The Polar Frontal and Subantarctic Zones are the regions between the PF and SAF, and between the SAF and the STF respectively. **a** Solid white circles symbolize the geographic location of sediment cores MD07-3088 (46.1°S, 75.7°W, 1536 m), TN057-13-4PC[30] (53.2°S, 5.1°E, 2850 m), and sites ODP 1233[29] (41.0°S, 74.4°W, 838 m) and 1238[15] (1.5°S, 82.5°W, 2203 m). The solid white square highlights the West Antarctic Ice Sheet Divide ice core[56] (WDC, 79.5°S, 112.1°W, 1766 m above sea level)

and 12.8–11.1 ka BP), with highest abundances reported for HS1 and the YD (Fig. 2a). Within Subantarctic ecosystems located away from the main dust sources, phytoplankton growth is typically modulated by the supply of dissolved phosphate ($PO_4^{3-}$) and nitrate ($NO_3^-$) via SO upwelling today[32] but also in the past[33,34]. This is particularly the case for the Chilean margin, where river runoff[35] and glacier erosion[36] provide additional sources of micronutrients alleviating the limitation Fe is imposing on phytoplankton growth in the open SO[34]. Besides, macronutrient concentrations within the SSW were not perennially high[37]. Therefore, the most reasonable explanation is that coccolith abundance primarily reflects the phytoplankton response to macronutrient supply. This assumption is corroborated by concomitant higher abundances of *H. carteri* and to a lesser degree *C. leptoporus* that have an affinity for meso- to eutrophic conditions in the SE Pacific[38] (Fig. 2c, d). Besides, the structure of this record closely resembles those of $\Delta\Delta^{14}C$ and $\delta^{13}C$ reconstructions[14] that reflect increased rates of vertical mixing, thus supporting the notion that upper ocean productivity at site MD07-3088 was directly modulated by regional ocean circulation changes. Furthermore, the downcore coccolithophore export productivity record mimics coccolith abundances in the northern part of the

SAZ[29] as well as opal fluxes in the AZ at sites influenced by similar processes[30]. This suggests that the inferred changes in productivity were neither limited to a specific phytoplankton group nor a specific area, but rather highlight a regional sensitivity of phytoplankton growth to ocean circulation and nutrient supply from below.

The second outstanding feature of the micropalaeontological records relates to the ~50% distinct increases in the mean Noëlaerhabdaceae coccolith mass (Fig. 2i), coincident with decreasing local $\Delta\Delta^{14}C$ and $\Delta\delta^{13}C$ values (Fig. 2f, g)[14], as well as with increasing surface water $\delta^{11}B$-based $\Delta p_{CO_2}$ values in the EEP[15] (Fig. 2j; Methods; Supplementary Fig. 1). Numerous studies related coccolith mass to the degree of coccolith calcification in Pleistocene and recent sediments[39–41]. Indeed, variations in coccolith mass can, under some circumstances, reflect variations in the thickness of an individual coccolith, and therefore relates to the calcite quota of a given cell[27]. However, changes in coccolith mass may additionally be driven by changes in coccolith area associated to changes in cell sizes[27,28,42] and as such coccolith mass must be size-normalized to represent the degree of coccolith calcification[27,28]. At site MD07-3088, we focus on changes that affected all narrowly restricted size classes within

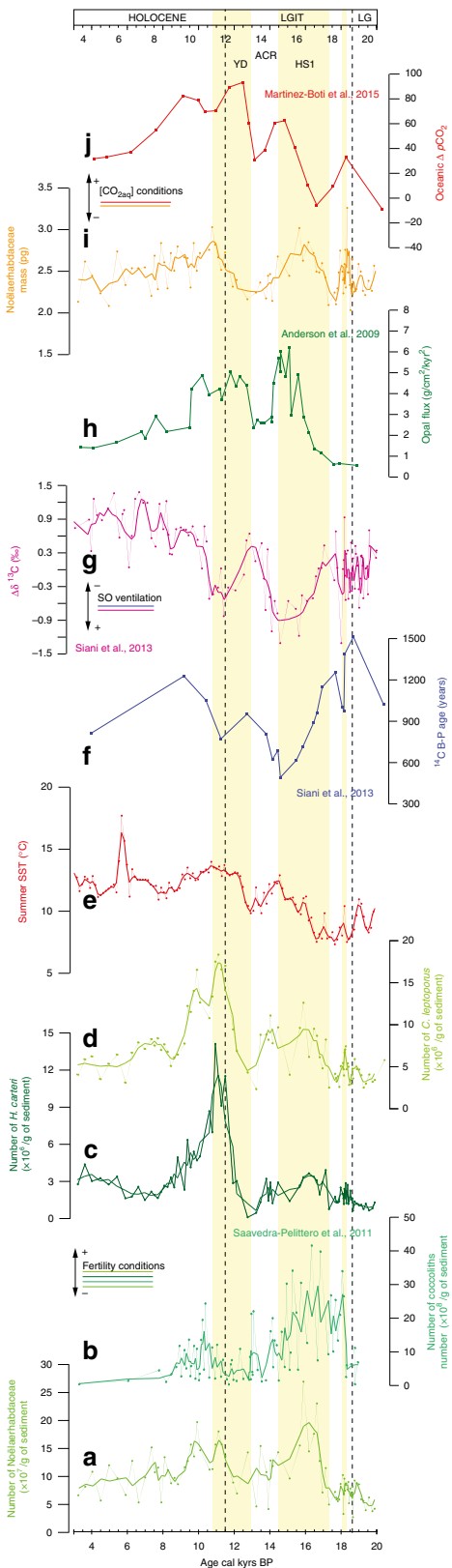

**Fig. 2** Southern Ocean productivity and circulation patterns during the last deglaciation. **a**, **c**, **d** Number of Noëlaerhabdaceae, *H. carteri* and *C. leptoporus* at site MD07-3088 (/g of sediment), with an error bar of ±1%. **b** Number of coccoliths at site ODP 1233 (/g of sediment)[29]. **e** Summer sea surface temperatures (SST, °C) using the Modern Analogue Technique[14, 31]. **f** [14]C age difference between paired benthic and planktonic foraminifera ([14]C B-P)[14]. **g** $\Delta \delta^{13}C = \delta^{13}C_{G.\ bulloides} - \delta^{13}C_{C.\ wuellerstorfi}$[14]. **h** Opal fluxes at TN057-13-4PC[30]. **i** Noëlaerhabdaceae mass (pg) at site MD07-3088 with an error bar of ±3%. **j** Surface ocean $\Delta p_{CO_2}$ reconstruction for ODP 1238[15]. LG and LGIT are for Late Glacial and Last Glacial-Interglacial Transition, respectively. Smoothed curves (thick lines of **a–e**, **g**, and **i**) use a three-point moving average. Yellow shading marks periods of enhanced deep-water ventilation and resumption of SO upwelling during the last deglaciation, in conjunction with higher ocean surface fertility, [$CO_{2aq}$], and sometimes increased SST conditions

between coccolith mass and the size normalized thickness index "SN"[27,44] (Methods) ($r^2 = 0.73$) as well as the coccolith aspect ratio "$AR_L$"[28] (Methods) ($r^2 = 0.63$) (Fig. 3), which both document the degree of Noëlaerhabdaceae coccolith calcification.

Batch culture experiments provided conflicting responses related to coccolithophore calcification rates with studies reporting both depressed[45,46] or elevated[47] calcification under high [$CO_{2aq}$]. However, in the geological record—when general selection for growth strategies[28] and phenotypic plasticity naturally occurred and regulated the carbon acquisition within the cell[48]—more heavily calcified coccoliths were systematically associated with increased atmospheric $p$CO$_2$[27,28,49,50]. Indeed, high $p$CO$_2$ favors intracellular competitive reallocation of dissolved bicarbonate (HCO$_3^-$) from the site of photosynthesis (chloroplast) to the site of calcification (coccolith vesicle)[48]. Such processes may be relevant in coastal ecosystems such as the Chilean margin, where highly calcified *E. huxleyi* morphotypes thrive under low-pH sea-surface conditions[39]. Therefore, it is most likely that more heavily calcified coccoliths reported from site MD07-3088 at times when the upwelling of CO$_2$-rich deep waters increased, reflect increasing surface [$CO_{2aq}$] and highlight the equatorward advection of SSW together with AAIW/SAMW, thus supporting the mechanisms behind upper-ocean acidification previously documented in the SAZ and the EEP during these time intervals[15]. The 50% increase in Noëlaerhabdaceae mass observed both during HS1 and YD occurred at times of ~2–3 μmol/L rises in SSW [$CO_{2aq}$] in the SAZ[15], which is in the exact same order of magnitude than the coccolith mass and [$CO_{2aq}$] increases (50%, ~3 μmol/L respectively) previously documented for the penultimate deglaciation within the southernmost Pacific[28]. We cannot exclude that increased temperatures during major upwelling phases (~+4 °C during HS1, and ~+3 °C during YD[14,31]), may have partially contributed in promoting coccolith production and calcification[28,51], but warming was not always in phase with the coccolith patterns, and may thus be of secondary importance (Fig. 2).

**Reduced BCP due to increased CCP during SO upwelling.** Sedimentary bromine (Br) and calcium (Ca) have been shown to be associated with biogenic organic carbon and carbonate in marine sediment records, respectively. Br is primarily associated with marine organic matter[52] as it is directly involved in the marine biological cycle and in non-biological reactions in the water-column that implicate marine organic matter[53]. Calcium may be of detrital, biogenic or diagenetic origin. At site MD07-3088, the organic origin of Br is clear as testified by the overall excellent linear correlation with discrete TOC measurements ($r^2 = 0.87$; Supplementary Fig. 3) as well as with the $\delta^{13}$C and C/

the Noëlaerhabdaceae family[43] (Supplementary Fig. 2). We find that changes in coccolith mass typically reflect changes in coccolith thickness (Fig. 3), suggesting that they document changes in the coccosphere calcite quota. Our assumptions are further corroborated by the clear positive relationship that exists

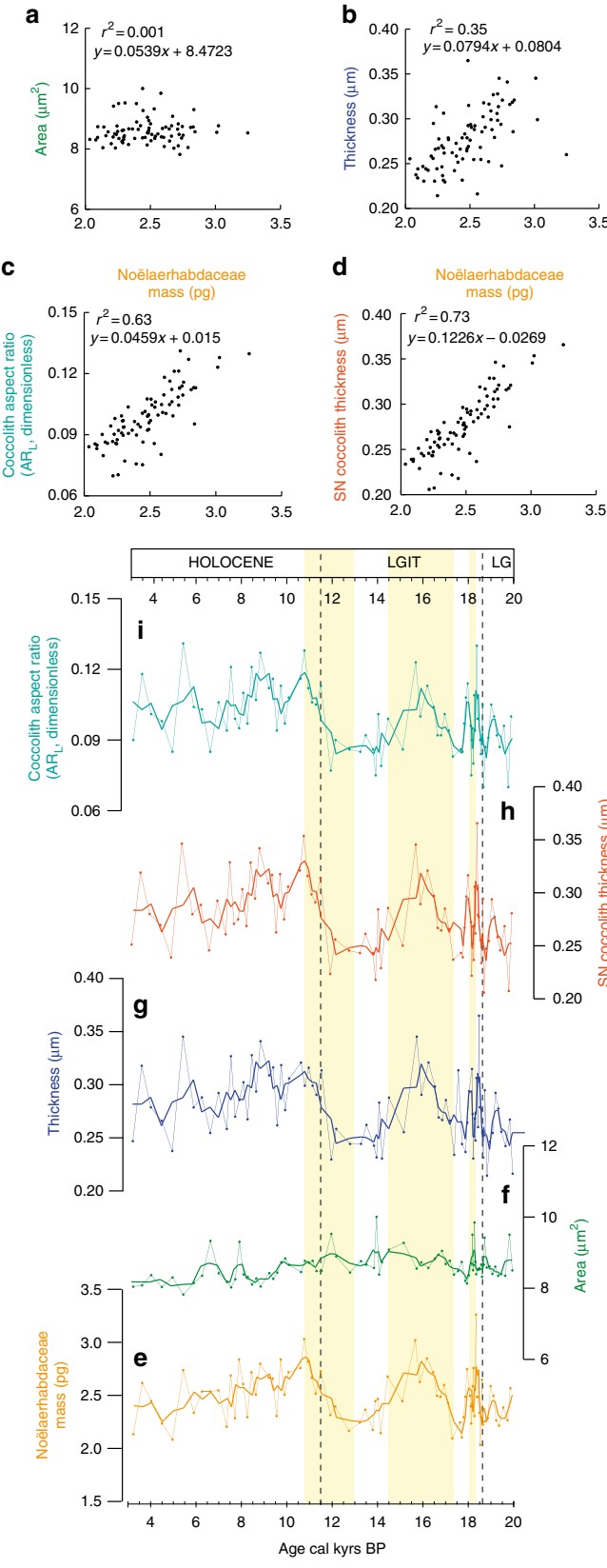

**Fig. 3** Noëlaerhabdaceae coccolith morphometrics, and their response to excepted [$CO_{2aq}$] at site MD07-3088 over the last deglaciation. **a–d** Relationships between coccolith mass (pg) and coccolith area (**a**), thickness (**b**), as well as coccolith aspect ratio ($AR_L$)[28] (**c**) and SN thickness[27] (**d**) (Methods section). **e–i** Coccolith morphometric changes during the deglaciation: **e** mass (pg), **f** area ($\mu m^2$), **g** thickness ($\mu m$), **h–i** SN[27] and $AR_L$[28] indices (Methods section). Coccolith mass show no relationship with coccolith area, but clear positive relationships with size normalized coccolith mass indices $AR_L$ and SN which indicates that the mass of coccoliths typically increases in proportion with their thickness. Therefore, changes in coccolith mass document changes in the degree of coccolith calcification (i.e. PIC/POC ratio). Obviously, the Noëlaerhabdaceae coccolith mass increases observed during enhanced SO upwelling associated with higher sea surface fertility conditions (yellow shading), document increased coccosphere calcite quota in response to SSW [$CO_{2aq}$] rises during HS1 and YD

free terrigenous material sourced from metamorphic and plutonic rocks of the Andes[35] that contain negligible amounts of calcium-bearing minerals (plagioclases), while Ca and Ca/Ti report excellent linear correlations with $CaCO_3$ ($r^2 = 0.75$ and $0.73$, respectively; Supplementary Fig. 3) thus excluding any Ca of terrigenous origin. As for other sites from the Chilean margin located well above the lysocline[55] (Methods), it is thus reasonable to consider that Ca is primarily associated with biogenic carbonates, and more particularly planktonic calcifiers. Indeed, bulk-sediment carbonate content fluctuations mimic those observed for the $CaCO_3$ produced by coccolithophores and planktonic foraminifera (Fig. 4, Supplementary Fig. 3; Methods). We suggest that, in our core, POC:PIC ratio changes in the sediments likely reflect changes in the C rain ratio (POC:PIC) (Methods). Therefore, the downcore Br/Ca ratio is used to provide an estimate of the strength of the STP relative to the CCP, which serves as a robust tool to reconstruct decadal changes in the BCP efficiency (Fig. 4d). This record depicts highest values during the Late Glacial (19.6–17.7 ka), the Antarctic Cold Reversal (ACR; 14.1–12.0 ka) and the Mid Holocene (8.6–3.0 ka). On the other hand, significant reductions in Br/Ca are observed at times intense SO upwelling prevailed, particularly during HS1 and the YD. These transient declines are mainly driven by changes in $CaCO_3$ export during HS1 since the TOC record is characterized by increasing values during this specific time interval (Supplementary Fig. 3), while both increasing $CaCO_3$ and relatively high yet decreasing TOC values might be associated during the YD. Furthermore, they match the pronounced peaks in surface ocean fertility (Fig. 2a–d) as well as coccolith and planktonic foraminifera abundances and masses that are well expressed by 3-fold to 10-fold and up to 20-fold increases in the overall amount of burial $CaCO_3$ produced by coccolithophores and planktonic foraminifera respectively, and coincide with prominent rises in atmospheric $pCO_2$[56] (Fig. 4). This increase in the CCP, associated with rising macronutrient availability and thus, enhanced fertility in the SAZ, is comparable to the 6-10 folds increase in deep-ocean PIC fluxes previously documented for naturally iron-fertilized sites from the Polar Frontal Zone compared to non-fertilized ones[24]. In both cases, increased fertility is linked to a rise of about one order of magnitude of the PIC flux, thus confirming the important role of CCP in mediating the reduction of deep-ocean $CO_2$ storage[24]. The impact of changing POC:PIC ratio ($1/\rho$) on $pCO_2$ is shown in Fig. 5 for HS1 and the ACR, in cases for which 10 to 50% of the exported POC is preserved in the sediments (Methods). In all cases, primary production decreased sea surface [$CO_{2aq}$], as $1/\rho$ is higher than the critical value of 0.54 for which the CCP would completely counteract the STP. Nevertheless,

N values of bulk organic matter (ranging from −24.9 to 20.05% and from 7.6 to 11.5, respectively) that cluster well within the typical ranges for well-preserved marine organic components[54] (Supplementary Fig. 4). The sediment mostly receives carbonate-

during HS1, the efficiency of the BCP was reduced compared to the ACR owing to lower POC:PIC ratios. The SAZ thus became a net source of $CO_2$ during HS1 and the YD[15], due to enhanced SO upwelling of aged, $CO_2$-enriched deep waters[14] and a concomitant weakening of the BCP. Indeed, the increase in calcite production by coccolithophores and planktonic foraminifera caused a decrease in surface-ocean alkalinity (ALK) (or in other

words, an increase in $pCO_{2aq}$), thus promoting the net outgassing of carbon from the ocean interior to the atmosphere (Fig. 5). In such a scenario, the CCP would have contributed to weaken the marine BCP, with significant impact on atmospheric $pCO_2$ since the very beginning of the last deglaciation. The STP was not efficient enough to offset the carbon release during the last deglaciation, in part due to the contribution of the CCP. Increased planktonic calcification in the SAZ has the potential to effectively amount to a reduction of the overall POC:PIC rain ratio in the SO (Fig. 5), with significant impacts on the net flux of $CO_2$ from the ocean to the atmosphere[24].

Our study reveals the intrinsic link between BCP strength and changes in Southern Ocean circulation in coordinating the

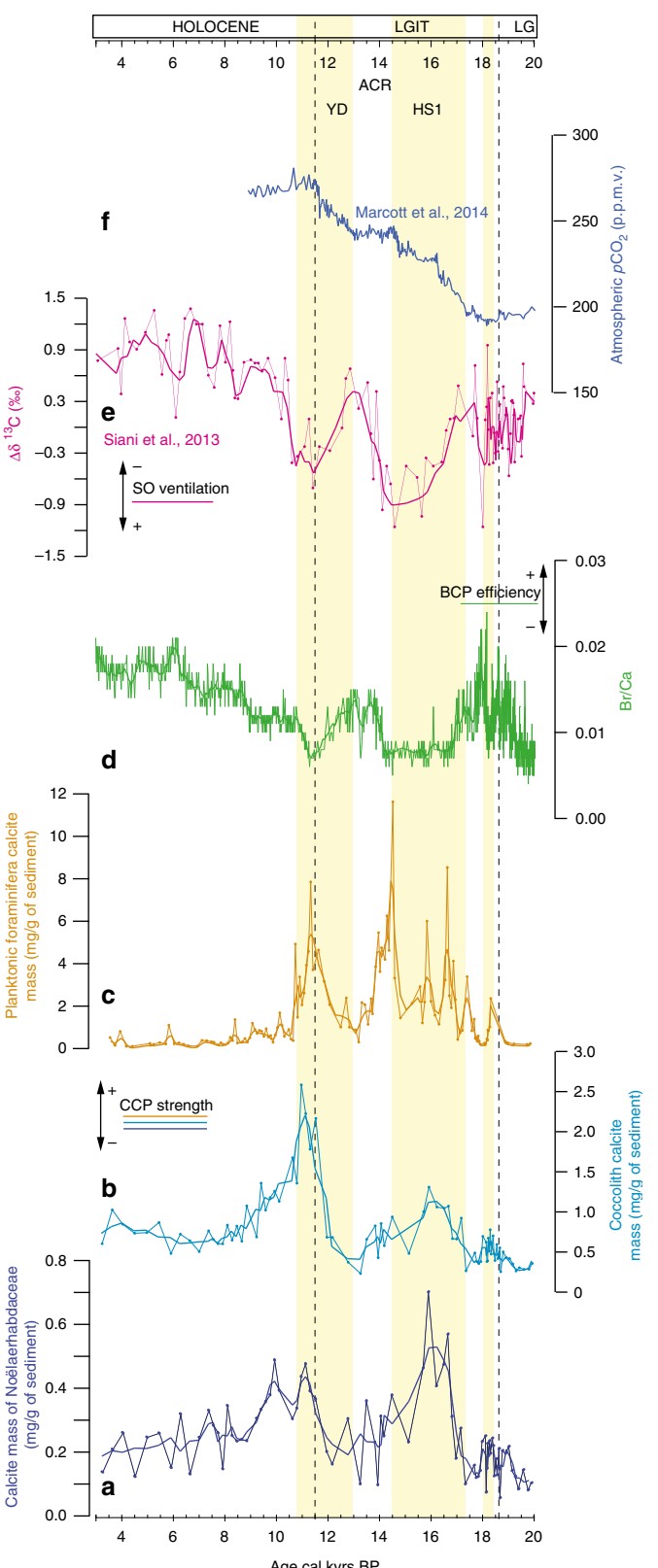

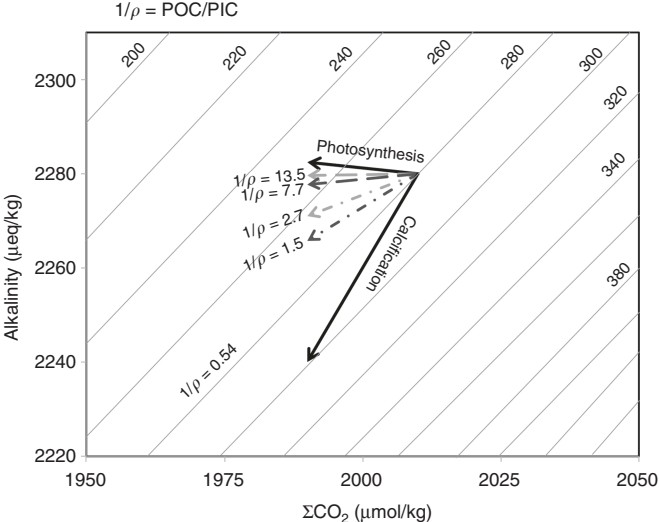

**Fig. 5** Influence of soft tissue and carbonate counter pumps on the $CO_2$ partial pressure of surface waters, as a function of the POC/PIC rain ratio ($1/\rho$). Solid black line isocontours represent $pCO_2$ (µatm) for constant salinity (34‰), temperature (14 °C), and depleted phosphate (0.5 µmol/kg) and silicate (3 µmol/kg) contents. The solid black arrows represent the effect of biogenic export production in the case of photosynthesis and calcification only. The dashed (or dot-dashed) black and grey arrows illustrate the influence of the biological pump (and particularly the CCP) during the HS1 relative to the ACR, i.e., when $1/\rho$ decreases by a factor of 1.8, assuming that 10 (or 50%) of the exported POC has been preserved within the sediment, i.e., under two probable export production conditions at site MD07-3088

**Fig. 4** Subantarctic carbonate counter pump strength and biological pump efficiency coupled with circulation pattern and atmospheric $pCO_2$ during the last deglaciation. **a–c** Noëlaerhabdacea, coccolith and planktonic foraminifera calcite masses (mg/g of sediment) at site MD07-3088, with errors bars of ±3% and ±20%, respectively. **d** Br/Ca ratio as an indicator of POC/PIC rain ratio and thus biological carbon pump efficiency (see Supplementary Fig. 3). **e** $\Delta \delta^{13}C = \delta^{13}C_{G.\ bulloides} - \delta^{13}C_{C.\ wuellerstorfi}$[14]. **f** Atmospheric $pCO_2$ from WDC[56]. Smoothed curves (thick lines of **a–c**, **e** and **d**) use three and eleven-point moving averages respectively. LG and LGIT are for Late Glacial and Last Glacial-Interglacial Transition, respectively. Yellow shading marks periods of reinvigorated SO upwelling (associated with enhanced sea surface fertility conditions and higher $[CO_{2aq}]$) during the last deglaciation, in conjunction to higher CCP strength and subdued biological pump efficiency, at times of increased atmospheric $pCO_2$

partitioning of carbon between the ocean interior and the atmosphere during the last glacial termination. As such, much more attention should be brought to the response of calcifying plankton at other sites within the Southern Ocean but also in low latitudes, to better quantify their relative contribution in the past global $p\mathrm{CO}_2$ budget.

## Methods

**Material and site description.** The CALYPSO core MD07-3088 was retrieved during the IMAGES PACHIDERME (MD 159) expedition by the French R/V *Marion Dufresne* off Southern Chile (46°04 S; 075°41 W), at a water depth of 1536 m, i.e., well above the modern lysocline (around 3700 m)[55] (Fig. 1). The site is bathed by the upper layer of southward flowing Pacific Deep Water (PDW), at the boundary with northward flowing AAIW[2]. The surface waters above the site are on the direct northward path of the SSW that is fed by the DIC- and nutrient-rich surface waters of the Antarctic Zone. These Antarctic and Subantarctic surface waters represent the source for AAIW/SAMW (i.e., intermediate depth waters), and have been hypothesized to be a major conduit through which high-latitude ocean changes are transmitted to the lower latitudes[2]. Core MD07-3088 has the advantage of being located within the Subantarctic Zone, well to the south and well to the north of the Tropical and Polar Frontal Zones, respectively, and has probably not been affected by potential shift of the Subtropical Front (STF) or the Subantarctic Front (SAF) in the past. The southern tip of Chile is the only continental mass intercepting the westerly winds within this latitude range, generating a zone of high precipitations that result in high fluvial sediment supplies to the South Pacific Ocean. Since any significant sediment reworking is precluded at site MD07-3088[14], the extremely high sedimentation rates recorded at site MD07-3088 (~300 cm/kyr during the Last Glacial and ~60 cm/kyr during the deglaciation and the Holocene[14]) provide a rare opportunity to study productivity patterns of the Subantarctic Zone with decennial to centennial resolution during the last glacial termination.

**Age models of core MD07-3088 and ODP cores 1238 and 1233.** The MD07-3088 age model has been determined using SH13[14] as the $^{14}$C Southern Hemisphere calibration curve[57]. In order to compare our micropaleontological and geochemical records with ODP sites 1233 and 1238 located in the South-Eastern Pacific (SEP, 41.0°S, 74.4°W) and the Eastern Equatorial Pacific (EEP, 1.5° S, 82.5°W) respectively, we established a common age model for these cores to test temporal phasing since the late glacial period. For ODP site 1233, we use the recently updated age model of ref. [58] and based on the reservoir $^{14}$C age estimates by ref. [14]. The age model of ODP site 1238 is based on 10 AMS $^{14}$C dates obtained on planktonic foraminifera *Neoglobboquadrina dutertrei* using a constant local sea-surface reservoir $^{14}$C age ($R_S$) correction ($\Delta R = 72 \pm 35$ yr) based on previous regional estimates[15]. However, this approach did not consider the possible advection of old subsurface waters in particular during the deglaciation as suggested by previous studies at local and regional scales[59,60]. Since no independent $R_S$ estimates are available in literature for the EEP, other methods must be considered in order to obtain a robust common stratigraphic framework. Hence, we first compared the planktonic foraminifera $\delta^{13}$C records of the two cores versus conventional $^{14}$C age[14,15], the $\delta^{13}$C record for ODP 1238 representing sub-surface record as it is measured on *N. dutertrei* (Supplementary Fig. 1). In general, the first order (and most of the second order) changes in planktonic $\delta^{13}$C are recorded in both cores. These results also match co-existing benthic–planktonic foraminifera (B-P) $^{14}$C and $\delta^{13}$C differences in core MD07-3088, indicating variations in oceanic ventilation (see ref. [14] for extensive discussion) versus conventional $^{14}$C age. Similarly, the ventilation changes expressed in term of upwelling increases observed in core MD07-3088 were coeval with changes in surface ocean carbon content in the EEP[15] (Supplementary Fig. 1). Through these comparisons, it is clear that enhanced mixing (between ~15 and ~13.1 $^{14}$C ka, and between ~12 and ~10.5 $^{14}$C ka) was characterized by a lower difference between planktonic and benthic carbon isotope signatures, and are globally synchronous with oceanic $p\mathrm{CO}_2$ changes. This finding supports the hypothesis that the planktonic foraminiferal records correspond to the water masses with the same history (SAW and SAMW) presenting similar radiocarbon contents. This allows us to deduce that EEP and SEP were characterized by similar reservoir $^{14}$C age changes at least since the last deglaciation.

**Coccolith slides and morphometric measurements (SYRACO).** Slides of 80 samples were prepared at GEOPS laboratory. Briefly, ~0.03 g of sediment was diluted in 28 mL Luchon water (pH = 8, bicarbonate = 78.1 mg per liter, total dissolved solid = 83 g per liter) within a flat beaker, and settled on a 12 × 12 mm coverslip for 4 h 30 min. The coverslip was then oven-dried at 70 °C, and mounted on slides with NOA74. This technique ensures a homogenous distribution of coccoliths and allows quantifying the amount of material per gram of sediment[61] as follow:

$$A = (\mathrm{Nc} \times \mathrm{Sf})/(\mathrm{No} \times \mathrm{So} \times \mathrm{Ws}) \qquad (1)$$

where $A$ is the number of coccoliths per gram of sediment; Nc is the number of counted coccoliths (between 505 and 3900); Sf is the surface of the flat beaker (3117 mm$^2$) in which suspended sediments (and coccoliths) settle; No is the number of view fields (165); So is the surface of a view fields (0.01 mm$^2$) and Ws is the weight of sediment that settled in the flat beaker (between 0.018 and 0.043 mg).

For each sample, abundance and morphometric analyses (length, width, area, mass) of individual coccoliths were automatically obtained with an average of 1591 coccoliths per sample, by the SYRACO software using automated microscope (Leica DM6000B). SYRACO performs pattern recognition under cross-polarized light using artificial neural networks[61]. It detects and classifies most of the coccoliths present in the samples throughout the time series (mainly represented by *Emiliania huxleyi*, *Gephyrocapsa muellerae*, *Gephyrocapsa oceanica*, *Calcidiscus leptoporus*, and *Helicosphaera carteri*). Coccolith mass were directly deduced based on a quasi-linear relationship that exists between their brightness (birefringence in grey scale colors) and their thickness under cross-polarized light. Because this method applies on coccoliths thinner than 1.55 µm that exhibit grey scale colors[61], we interpret only thickness and mass measurements for Noëlaerhabdaceae coccoliths. Indeed, their abundance and morphometric parameters show standard error of ±1% and ±3% in each sample respectively. Morphometric analyses for *C. leptoporus* and *H. carteri* that display third-order interference colors (and thus increasing standard errors), are only presented within the Supplementary Information (Supplementary Fig. 2).

Since coccolith mass are not independent of coccolith size, we calculated size-normalized thickness indices for all the Noëlaerhabdaceae coccoliths within each sample to verify that changes in coccolith mass represent changes in calcification, according to the two equations that exist so far[27,28,44] (Fig. 3). We obtained the Size Normalized Thickness index SN[27,44] that considers coccolith thickness related to cell surface area as follow:

$$\mathrm{SN}_{\mathrm{thickness}}\,(\mu m) = [(\mathrm{ML} - \mathrm{CL}) \times S] + \mathrm{CT} \qquad (2)$$

Where ML is the mean coccolith length over the whole time serie, CL is the length of coccolith X in Sample A, $S$ is the slope of the linear regression between coccolith length and coccolith thickness for all coccolith in Sample A, and CT is the original thickness of coccolith X in Sample A (i.e., coccolith mass/coccolith area ratio).

We calculated the lateral cross-sectional aspect ratio $\mathrm{AR}_L$[28] that considers coccolith thickness related to cell volume as follow:

$$\mathrm{AR}_L\,(\text{dimensionless}) = T_L\big/\sqrt{A_L} \qquad (3)$$

where $T_L$ and $A_L$ are the thickness and the area of coccolith X in Sample A respectively. $T_L = M_L/A_L$, i.e. coccolith mass ($M_L$)/coccolith area ($A_L$).

Coccolith area and mass values as well as $\mathrm{SN}_{\mathrm{thickness}}$ and $\mathrm{AR}_L$ values obtained herein are in the same order of magnitude than published data using similar birefringence-based methods[27,28,39].

**Coccolith taxonomy and preservation.** More than 96% of the assemblages were composed of five species: *Emiliania huxleyi*, *Gephyrocapsa muellerae*, *Gephyrocapsa oceanica*, *Calcidiscus leptoporus*, and *Helicosphaera carteri*. As for modern settings, Emiliania and Gephyrocapsa, that constitute the Noëlaerhabdaceae family, represent the most prominent genera (from 81 to 97% of the assemblages) and reflect the main patterns of the total coccoliths. For that reason, but also because smallest Emiliania and Gephyrocapsa from the SE Pacific present a wide range of morphotypes[62] that are not easily classified under light microscope, we mainly considered the Noëlaerhabdaceae family instead of Emiliania and Gephyrocapsa species. Besides, species assignations within the Noëlaerhabdaceae family are primarily based on size[43], and all narrowly restricted size classes of Noëlaerhabdaceae present the same main patterns (Supplementary Fig. 2). Indeed, generally, <3 µm Noëlaerhabdaceae represent *E. huxleyi* type C and small Gephyrocapsa; 3–4 µm Noëlaerhabdaceae are associated to *E. hyxleyi* type B/C and *G. muellerae*; and >4 µm Noëlaerhabdaceae document *E. huxleyi* type A and B and *G. oceanica* patterns[43,62].

This study gathers specific morphological parameters of exactly 152,809 coccoliths that appear to reflect primary biomineralization features. The core MD07-3088 has been retrieved well above the lysocline. It is mainly made of homogenous fine-grained material that, together with high sedimentation rates (~300 cm/kyr during the Last Glacial and ~60 cm/kyr during the deglaciation and the Holocene[14]), prevent post-depositional fluid circulations. Besides, dissolution processes trigger a strong differential preservation of coccoliths keeping resistant specimens and losing delicate ones. The most delicate morphotypes belong to the Noëlaerhabdaceae family that represent the main coccolith of the assemblage. The smallest Noëlaerhabdaceae (<3 µm, i.e., mainly *E. huxleyi* type C and small Gephyrocapsa) depict the same exact pattern as the larger ones, with higher masses when the oceanic carbon reservoir is reconnected to the surface waters and bring $\mathrm{CO}_2$-rich waters into the photic zone (Supplementary Fig. 2), while such conditions could have favored the dissolution of coccoliths in the water column. At last, the three main increases observed during the deglaciation in the mean Noëlaerhabdaceae coccolith mass, would not be biased by diagenetic overgrowth that would also affect *C. leptoporus* and to a lesser degree *H. carteri*, that generally depict however, reducing coccolith masses during these time intervals (Supplementary Fig. 2). Indeed, diagenetic processes (dissolution or overgrowth)

would simultaneously impact all coccolith morphotypes, without any discrimination between morphotypes.

**Foraminifera abundance and mass data**. Planktonic foraminifera assemblages were determined at the LSCE (Laboratoire des Sciences du Climat et de l'Environnement) counting at least 300 specimens per sample. From three different depths (570, 950, and 990 cm), we weighted 30 individuals from the most abundant species (*Neogloboquadrina pachyderma* (sinistral and dextral coilings), *Globigerina bulloides*, *Globorotalia inflata*, *Turborotalita quinqueloba* and *Globigerinita glutinata*) for different sizes (>450 μm, 315–450 μm, 250–315 μm, 150–250 μm) to determine their mean weight. For this core, we obtained mean weights of: 7 ± 2 μg for *N. pachyderma*, *T. quinqueloba*, and *G. glutinata*, 18 ± 3 μg for *G. bulloides*, and 19 ± 5 μg for *G. inflata*. For *Globigerinella calida*, *Globigerina falconensis*, *G. ruber*, *Globigerina hexagonus* (representing <2% in all samples), *Neogloboquadrina dutertrei* and *Hastigerina digitata* (representing <0.5% in all samples), we assumed a mean weight similar to *G. bulloides*. At last, for *Globorotalia truncatulinoides*, *Globorotalia crassaformis*, and *Globorotalia hirsuta*, we assumed a mean weight similar to *G. inflata*. From the assemblage and the mean weight of the different species, we estimated the planktonic foraminifera calcite mass for each sample, $CaCO_{3pl.foram.mass}$ in mg/g as follow:

$$CaCO_{3pl.foram.mass} = \frac{N \times 2^{split}}{M} \times \sum_i (m_i \times X_i) \qquad (4)$$

where $N$ is the total amount of determined foraminifera (≥300), split is the number of split done before establishing a planktonic assemblage, $M$ is the total dry mass of the sample (g), $m_i$ is the mean weight of the species $i$ (mg), and $X_i$ the percentage of the species in the sample.

This approach is a first order estimate of the foraminifera mass percentage as it does not fully take into account smaller species often <150 μm (such as *G. uvula* and partly *T. quinqueloba*) and juveniles. Besides, for 16 depths (covering LGM, HS1, ACR, YD and the Holocene), we weighted 6 to 60 specimens of *G. bulloides* (the most abundant foraminifera) from different size ranges (150–200, 200–250, 250–315, 315–355, 355–400, and 400–450 μm) in order to statistically characterize potential weight changes within a narrow size range. Mean weights for the different size classes decrease of about 20% from LGM to Holocene, and of about 7 and 18% during HS1 and YD respectively. If similar weight decreases are observed within the other planktonic species, the magnitude of the changes in the overall weight (~20%) would be not sufficient enough to significantly change the estimated planktonic foraminifera mass flux. Indeed, because of the drastic increases within the planktonic foraminifera abundance during these time intervals (more than one order of magnitude), fluctuations in the planktonic foraminifera weights would imply changes in the flux of planktonic foraminifera calcite mass that remain within the error bars.

**Total $CaCO_3$ and organic carbon analyses**. Total $CaCO_3$ was determined at GEOPS laboratory using the vacuum-gasometric technique with a precision better than ±2%. 100 mg (±5) of crushed-dried sediments react with a few milliliters of HCl 6 N in a hermetic reaction chamber (22.4 cm³) that is connected to a manometer MANO MEX2-420 that measures the amount of outgassed $CO_2$. The system is calibrated so that 100 mg of $CaCO_3$ (100%) trigger a pressure rise to 1 bar.

Total organic carbon and nitrogen contents together with organic matter $\delta^{13}C$ analyses were obtained at the LSCE, using an Elementary Analyzer (Flash EA 1112) and the online continuous EA coupled with an Isotopic Ratio Mass Spectrometer (Finigan Delta + XP). The results are expressed in % C, % N, and in $\delta^{13}C$ per mL (‰) against the international standard V-PDB (Vienna Pee Dee Belemnite). Error margin is defined according to the source linearity checked for each run based on internal home-standard ($\Delta C < 0.03\%$ and $\Delta\delta^{13}C < 0.2‰$). A aliquot of <250 μm of dry sediment is softly leached with ultra-pure HCl 6 N to remove carbonate and dry at 50 °C. The samples were then crushed in a pre-combusted glass mortar for homogenization prior to carbon, nitrogen content and $\delta^{13}C$ analyses.

**XRF scanner measurements**. The high-resolution elemental analysis of Br and Ca was performed using an Avaatech profiling X-ray fluorescence (XRF) core scanner at Royal Netherland Institute for Sea Research (NIOZ) at a 1 cm downcore resolution. The external reproducibility of this core-scanner for Br and Ca in the range of the measurements is below 2% (1σ).

**Sedimentary POC: PIC ratio vs POC: PIC rain ratio (1/ρ)**. It remains difficult to evaluate the influence of changes in the TOC relative to the $CaCO_3$ (POC/PIC ratio, $1/\rho$) water column export and sedimentary burial on past $pCO_2$ variability. Indeed, the amount of particulate organic and inorganic carbon in the sediments is not necessarily directly related to the fraction exported from the surface waters. While it is probably reasonable to assume that the $CaCO_3$ accumulated in the sediment is representative of the PIC exported from the mixed layer to deep waters as core MD07-3088 was retrieved well above the lysocline (located around 3700 m depth nowadays[55]), it is probably not the case for TOC that might be more easily mineralized within the water column and upper sediments. However, at site MD07-3088, the combination of high sedimentary TOC contents (up to 1.9%),

high sedimentation rates, and homogeneous fine-grained lithology, lead us to assume that post-depositional remineralization processes associated to ($O_2$-rich) fluid circulations within the sediments must be of secondary importance. Moreover, it has been shown that it is in fact the oxygen exposure time that determines organic carbon degradation (i.e., ref. [63]), and based on the considerations above, we infer that labile organic compounds must have been buried rapidly, minimizing the potential for selective alteration. There is no doubt that remineralization processes that occurred within the water column (and particularly the twilight zone), altered the downward flux of POC, and thus the efficiency of carbon sequestration. However, the latitudinal distribution pattern of POC in surface sediments along the Chilean margin[55] reflects satellite-derived surface-ocean chlorophyll concentrations[64], which indicates that sedimentary TOC concentrations primarily reflect OC export rather than selective degradation processes within the water column. Besides, the high-latitude, iron-fertilized, near-shore ecosystem that characterize site MD07-3088, seems to be the perfect candidate to promote the sinking of organic matter to the deep seafloor[65–67]. Therefore, in order to consider a wide range of POC transfer efficiencies[65–67], we have tested the impact of BCP for HS1 and ACR, in cases where 10 to 50% of the exported POC is preserved within the sediments. Figure 5 indicates the influence of $1/\rho$ on seawater carbonate chemistry for cases ranging from photosynthetic processes to calcification processes only (solid black arrows) and for 10% to 50% of the TOC exported flux preserved in core MD07-3088 sediments for the HS1 and ACR periods.

**Data availability**. The data that support the findings of this study are available from the corresponding author (S.D.-A.) upon reasonable request.

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

## Acknowledgements

This research was financially supported by the French INSU/LEFE - CHICO (2013-2016) project, the Swedish Research Council (VR-349-2012-6278) and the Swiss National Science Foundation (grants PP00P2-144811 and PP00P2_172915 to S.L.J). The research has been conducted within the framework of the international IMAGES program and the MD159-PACHIDERME/IMAGES cruise, with technical support from the Institute Paul Emile Victor (IPEV). We express our thanks to G. Isgüder that assisted in foraminifera picking.

## Author contributions

All authors contributed extensively to this work. S.D.-A. prepared the manuscript and collected the organic and inorganic carbon data together with the coccolith data. Coccolith abundance and mass collection, using the SYRACO software, would not have been possible without L.B. and Y.G. E.M. and G.S. provided the XRF and foraminifera data. E.M., S.L.J., and G.S. particularly contributed to the redaction of the article.

## Additional information

**Competing interests:** The authors declare no competing interests.

