## [Peer Review File · Nature Communications]

Reviewers' comments:

Reviewer #1 (Remarks to the Author):

The present paper reports sedimentary records from several core sites in the Southern Ocean. The primary objective is to address the role of the carbonate counter pump in offsetting the ability of the biological carbon pump to sequester atmospheric CO₂ over glacial inter-glacial transitions. This is certainly an interesting idea that was originally suggested in a recent paper (Salter et al. 2014). The authors compile an impressive dataset to test this hypothesis. I think the paper is well written and of a high quality. I have some minor reservations about linking proxy data between core sites, the application of TOC and CaCO₃ sedimentary contents to the strength of the STP and CCP which are described in detail below.

I am not a paleoceanographer, but one concern I had during the review process was the notion that several different cores across the Southern Ocean were used. Frontal zonation in the Southern Ocean is critical for biogeochemistry and that these fronts have migrated over glacial cycles. Whilst the authors use the different locations of the cores to argue for broader response in the Southern Ocean, my feeling is that the regional structure of the cores is a limitation for the present study (with the exception WDC core used for atmospheric CO₂) More specifically, the authors rely heavily on coccolith abundance and mass data from the Chilean margin (MD07-3088). This core record may be affected by processes attributed to the margin (e.g. fertilization effects) that might not be representative of the subantarctic as a whole. The authors do in fact mention micronutrient supply from river run-off affecting the site (line 109-110). This is to my mind different from the glacial-interglacial iron fertilization effects proposed to explain changes in the biological carbon pump in the Southern Ocean (e.g. Martin 1990), which is linked to aeolian deposition, recently corroborated in the sub-antarctic Southern Ocean (Martinez-Garcia 2014, Science).

The authors compare the increase in productivity to macronutrient supply and cite opal fluxes in the AZ. However, to the best of my knowledge macronutrients are in sufficient supply here now and over the last glacial cycles and productivity was in fact limited by iron, not macronutrient supply.

I am not familiar with proxy data, but it seems to me that the spread of data in Figure 2a and 2b is quite large. The authors have plotted some kind of smoothing line for this data but with no representation of the residual variance. I don't know whether these smoothing techniques are commonly accepted with this kind of proxy data?

The authors use the Br/Ca ratio as a proxy for the strength of the biological carbon pump. Its use seems to be borne out of strong correlations between sedimentary TOC and Br, and Ca and CaCO₃. The authors then use the Br/Ca ratio as a proxy for the relative strength of the STP (soft-tissue pump) and CCP (calcium carbonate pump).

My first question is why not simply use the TOC:CaCO₃ ratio in the sediments? Does it not report the same thing, i.e. changes in the sedimentary content of TOC and CaCO₃? By using bromine to calcium the authors introduce residual variance in TOC:CaCO₃ ratio that originates from the Br to TOC and Ca to CaCO₃ correlations.

Aside from this methodological aspect, there is to my mind a conceptual discrepancy in using these sedimentary records for reconstructing the strength of the STP and CCP. The strength of the STP is normally taken at the ventilation depth. There is significant remineralization of POC below the ventilation depth, i.e. before it arrives at the sediment interface. It is well established in the present ocean that the attenuation of POC flux with depth is highly dependent on phytoplankton community structure comprising export assemblages. This is the case for different diatom species, i.e. those that form resting spores and settle with low Si:C ratios, and those that settle primarily as vegetative cells that are characterized with high Si:C ratios (Rembauville et al. 2015,

Rembauville et al Salter et al, 2012). Patterns in export flux at different times during the glacial transitions is marked by major shifts in these kinds of diatom species (i.e. Jacot des Combes et al. 2008; . Abelmann, 2006). Although these changes would not be detected in sedimentary opal records, they would have a major implication for extrapolating sedimentary TOC content to the strength of the STP. Unfortunately this is often neglected in paleoceanographic studies. Additionally the attenuation of POC is known to differ widely between coccolith flux and diatom flux. Again this has implications for linking TOC content to the strength of the STP, which is particularly relevant here considering the emphasis on coccolith export over diatoms.

The situation for CaCO₃ is similarly complicated. There is evidence of significant biologically mediated dissolution of CaCO₃ above the chemical lysocline, which may account for 60-80% in 500-1000m of the ocean (Milliman et al., 1999). This is likely different for coccolith versus foraminifer calcite. What is the contribution of calcifying benthic foraminifer to sedimentary CaCO₃ concentrations? These are represented in your Br:Ca estimate of the relative strength of the STP and CCP, but would play no role in the atmospheric CO₂ sequestration over the same timescales.

The authors also have their terminology confused. For example in lines 187-188 they state: "In such a scenario, the CCP would be instrumental in weakening the marine STP". To clarify, the biological carbon pump (BCP) refers to the sequestration of atmospheric CO₂. It is composed of two components: the soft-tissue pump (STP) which is a sink of CO₂ through the settling of organic matter, and the carbonate counter pump (CCP) which is a source of atmospheric CO₂ if calcification occurs above the ventilation depth. The net sequestration of atmospheric CO₂ by the BCP is therefore a balance between the STP and CCP components. To summarise, it is incorrect terminology and misleading to say that the CCP can weaken the STP. Rather it weakens the net efficiency of BCP to sequester atmospheric CO₂

Salter et al. 2014 is a significant study for this paper as it is the first and only attempt to directly quantify the effect of the CCP on the efficiency of the BCP in the Southern Ocean. Although study of Salter et al. 2014 is cited in the introduction material, it is somewhat misrepresented and overlooked in the results and discussion section of the paper in the following places:

(1) In the introduction the authors state ..."the relative contribution of the CCP has been largely ignored, despite it's fundamental role in the marine carbon cycle"

The effect of the CCP in carbon cycling across different iron-fertilized productivity regimes, as an experimental case study for represent glacial-interglacial transitions (Pollard et al. 2009 Nature), had been ignored up until the publication of Salter et al. 2014. This would be the correct place to reference this paper in the introduction by stating that it has been shown to have an effect in contemporary sediment trap studies.

(2) In particular Salter et al. 2014 quantifies changes of CCP strength in the sub-Antarctic across different iron-fertilized productivity regimes, thought to represent glacial-interglacial transitions. The study measures calcite fluxes and shows that a 7-10 fold increase in CaCO₃ flux corresponds to a weakening of the BCP by the CCP of 6-30%. This is comparable to the 3-10 fold increase in CaCO₃ burial reported in the present study (line 179-180) used to invoke an effect on atmospheric CO₂ "amount of burial CaCO₃ produced by coccolithophores and foraminifera, respectively (Fig. S7), and coincide with prominent rises in atmospheric pCO₂. These patterns suggest that the SAZ became a significant source of atmospheric CO₂ during the last deglaciation" The authors should make reference to the quantification of this effect in Salter et al. 2014, especially considering it corresponds to the similar increases in carbonate export/burial.

(3) Line 193-195 – It should be cited here too as this has already been quantified.

Reviewer #2 (Remarks to the Author):

In my respect the paper of Duchamp-Alphonse et al., addresses an important and often neglected aspect of the marine-atmosphere carbon cycle. Despite having read about the CCP before, somehow this aspect has slipped my attention, when looking at the glacial-interglacial carbon cycle. With respect to other publications, focusing on one or the other processes that influenced the deglacial rise in CO₂, I think this is a very valuable contribution, which will be of high interest to the scientific community.

Despite being no expert on coccolithophores, I think that the paper relies on a decent and robust methodology. As the age model of MD07-3088 is based on the study of Siani et al. (2013), I have no doubt in the timing (which is crucial to the overall story) of the processes discussed in the current paper.

In the text and methods, the authors should address following points:

- In line 42 the authors mention that the Southern Ocean is one of the main oceanic sources of natural CO₂ to the atmosphere. While this is technically correct, one should take into account that under current (anthropogenic) pCO₂ values, the Southern Ocean acts as a sink of CO₂ rather than a source (Gruber et al., 2009). To simplify matters, the authors could refer to the pre-industrial Southern Ocean.
- In line 159 ff. the origin of Ca (marine vs. terrestrial) is discussed. In my respect it would help, if a supplementary figure would be included, showing the Ca/Al or Ca/Ti-ratio over the entire record. As Al and/or Ti are of predominantly terrestrial origin, I think this figure would help to assess whether the marine source of Ca or CaCO₃ changed or not.
- In lines 176-177 the authors exclude changes in TOC as the driver for the transient declines in the Br/Ca-ratio. However, comparing the Br/Ca- (Fig. 2d) and TOC-records (Fig. S5b), I'm afraid that Br/Ca follows TOC from about 14.5 cal ka BP toward the present. While the authors state that TOC increases throughout the time intervals in question (HS1 and YD), I think that the YD is marked by a slight decrease in TOC, matching the pattern observed in Br/Ca. While I see no problem for HS1, I think the authors should be a bit more conservative, when discussing the YD (e.g. line 183).
- When estimating the mean weight of different species (methods), I wonder if the mean weight changed from the glacial to the interglacial, as the presence of CO₂-rich (more corrosive) waters might affect the mean weight from one sample to another.
- The Br-K β and Rb-K α lines in the Avaatech XRF-scans tend to overlap to some extent. Did you check whether or not this might have affected your Br-record? As the Rb-K α line is usually pretty weak I don't think this will be a major issue, but I think it would be good to check it anyway.

Some minor typos or technical problems:

- Line 457: Please cite the references for WOA09 nitrate and temperature. Are the fronts based on Orsi et al. (1995)? If so, please also include this reference.
- Lines 469 & 470: I'm aware that Siani et al. (2013) call the species they used *C. wuellerstorfii* in the caption of their Figure 2d. However, I think that's a typo (they call it *wuellerstorfii* in their method section. Unfortunately, I also think that *wuellerstorfii* is still not correct. In my view, you should go with the original 1866 definition of Schwager and call it *C. wuellerstorfi*.
- Supplements: I think you mean 152,809 and not 152 809.
- Capture of Figs. S3 & S7: Please replace *specie* by *species*.
- Reference 34: Is there any manuscript in prep. or under review?
- Please state in the captions of Figs. 2; S1; S3; S4; S5; S7 what you mean with LGIT. Last Glacial Interglacial Transition, I assume?

I would think these topics are fairly straightforward, hence my recommendation is minor revisions.

Gruber et al., 2009; Oceanic sources, sinks and transport of atmospheric CO₂; DOI: 10.1029/2008GB003349

Reviewer #3 (Remarks to the Author):

Duchamp-Alphonse and coauthors present new records of coccolith abundance and mass, and new XRF records, from a core on the Chilean Margin with the aim of elucidating temporal changes in the strength of the soft tissue and carbonate pumps, with implications for deglacial pCO₂ rise. The subject matter is relevant to the paleoclimate community, and the data are novel and interesting. However, some significant re-writing is needed in order to better showcase key findings and convince readers that the interpretations made are robust. Specifically, arguments need to be better set up so that new findings are presented in a logical order and in the context of the state-of-the-art and all supporting data. The interpretations made will then be easier to follow and more convincing. I have made detailed comments below on specific sections, which I hope the authors will find useful. I also have a number of questions regarding the interpretations, detailed below. Given that more main text figures are allowed in Nature Communications compared to Nature, the authors may consider moving the more important figures from the supplement to the main text when revising the paper.

Specific comments to authors:

A few relevant papers that you may want to consider: Flores et al 2012 (Frontiers in Microbiology), on SAZ coccolithophore productivity in the Pleistocene. Saavedra-Pellitero et al (Paleoceanography, 2017) on potential coccolithophore impacts on the C cycle in the Pacific sector of the Southern Ocean. Meier et al 2014 (Marine Micropal) on coccolith mass changes during Termination II in the context of carbonate chemistry changes.

Lines 86-93 belong in the introduction. The role of coccolithophores in the STP and the CCP and the importance of the rain ratio to climate is vital background to your paper, and needs to be described more clearly to set up your study.

Line 91: changes in their abundance/mass relative to what?

Spelling: Noëlaerhabdaceae

Why is the Chilean Margin an ideal location to answer your scientific questions on deglacial CO₂ rise and the STP/CCP? Most of the intro seems to focus on the Southern Ocean, so you need to connect the two areas better before the results are presented. Also, you mention AAIW and PDW, but what surface water masses are affecting the productivity of coccolithophores above your core and how is this thought to have changed since the LGM? Is there upwelling of AAIW above your site? Is it considered to be in the SAZ? What are the modern disequilibrium wrt CO₂ (Source or sink to the atmos?) and upwelling conditions?

Figure 2: For the new records I suggest plotting data points connected with a line, rather than a smoothed line with points in the background. Add running averages as well if you like, but do so for all records in the figure if you choose to. Different fonts, font sizes, tick mark lengths etc are present in the same figure and some axis numbers/ titles are squashed. The yellow bars in figure S2 and S5 cover different intervals from each other and from those in other figures.

One feature of the data that immediately strikes me is the correspondence between higher Noel abundance and higher (average) Noel mass. This is intriguing, because I'd expect higher fertility/productivity intervals to be associated with a larger number of smaller coccolithophore cells (as is the case in modern blooms). The authors interpret this as reflecting periods of increased upwelling of SAMW, which brings both macronutrients and [CO₂] to the surface, stimulation both primary

productivity and coccolith calcification (increasing cellular PIC:POC ratio I assume, although this is not explicitly stated). Has a similar correspondence between inferred primary productivity and coccolith degree of calcification been seen in other cores? At the moment, no comparison with published coccolith mass or abundance data for the last deglaciation is made, although such data exist (e.g. the cores studies in Beaufort et al 2011). A detailed comparison with published data would increase the impact of your paper.

Line 91: by mass, do you mean cellular PIC:POC ratio?

Line 94: "coccolith abundance and mass" of the dominant Noel family... in samples from a marine sediment core...

Line 103: "Noëlaerhabdacea coccolith abundance and by inference surface ocean fertility" This is a sweeping statement; the authors need to clearly explain what this inference is based on, citing supporting evidence, before making it. As pointed out in the introduction, carbonate (and therefore coccolith) accumulation in sediments can be affected by many processes. If you feel you have confidently ruled out dissolution/dilution influences (detailed in the SI?) then say so.

Line 116: be more specific about what you mean by "large-scale ocean circulation changes".

Line 117: "the downcore primary production record mimics opal fluxes in the AZ at sites influenced by similar processes. This suggests that the inferred changes in productivity are neither limited to a specific phytoplankton group nor a specific area, but highlights a large-scale sensitivity of phytoplankton growth to ocean circulation and nutrient supply from below"

Be careful about calling your Noel abundance record a "primary production record"; at best, it represents coccolithophore export productivity. Do sedimentation rates also increase at times of inferred higher productivity? Do your coccolith abundance records have the same structure when expressed as MARs?

To say that the entire region and all phytoplankton groups are behaving the same in terms of primary productivity because you see covariation between your Noel abundance record from the Chilean Margin and an opal record from a site in the Atlantic sector of the PFZ is a gross oversimplification. It is true that coccolithophores and diatoms often increase together in upwelling regions and also in Fe fertilization experiments, and much has been published on this. However, the SAZ/AZ is an extremely complex region oceanographically, and there is much heterogeneity in terms of productivity in these (shifting) frontal regions (for example, see Hillenbrand and Cortese 2006 for a longer-term perspective). Many records of opal (and other export productivity proxies) exist for your time interval and region and if you'd like to extrapolate your results to the whole region a more thorough analysis of published records is needed, in my opinion.

Line 107: My understanding is that SAMW forms in the SAZ, subducts into the pycnocline and flows northwards at depth, providing nutrients to the low-latitudes (<30°) via advective mixing and upwelling. Ref 18 shows modelling evidence for SAMW fueling productivity between 30S and 30N – so I am confused as to whether you are saying that upwelling of SAMW occurs at your site (and is the dominant source of macronutrients) or not. Please clarify.

Line 115: to use this as an argument, you need to remind the reader what the $\Delta\delta^{14}\text{C}$ and $\Delta\delta^{13}\text{C}$ records mean.

Lines 114-125: I am not able to follow your arguments here. First you say that coccolithophore and diatom productivity appear to be going hand in hand throughout the region, and that all groups seem to be stimulated by upwelled nutrients, then you go on to say that your data support existing records showing a shift from diatom to coccolithophore productivity due to silica leakage. Very confusing.

Again (similar to the statement at line 103), the statement "mean coccolith mass, a proxy for sea-

surface [CO_{2aq}]” is controversial, and not supported by arguments or references in the text where it appears. You need to be more careful with your wording, and order your arguments appropriately (i.e. We observe X and Y. Because of this and this, supported by these other things/existing studies, and because we can exclude a and b as potential drivers of coccolith mass in our core, we hypothesize that coccolith mass reflects degree of calcification and is responding to changes in [CO_{2aq}] driven by changes in upwelling intensity).

I would like to see the mean coccolith size data (for the same size classes as mass) to be convinced that changes in coccolith mass are independent of size, as stated in the text, and are “mainly an effect of increasing coccolithophore calcification” (does this mean increasing inferred cellular PIC:POC ratios?). Also, in my mind “increasing coccolithophore calcification” with no size changes equates to “more heavily calcifying morphotypes”, unless you are saying there is an increase in coccolith production rate only? In which case the mean mass should stay the same when coccoliths/g increases. I don’t understand how you can exclude the possibility that the dominant morphotype is changing.

Are the [CO_{2aq}] estimates suggested for the last deglaciation in your core anywhere near the range used in experiments (don’t forget about the temp effect on [CO_{2aq}])? i.e. would you expect strong sensitivity? Same for the comparisons with fossil data (refs 29-32) – which of these represent a similar range of [CO_{2aq}] variability, and how does sensitivity of mass to [CO_{2aq}] compare?

Please explain how increased upwelling intensity (of SAMW?) during HS1 and the YD leads to increased SST, when usually enhanced upwelling is associated with decreased SST.

Has the ratio of Br/Ca ever been applied before to estimate the strength of the STP/CCP, or is your study the first application of this proxy? How is it better than using TOC/CaCO₃ to characterize the export Corg to CaCO₃ ratio (for ex. is the Br signal preserved when OC is not?)? I don’t fully understand how you derive STP strength (is this the same as “biological pump efficiency” mentioned on Fig. 2?), so some clarification is needed here. Again, the set-up of the arguments needs work. For example, starting a sentence with “this pattern could potentially be explained by lower TOC preservation” immediately puts doubt in the reader’s mind, even though you then go on to explain why this is unlikely.

For the last ~6 kyr, Br/Ca is high, but coccolith abundance is low. What is your interpretation of this Holocene interval?

Is 1/rho standard notation for POC/PIC ratio? I thought that rho is usually used for density. Figure 3, as it is currently discussed in the text, doesn’t really add much to the paper. Most of the important info pertaining to this figure seems to be in the methods and caption.

Responses to Reviewers on the manuscript by Duchamp-Alphonse et al. "Enhanced ocean-atmosphere carbon partitioning via the carbonate counter pump during the last deglacial"

Ref: NCOMMS-17-18833A

Reviewer #1 (Remarks to the Author):

Comment #1 (C1): The present paper reports sedimentary records from several core sites in the Southern Ocean. The primary objective is to address the role of the carbonate counter pump in offsetting the ability of the biological carbon pump to sequester atmospheric CO₂ over glacial inter-glacial transitions. This is certainly an interesting idea that was originally suggested in a recent paper (Salter et al. 2014). The authors compile an impressive dataset to test this hypothesis. I think the paper is well written and of a high quality. *I have some minor reservations* about linking proxy data between core sites, the application of TOC and CaCO₃ sedimentary contents to the strength of the STP and CCP which are described in detail below.

I am not a paleoceanographer, but one concern I had during the review process was the notion that several different cores across the Southern Ocean were used. Frontal zonation in the Southern Ocean is critical for biogeochemistry and that these fronts have migrated over glacial cycles. Whilst the authors use the different locations of the cores to argue for broader response in the Southern Ocean, my feeling is that the regional structure of the cores is a limitation for the present study (with the exception WDC core used for atmospheric CO₂)

Reply #1 (R1): We certainly agree with Reviewer #1. Our study focuses on documenting the role the subantarctic CCP might have exerted on the Biological Carbon Pump in the past as highlighted lines 24-27, 88-91, 122-123, 245-247 and 253-256 in the revised manuscript. The comparison with other records merely relates to the impact of enhanced upwelling/vertical mixing at the Antarctic divergence, and associated increase in nutrient and CO_{2aq} concentrations in SSW, but also SAMW and AAIW as surface waters are advected equatorwards. Therefore, in the revised version of the manuscript, we emphasize that core MD07-3088 remained within the Subantarctic zone during the entire duration of the last glacial termination (l. 275-278). We characterise the position of the core with respect of the regional oceanic circulation pattern and demonstrate that neither the Subtropical nor the Subantarctic fronts reached the core site during the time interval under consideration (Haddam, 2016; Haddam et al., Submitted) (New section "Material and site description" within the "Methods").

C2: More specifically, the authors rely heavily on coccolith abundance and mass data from the Chilean margin (MD07-3088). This core record may be affected by processes attributed to the margin (e.g. fertilization effects) that might not be representative of the subantarctic as a whole. The authors do in fact mention micronutrient supply from river run-off affecting the site (line 109-110). This is to my mind different from the glacial-interglacial iron fertilization effects proposed to explain changes in the biological carbon pump in the Southern Ocean (e.g. Martin 1990), which is linked to aeolian deposition, recently corroborated in the sub-antarctic Southern Ocean (Martinez-Garcia 2014, Science). The authors compare the increase in productivity to macronutrient supply and cite opal fluxes in the AZ. However, to the best of my knowledge macronutrients are in sufficient supply here now and over the last glacial cycles and productivity was in fact limited by iron, not macronutrient supply.

R2: Core MD07-3088 has been retrieved off Southern Chile (46°04 S; 075°41 W). Thanks to its proximity with the continent, this core shows several characteristics that are particularly suitable to fulfill our research objectives (lines 265-291 in the revised manuscript). We agree with Reviewer #1 that its specific characteristics were not sufficiently documented in the initial version of the manuscript. Particularly, it is true that the difference between the South Atlantic sector of the SAZ (that is under the direct influence of iron-rich dust plumes originating from Patagonia, and where previous BCP studies have been mainly published), and the South East Pacific sector (that is where macronutrients and not micronutrients are limiting algal growth), was not well underlined. We emphasize it now in the introduction, lines 63-64, 70-75, 88-91 and 104-109 and lines 134-140).

C3: I am not familiar with proxy data, but it seems to me that the spread of data in Figure 2a and 2b is quite large. The authors have plotted some kind of smoothing line for this data but with no representation of the residual variance. I don't know whether these smoothing techniques are commonly accepted with this kind of proxy data?

R3: Smoothed curves are indeed widely used in palaeoceanography to document the main trends of high-resolution data (Siani et al., 2013 and Gottschalk et al., 2016 as examples for Nature Communications publications). In the present study, such procedure is crucial to compare our results with lower-resolution studies (in situ $\Delta^{14}\text{C}$ measurements (Siani et al., 2013), opal flux core TN057-13-4PC (Anderson et al., 2009), and oceanic ΔpCO_2 from ODP 1238 (Martinez-Boti et al., 2015)). As recommended by Reviewer #3, we connected

data points with lines in the revised version of the manuscript to also represent first order data variance (Figs 2-3 and Supplementary Figs. 1-4).

C4: The authors use the Br/Ca ratio as a proxy for the strength of the biological carbon pump. Its use seems to be borne out of strong correlations between sedimentary TOC and Br, and Ca and CaCO₃. The authors then use the Br/Ca ratio as a proxy for the relative strength of the STP (soft-tissue pump) and CCP (calcium carbonate pump).

My first question is why not simply use the TOC:CaCO₃ ratio in the sediments? Does it not report the same thing, i.e. changes in the sedimentary content of TOC and CaCO₃?

R4: Yes, Br/Ca and TOC/CaCO₃ ratios essentially both report changes in the sedimentary content of TOC relative to CaCO₃. Biogenic CaCO₃ and Total Organic Carbon were quantified on discrete samples using standard techniques (see Methods). The downcore distribution of Br and Ca on the other hand has been obtained by X-Ray Fluorescence (XRF). This widely-used technique is extremely useful to semi-quantitatively determine the elemental composition of sediment cores, allowing for very high-resolution analyses. The technique is robust, non-destructive and automatically scans sediment core sections. Indeed, in the present study, Ca_{XRF} and Br_{XRF} data (among over elements) have been obtained every centimeter, along a 18 m-long core section (~ 1800 samples) in less than 10 days. Our study highlights the good correspondence between discrete and XRF scanning measurements (Supplementary Fig. 4), justifying the use of Br/Ca to reconstruct changes in the sedimentary TOC/CaCO₃ at high temporal resolution.

C5: By using bromine to calcium the authors introduce residual variance in TOC:CaCO₃ ratio that originates from the Br to TOC and Ca to CaCO₃ correlations.

R5: We agree with the reviewer. However, the residual variance that is inherent to the use of Br/Ca instead of TOC/CaCO₃ is negligible compared to the magnitude of the decadal Br/Ca ratios. We added uncertainties for the CaCO₃ (+/- 0.83%) and TOC (+/- 0.23%) curves derived from Ca_{XRF} and Br_{XRF} analyses within the Supplementary Figure 4 caption to further document this point (lines 42 and 46 of the supplementary information, respectively).

C6: Aside from this methodological aspect, there is to my mind a conceptual discrepancy in using these sedimentary records for reconstructing the strength of the STP and CCP. The strength of the STP is normally taken at the ventilation depth. There is significant remineralization of POC below the ventilation depth, i.e. before it arrives at the sediment interface. It is well established in the present ocean that the attenuation of POC flux with depth is highly dependent on phytoplankton community structure comprising export assemblages. This is the case for different diatom species, i.e. those that form resting spores and settle with low Si:C ratios, and those that settle primarily as vegetative cells that are characterized with high Si:C ratios (Rembauville et al. 2015, Rembauville et al Salter et al, 2012). Patterns in export flux at different times during the glacial transitions is marked by major shifts in these kinds of diatom species (i.e. Jacot des Combes et al. 2008; Abelmann, 2006). Although these changes would not be detected in sedimentary opal records, they would have a major implication for extrapolating sedimentary TOC content to the strength of the STP. Unfortunately this is often neglected in paleoceanographic studies. Additionally the attenuation of POC is known to differ widely between coccolith flux and diatom flux. Again this has implications for linking TOC content to the strength of the STP, which is particularly relevant here considering the emphasis on coccolith export over diatoms.

R6: Paleoceanographic studies indeed often rely on the burial of TOC and biogenic CaCO₃ to reconstruct temporal changes in primary/export production. While many studies have highlighted a direct link between particle export and burial, we recognize that this link is modulated by multiple factors difficult to quantify based on sedimentary measurements alone. We note however, that only very few paleoceanographic studies have addressed this issue as thoroughly as we did. As summarized by Reviewer #1, some phytoplanktonic taxa may export carbon more effectively than others (diatom species *Eucampia antarctica* as an example) (Smetacek, 2004), while selective preservation of phytoplankton remains can affect particle composition during settling and burial. Therefore, changes in phytoplankton assemblages may well affect carbon export, with implications for the amount of organic carbon buried and preserved in the sedimentary record. Unfortunately, no diatom assemblage record exists for core MD07-3088 to unravel the influence floral assemblage changes may exert on sedimentary TOC contents. A few deglacial diatom assemblage records (relative abundances (%)) have been reported for the Antarctic Zone of the Southern Ocean (e.g. Abelmann et al., 2006; Jacot des Combes et al., 2008), but their low temporal resolution precludes making inferences about sub-millennial changes in export production. The only comparative down-core micropalaeontological data that exists so far to the best of our knowledge, is represented by the coccolith assemblage record from ODP site 1233, located in the Pacific sector of the Subantarctic Zone (Saavedra-Pellitero et al., 2011) and has been included within Fig. 2, and discussed within the paper (lines 126, and 148-153 of the revised text). The sediment-trap studies referred to by Reviewer #1 mainly deal with data-sets from traps deployed at 2000 m (Rembauville et al., 2016 (only one sediment trap was deployed at 1500 m) or below 3000 m water depth (Salter et al., 2010; Salter et al., 2012)). We are aware that selective preservation is playing a major role as particle sink through the water-column. However, these processes increase as a function

of depth, and we believe that core MD07-3088, retrieved from intermediate water depth (1536 m), and characterized by very high sedimentation rates (~ 300 cm/kyr during the Last Glacial and ~ 60 cm/kyr during the deglaciation and the Holocene (Siani et al., 2103)), is ideally located to minimize the typically obfuscating effects of selective preservation (see in the Methods section, the new paragraph “Material and site description”). Indeed, in such a context, particles might have rapidly sunk through the water column, and a substantial portion of organic components (and most carbonate fraction) is likely to have reached the sea-floor minimally altered. Therefore, temporal changes in sedimentary POC concentration are primarily documenting changes in primary production, while selective preservation likely played a secondary role. Moreover, core MD07-3088 is mainly constituted of silt-bearing clays that would prevent post-depositional (O₂-rich) fluid circulations, and by inference remineralization processes within the sediment (see methods I 288-291). More regionally, it appears that i) the latitudinal distribution pattern of POC in surface sediments along the Chilean margin closely resembles the chlorophyll-like average pigment concentration derived from satellite data (Thomas et al., 1994, Hebbeln et al., 2000), and ii) long-term trends in TOC contents recorded at site MD07-3088 mirror those from sites 1233 and 1234, located in the central Chilean margin (Muratli et al., 2009), thus pointing to the good suitability of sedimentary POC as a paleoproductivity indicator in the area, and probably more generally, in Southern Ocean shallow-water environments (Smetacek et al., 2012).

As suggested by Reviewer #1, we now specify in the manuscript that quantitative changes in BCP efficiency have to be considered with some caution and provide two examples of possible POC:PIC changes for HS1 and the ACR, assuming that 20 or 30% of the exported POC is preserved in the sediments (Figure 4 and lines 238-247).

C7: The situation for CaCO₃ is similarly complicated. There is evidence of significant biologically mediated dissolution of CaCO₃ above the chemical lysocline, which may account for 60-80% in 500-1000m of the ocean (Milliman et al., 1999). This is likely different for coccolith versus foraminifer calcite. What is the contribution of calcifying benthic foraminifer to sedimentary CaCO₃ concentrations? These are represented in your Br:Ca estimate of the relative strength of the STP and CCP, but would play no role in the atmospheric CO₂ sequestration over the same timescales.

R7: Several problems/discrepancies have appeared in addressing epipelagic dissolution of CaCO₃ (Milliman et al., 1999). Coccoliths seen in trap material, almost always remain intact (Milliman et al., 1999). In the same way as Milliman et al., (1999) find evidence suggesting supralysocline dissolution, Beaufort et al., (2007), show that acidification does not significantly change the weight and size of coccoliths. Besides, there is ample evidence from the literature suggesting good preservation of coccoliths (and by inference of CaCO₃) in sediments collected well above- (Howard and Prell, (1994): MD84-55, 2230m and MD84-529, 2600 m; Flores et al., (2000): PS2709-1, 2707 m and PS2703-1, 2747m; Hodell et al., (2000): ODP Hole 704, 2532 m; Saavedra-Pellitero et al., (2011): ODP 1233, 838 m) but also close to the lysocline (Flores et al., (2003), (2012): ODP Site 1089, 4620 m; Saavedra-Pellitero et al., (2017): PS75/059-2, 3613 m) within the Southern Ocean, with specific implications for paleoceanographic and paleoclimatic reconstructions. Similarly to discussion above focusing on organic carbon burial, we contend that the relative shallowness of the sediment core (1536 m) as well as high sedimentation rates, and a silt-bearing clay lithology would favor good preservation of sedimentary carbonates phases (see in the methods section “material and site description”). Furthermore, as presented in detail in the Methods section (in “coccolith taxonomy and preservation” (lines 382-414), coccolithophore calcification patterns show no evidence of differential preservation. Particularly, heavier Noëlaerhabdaceae coccoliths are observed when the oceanic carbon reservoir is reconnected to the surface waters and bring more corrosive waters into the photic zone (Fig. 2 and Supplementary Figs. 2-3). Concerning planktic foraminifera, no dissolution pattern was observed on the tests and a particularly low number of fragments has been counted for this core (usually less than 1% and never exceeding 4%). We are thus confident that changes in coccolith and foraminifera calcification patterns primarily reflect changes in PIC fluxes at the ventilation depth.

Reviewer #1 raises herein the urgent need for species-centered approaches, and studies coupling geochemical analyses with micropalaeontological data in order to better understand the regulation biogeochemical fluxes might exert on the C cycle. We are convinced that the overall good correlation of our micropalaeontological and geochemical data with geochemical data from EDC ice core and from a wide range of locations within the SO and the EEP, highlight palaeoceanographic and palaeoclimate signals that are not primarily affected by dissolution or diagenetic processes.

Concerning benthic foraminifera we agree that they are represented in the Br/Ca curve while they do not play a role in the BCP strength. The clarification related to this potential issue is now presented within the Method section, lines 440-443). We agree with Reviewer #1 that the present study should focus on qualitatively documenting the leverage CCP changes is imposing on [CO_{2ad}] and therefore atmospheric pCO₂ as i) the fraction of planktic producers within the bulk CaCO₃ content is not exactly known, ii) primary/export production cannot be quantified based on sedimentary TOC (and CaCO₃) sedimentary record, an issue inherent to many paleoceanographic studies.

C8: The authors also have their terminology confused. For example, in lines 187-188 they state: “In such a scenario, the CCP would be instrumental in weakening the marine STP”. To clarify, the biological carbon pump

(BCP) refers to the sequestration of atmospheric CO₂. It is composed of two components: the soft-tissue pump (STP) which is a sink of CO₂ through the settling of organic matter, and the carbonate counter pump (CCP) which is a source of atmospheric CO₂ if calcification occurs above the ventilation depth. The net sequestration of atmospheric CO₂ by the BCP is therefore a balance between the STP and CCP components. To summarise, it is incorrect terminology and misleading to say that the CCP can weaken the STP. Rather it weakens the net efficiency of BCP to sequester atmospheric CO₂

R8: We agree with Reviewer #1 (and Reviewer #3, see Reply # R56) that the terminology related to the Biologic Carbon Pump can be confusing and can have different meanings in different communities. Sometimes, the “BCP strength” refers to the amount of organic carbon that is exported from the surface to the deep ocean, while the “BCP efficiency” refers to the balance between the exported carbon and the carbon that is resupplied to the surface by the physical ocean circulation (upwelling). Further constraints related to the nutrient cycling would be needed to have accurate information about the BCP efficiency. Besides these definitions do not clearly take into account the CCP.

For the sake of clarity, we decided to follow Reviewers #1 and #3’s recommendations and now define the BCP as the sum of the STP and CCP (without any consideration of ocean circulation changes). Therefore, we changed STP by BCP each time the STP and CCP were both considered (for example in “In such a scenario, the CCP would be instrumental in weakening the marine STP”, STP has been changed to “In such a scenario, the CCP would have contributed to weaken the marine BCP...” (lines 250-252).

C9: Salter et al. 2014 is a significant study for this paper as it is the first and only attempt to directly quantify the effect of the CCP on the efficiency of the BCP in the Southern Ocean. Although study of Salter et al. 2014 is cited in the introduction material, it is somewhat misrepresented and overlooked in the results and discussion section of the paper in the following places:: (1) In the introduction the authors state ...”the relative contribution of the CCP has been largely ignored, despite it’s fundamental role in the marine carbon cycle”

R9: Yes, Salter et al. emphasized the role of the CCP particularly in the Southern Ocean, and it is now cited in the introduction (Ref. 31 lines 83 and 86) and the discussion parts (lines 236, 238 and 256).

C10: The effect of the CCP in carbon cycling across different iron-fertilized productivity regimes, as an experimental case study for represent glacial-interglacial transitions (Pollard et al. 2009 Nature), had been ignored up until the publication of Salter et al. 2014. This would be the correct place to reference this paper in the introduction by stating that it has been shown to have an effect in contemporary sediment trap studies.

R10: Indeed, Pollard et al., 2009 address an important and often neglected aspect of the Biological Pump efficiency related to natural iron fertilization. The study is mainly based on annual POC and chlorophyll a fluxes near Crozet Islands and Plateau, and highlights the control iron inputs might exert on STP efficiency within the modern Subantarctic Zone. This is a perfect natural case study that support the “iron hypothesis” (Martin et al., 1990). We add this reference in the introduction (Ref. 6, l. 43), when discussing the iron (Fe) limitation on phytoplankton growth in the modern Southern Ocean.

C11: (2) In particular Salter et al. 2014 quantifies changes of CCP strength in the sub-Antarctic across different iron-fertilized productivity regimes, thought to represent glacial-interglacial transitions. The study measures calcite fluxes and shows that a 7-10 fold increase in CaCO₃ flux corresponds to a weakening of the BCP by the CCP of 6-30%. This is comparable to the 3-10 fold increase in CaCO₃ burial reported in the present study (line 179-180) used to invoke an effect on atmospheric CO₂ “amount of burial CaCO₃ produced by coccolithophores and foraminifera, respectively (Fig. S7), and coincide with prominent rises in atmospheric pCO₂. These patterns suggest that the SAZ became a significant source of atmospheric CO₂ during the last deglaciation” The authors should make reference to the quantification of this effect in Salter et al. 2014, especially considering it corresponds to the similar increases in carbonate export/burial.

R11: We thank Reviewer #1 for highlighting this important aspect. Since the increases in burial CaCO₃ produced by coccolithophores and planktonic foraminifera (triggering decreasing Br/Ca ratios) at times of atmospheric pCO₂ rises are in the same order of magnitude (7-10 fold and up to 20 fold increases for coccolithophores and foraminifera respectively) than those reported by Salter et al., (2014) under comparable conditions (9 and 6-8 times higher at iron-fertilized sites), it seems important to make reference of the quantification proposed by Salter et al. (2014). Therefore, this reference has been added to the text in the discussion and better discussed herein (lines 233-238).

C12: (3) Line 193-195 – It should be cited here too as this has already been quantified.

R12: This is true. We also made reference to Salter et al., 2014 in this part of the manuscript (lines 253-256).

Reviewer #2 (Remarks to the Author):

In my respect the paper of Duchamp-Alphonse et al., addresses an important and often neglected aspect of the marine-atmosphere carbon cycle. Despite having read about the CCP before, somehow this aspect has slipped my attention, when looking at the glacial-interglacial carbon cycle. With respect to other publications, focusing on one or the other processes that influenced the deglacial rise in CO₂, I think this is a very valuable contribution, which will be of high interest to the scientific community.

Despite being no expert on coccolithophores, I think that the paper relies on a decent and robust methodology. As the age model of MD07-3088 is based on the study of Siani et al. (2013), I have no doubt in the timing (which is crucial to the overall story) of the processes discussed in the current paper.

In the text and methods, the authors should address following points:

C13: • In line 42 the authors mention that the Southern Ocean is one of the main oceanic sources of natural CO₂ to the atmosphere. While this is technically correct, one should take into account that under current (anthropogenic) pCO₂ values, the Southern Ocean acts as a sink of CO₂ rather than a source (Gruber et al., 2009). To simplify matters, the authors could refer to the pre-industrial Southern Ocean.

R13: In accordance to Reviewer #2's remark, we modified the sentence for "the pre-industrial Southern Ocean STP has been unable to fully compensate the CO₂ outgassing and this area represents one of the main oceanic sources of natural CO₂ to the atmosphere (Morrison et al., 2015)" (lines 43-45).

C14: • In line 159 ff. the origin of Ca (marine vs. terrestrial) is discussed. In my respect it would help, if a supplementary figure would be included, showing the Ca/Al or Ca/Ti-ratio over the entire record. As Al and/or Ti are of predominantly terrestrial origin, I think this figure would help to assess whether the marine source of Ca or CaCO₃ changed or not.

R14: We totally agree with Reviewer #2. Indeed, Ca/Ti or Ca/Al ratios are often used to represent biogenic CaCO₃ in sediments (Cartapanis et al., 2011 as an example). Therefore, we plotted the Ca/Ti ratio in the new Supplementary Fig. 4, and discussed within the main body of the manuscript, the excellent linear correlation Ca and Ca/Ti report with CaCO₃ ($r^2=0.75$ and 0.73 respectively) thus definitely excluding any Ca of terrigenous origin (lines 210-214).

C15: • In lines 176-177 the authors exclude changes in TOC as the driver for the transient declines in the Br/Ca-ratio. However, comparing the Br/Ca- (Fig. 2d) and TOC-records (Fig. S5b), I'm afraid that Br/Ca follows TOC from about 14.5 cal BP toward the present. While the authors state that TOC increases throughout the time intervals in question (HS1 and YD), I think that the YD is marked by a slight decrease in TOC, matching the pattern observed in Br/Ca. While I see no problem for HS1, I think the authors should be a bit more conservative, when discussing the YD (e.g. line 183).

R15: That is true. TOC concentration slightly decreases during YD. This is particularly visible in the distribution of TOC data derived from Br_{XRF} (to the difference of low-resolution TOC data obtained by Elementary Analyzer) (Supplementary Fig. 4 in the revised manuscript). We have been more conservative when discussing the YD, mentioning that both increasing CaCO₃ and relatively high but decreasing TOC export productions might be involved in subduing Br/Ca ratio (and thus Biological Pump efficiency) during YD (lines 225-228). Previous line 183 is now represented by lines 245-247: "The SAZ became a net source of atmospheric CO₂ during HS1 and YD (Martinez-Boti et al., 2015), due to enhanced SO upwelling of aged CO₂-enriched deep-waters (Siani et al., 2013) and a weakening of the BCP." Since both processes (physical and biological patterns) are involved in the CO₂ outgassing mentioned during HS1 and YD, this sentence seems still correct.

C16: • When estimating the mean weight of different species (methods), I wonder if the mean weight changed from the glacial to the interglacial, as the presence of CO₂-rich (more corrosive) waters might affect the mean weight from one sample to another.

R16: At site MD07-3088, coccolithophores are mainly represented by the Noelaerhabdaceae family (*E. huxleyi* and *Gephyrocapsa* sp.), *C. leptoporus* and *H. carteri*.

Noelaerhabdaceae family

Actually, as discussed in the manuscript (in the chapter "Increase in subantarctic surface water fertility and [CO_{2aq}] during Southern Ocean upwelling») the mean weight of individual Noelaerhabdaceae coccolith is affected by [CO_{2aq}] of surface waters. Indeed, in geological records that is when general selection for different growth strategies (McClelland et al., 2016) as well as the phenotypic plasticity of coccolithophores naturally

occurred and regulated the carbon acquisition within the cell (Bolton and Stoll, 2013), heavier calcified coccoliths were always associated with increased atmospheric (and subsequently sea surface) carbon dioxide concentrations (Henderiks and Pagani, 2007; Hannisdal et al., 2012; Bolton et al., 2016; McClelland et al., 2016). This is the case at site MD07-3088, when more heavily calcified Noelaerhabdaceae coccoliths are observed at times of intense Southern Ocean upwelling, i.e. when CO₂-rich waters are brought into the photic zone (Fig. 2, Supplementary Fig. 2, and lines 184-188). Therefore, at decennial to centennial scales, opposite trends are observed to what we would expect regarding the potential control corrosive waters might exert on coccolith mass through dissolution processes. Such pattern, as well as the potential impact of dissolution processes are discussed in detail within the methods (chapter “Coccolith taxonomy and preservation” (lines 396-414). At a glacial/interglacial scale, no specific trends are observed in the Noelaerhabdaceae coccolith mass from site MD07-3088 (average masses of 2.44 pg and 2.48 pg are recorded during glacial and interglacial times respectively).

C. leptoporus and H. carteri

At decennial to centennial scales, to the difference of the Noelaerhabdaceae family, SO upwelling dynamic doesn't seem to exert any control on *C. leptoporus* and *H. carteri* coccolith masses (Supplementary Fig. 2). However, at a glacial/interglacial scale, both species seem to be generally represented by identical or slightly heavier specimens during glacial (average masses of 10.57 pg for *C. leptoporus* and 153.69 pg for *H. carteri*), compare to deglacial (average masses of 9.70 pg for *C. leptoporus* and 151.33 pg for *H. carteri*) times (Supplementary Fig. 2). Therefore, during late glacial, i.e. when more corrosive conditions are expected in surface waters, *C. leptoporus* and *H. carteri* do not seem to be affected by potential dissolution processes.

C17 : • The Br-K β and Rb-K α lines in the Avaatech XRF-scans tend to overlap to some extent. Did you check whether or not this might have affected your Br-record? As the Rb-K α line is usually pretty weak I don't think this will be a major issue, but I think it would be good to check it anyway.

R17: Actually, XRF data have been treated by Thomas Richter at the NIOZ laboratory. Unfortunately, since he left this laboratory, we are not able to check if Rb-K α lines overlap Br-K β ones. However, the excellent linear correlation that exists between Br and discrete TOC measurements ($r^2 = 0,87$; see Supplementary Fig. 4) show that changes in Br signal reflect changes in TOC contents, thus pointing that a potential overlap by Rb-K α signal (if existing) is of minor importance.

Some minor typos or technical problems:

C18 : • Line 457: Please cite the references for WOA09 nitrate and temperature. Are the fronts based on Orsi et al. (1995)? If so, please also include this reference.

R18: Thanks to Reviewer #2, we corrected it and now cite Orsi et al. (1995) for the different SO front positions and Boyer et al. (2009) for WOA09 (see Figure 1 caption, lines 707-709). We apologize for this oversight.

C19 : • Lines 469 & 470: I'm aware that Siani et al. (2013) call the species they used *C. wuellerstorffii* in the caption of their Figure 2d. However, I think that's a typo (they call it *wuellerstorfi* in their method section. Unfortunately, I also think that *wuellerstorfi* is still not correct. In my view, you should go with the original 1866 definition of Schwager and call it *C. wuellerstorfi*.

R19 : We agree with the remark of Reviewer #2. In Siani et al., (2013), we used the nomenclature of Egger (1895) for *Truncatulina wuellerstorfi* (*wuellerstorfi*). We have made the required changes in the Figure 2 caption (l. 722).

C20 : • Supplements: I think you mean 152,809 and not 152 809.

R20 : The change has been made line 396 of the new text (Methods)

C21 : • Capture of Figs. S3 & S7: Please replace specie by species.

R21 : This has been done in the Supplementary Figure 2 caption (previously Sup. Fig. 3) (line 15 of the revised text of the Supplementary information). The supplementary Fig. 7 doesn't exist anymore.

C22 : • Reference 34: Is there any manuscript in prep. or under review?

R22 : An article discussing front dynamic has been submitted to Quaternary Science Reviews. Previous reference 34 has been replaced by this reference (Ref. 68) : « Haddam, N. A., et al. Changes in latitudinal sea surface temperature gradients along the Southern Chilean margin since the last glacial, Submitted to Quaternary Science Reviews ».

C23: • Please state in the captions of Figs. 2; S1; S3; S4; S5; S7 what you mean with LGIT. Last Glacial Interglacial Transition, I assume?

R23: Yes. Changes have been made in the new figure captions (Figs. 2-3 and Supplementary Figs. 1-4).

I would think these topics are fairly straightforward, hence my recommendation is minor revisions.

Gruber et al., 2009; Oceanic sources, sinks and transport of atmospheric CO₂; DOI: 10.1029/2008GB003349

Reviewer #3 (Remarks to the Author):

C24: Duchamp-Alphonse and coauthors present new records of coccolith abundance and mass, and new XRF records, from a core on the Chilean Margin with the aim of elucidating temporal changes in the strength of the soft tissue and carbonate pumps, with implications for deglacial pCO₂ rise. *The subject matter is relevant to the paleoclimate community, and the data are novel and interesting.* However, some significant re-writing is needed in order to better showcase key findings and convince readers that the interpretations made are robust. Specifically, arguments need to be better set up so that new findings are presented in a logical order and in the context of the state-of-the-art and all supporting data. The interpretations made will then be easier to follow and more convincing. I have made detailed comments below on specific sections, which I hope the authors will find useful. I also have a number of questions regarding the interpretations, detailed below. Given that more main text figures are allowed in Nature Communications compared to Nature, the authors may consider moving the more important figures from the supplement to the main text when revising the paper.

R24: We agree with the general remarks raised by Reviewer #3 and all the required changes have been made, based on the comments detailed below, to provide what we feel is a much improved manuscript. The principal figures have now been moved to the main body of the revised manuscript, that presents 4 figures (instead of 3 previously). 5 figures (instead of 7 previously) are now presented in the supplementary information. Figure S3 presents news curves, in response to Reviewer #3 comments (C36, C38 and C51).

Specific comments to authors:

C25: A few relevant papers that you may want to consider: Flores et al 2012 (Frontiers in Microbiology), on SAZ coccolithophore productivity in the Pleistocene. Saavedra-Pellitero et al (Paleoceanography, 2017) on potential coccolithophore impacts on the C cycle in the Pacific sector of the Southern Ocean. Meier et al 2014 (Marine Microbial) on coccolith mass changes during Termination II in the context of carbonate chemistry changes.

R25: We thank Reviewer #3 for highlighting these manuscripts. The three papers cited above address important aspects related to coccolithophore productivity in the past that essentially corroborate our arguments. They all have been included in the discussion.

Flores et al., (2012) present coccolithophore productivity changes from the SAZ of the South Atlantic (ODP Site 1089), over the last 500 kyrs, and reveal that coccolith abundances in this area, were closely related to the marine phosphorus inventory, related to the dynamics of the SO upwelling. Therefore, this reference was added line 137 (Ref. 42) when discussing the control macronutrient availability exerts on subantarctic primary productivity in the past.

Saavedra-Pellitero et al. (2017) focuses on coccolithophore productivity pattern from the Pacific sector of the SAZ during MIS 11. Even if this study focuses on a time interval that is disconnected from our work, it represents one of the few papers that discuss changes in coccolith accumulation rates in the South Pacific related to SST and nutrient budget, and their possible contribution to atmospheric CO₂ levels. This reference has been added to the text l. 195 (Ref. 61), when discussing the potential control SST might exert on coccolith production.

Meier et al., 2014 has been added to the cited references when presenting studies that related coccolith mass to cellular calcification in Pleistocene and recent sediments (l. 160, Ref. 50).

C26 : Lines 86-93 belong in the introduction. The role of coccolithophores in the STP and the CCP and the importance of the rain ratio to climate is vital background to your paper, and needs to be described more clearly to set up your study.

R26: The cited paragraph has been moved up to the introduction (lines 91-99 of the revised text) as recommended.

C27: Line 91: changes in their abundance/mass relative to what?

R27 : The term « relative” has been removed from the text to avoid confusion (lines 96-99).

C28 : Spelling: Noëlaerhabdaceae

R28 : Amended

C29: Why is the Chilean Margin an ideal location to answer your scientific questions on deglacial CO₂ rise and the STP/CCP? Most of the intro seems to focus on the Southern Ocean, so you need to connect the two areas better before the results are presented.

R29: This very valid point has also been raised by Reviewer #1. We paid particular attention to clearly outline the characteristics of core MD07-3088 and discuss its specificities location in the introduction (lines 100-109) and the methods (a detailed description of the core has been added lines 266-291, in the chapter “Material and site description”. To summarize, this core gathers many important characteristics to help fulfilling our objectives: the sedimentary archive was retrieved well above the lysocline to minimize the potential biasing effects of selective preservation on export production reconstructions. The sediment is mainly constituted of silt-bearing clays, and presents high sedimentation rates ensuring conditions favorable for the preservation of (in)organic settling particles. This study is the first to our knowledge to document high-resolution deglacial patterns of both STP and CCP outside the Atlantic sector, i.e. far from the main dust sources today, but also in the past (Mahowald et al., 2006). Located under the direct influence of the northward transport of nutrient rich-SSW (Strub et al., 1998) the provides an ideal archive to document the leverage Southern Ocean upwelling and associated changes in surface water chemistry (macronutrient budget and [CO_{2aq}]) might have exerted on deglacial South Pacific coccolithophore productivity at a centennial timescale. Indeed, while iron-dust fertilization (Martin et al., 1990) is widely proposed to explain past changes in SAZ productivity patterns (Martinez-Garcia et al., 2014), such mechanism, might not be the only forcing factor behind past productivity patterns in the SAZ (Flores et al., 2012), and particularly, in the Pacific sector (Chase et al., 2014).

C30: Also, you mention AAIW and PDW, but what surface water masses are affecting the productivity of coccolithophores above your core and how is this thought to have changed since the LGM?

R30: We agree that the previous version of the manuscript lacked a clear description of these water masses. The introduction of the revised text now presents a detailed description of the oceanographic setting (lines 33-40). Because SWW are mainly fed by surface waters from the AZ, as a result of wind-driven (Ekman) upwelling, down-core changes in productivity, will reflect past changes in SSW biogeochemistry (fertility and [CO_{2aq}]), that are ultimately paced by changes in SO upwelling dynamics. This process is now hopefully better exposed within the result and interpretation sections of the revised manuscript (lines 134-141, 145-150 and 184-188).

C31: Is there upwelling of AAIW above your site?

Is it considered to be in the SAZ?

What are the modern disequilibrium wrt CO₂ (Source or sink to the atmos?) and upwelling conditions?

R31: There is no upwelling s.s. of AAIW at the studied site. This is more clearly specified in the introduction (lines 33-40, and 100-109) and described in the methods section (see “Material and site description, lines 266-275). It can be also observed from the Phosphate surface distribution in Fig. 1. The site was located within the SAZ during the entire studied period, and this aspect is now clearly outlined in the text.

C32: Figure 2: For the new records I suggest plotting data points connected with a line, rather than a smoothed line with points in the background. Add running averages as well if you like, but do so for all records in the figure if you choose to.

R32: We connected data points with lines in the revised version of the manuscript to also represent first order variance of data that have been smoothed as recommended (Figs. 2-3 and Supplementary Figs. S2-S4). We smoothed data of the high-resolution records (micropaleontological and XRF data from this study as well as the coccolith abundance data from Saavedra-Pellitero et al., (2011), Fig. 2), to facilitate the comparison between high-resolution records and low-resolution data from literature (Siani et al., 2013; Anderson et al., 2009 and Martinez-Boti et al., 2015).

C33: Different fonts, font sizes, tick mark lengths etc are present in the same figure and some axis numbers/ titles are squashed.

R33: In the revised version of the manuscript, all the figures have been consistently reviewed and now comply with the submission guide for authors.

C34: The yellow bars in figure S2 and S5 cover different intervals from each other and from those in other figures.

R34: The small difference that previously existed in Fig. S5 (now Supplementary Fig. 4) has been rectified in the revised manuscript, so that all yellow bars presented on figures with calendar ages cover the same intervals (Figs. 2-3 and Supplementary Figs 2-4). Yellow bars are shifted in Fig. S2 (now Sup. Fig. 1) in comparison to the other ones, as the set of geochemical records documented here, is expressed in conventional ^{14}C age (not in calibrated ^{14}C age).

C35: One feature of the data that immediately strikes me is the correspondence between higher Noel abundance and higher (average) Noel mass. This is intriguing, because I'd expect higher fertility/ productivity intervals to be associated with a larger number of smaller coccolithophore cells (as is the case in modern blooms).

R35: Higher Noelaerhabdaceae abundance and mass are reported from site MD07-3088 at times when the upwelling of CO_2 -rich subsurface waters increased. Here, increasing Noelaerhabdaceae coccolith mass reflects increasing thickness of individual coccoliths (i.e. PIC/POC_{coccolith} ratio) and likely represents increasing cellular calcification. It is not associated to higher coccolith area, and is thus most probably not associated to increasing cell size (Henderiks, 2008 ; Bolton et al., 2016 ; McClelland et al., 2016) (see Reply #51). Actually, insufficient studies report coccolith mass data from fertile upwelling environments in modern settings but also in the past, to provide robust constraints on the relationship that exists between coccolithophore calcification and trophic levels. Consistent with the observation we report, several studies depict higher coccolith weight and higher primary production, at times of increased macronutrient concentrations (for *E. huxleyi*: Engel et al., 2005 ; Beaufort et al., 2007 ; Noelaerhabdaceae family (Beaufort et al, 2011), but also *Calcidiscus leptoporus* (Henderiks and Renaud, 2004)), thus demonstrating that our results are not a case apart. Further studies of coccolithophores covering a large variety of environmental conditions would be required to better understand the forcing factors behind calcification patterns.

C36: The authors interpret this as reflecting periods of increased upwelling of SAMW, which brings both macronutrients and $[\text{CO}_2]$ to the surface, stimulation both primary productivity and coccolith calcification (increasing cellular PIC:POC ratio I assume, although this is not explicitly stated).

R36: Higher Noelaerhabdaceae abundance and mass are interpreted as reflecting increased macronutrient and $\text{CO}_{2\text{aq}}$ concentrations in SSW, directly linked to increased SO upwelling at the Antarctic Divergence, particularly during HS1 and YD. This is stated more clearly, we hope in the revised version of the manuscript (lines 134-153 and 165-188).

When discussing coccolith calcification in the first version of the manuscript, we were referring to the PIC:POC ratio. This has been more thoroughly developed and discussed in the revised text (lines 159-173, and see the detailed reply to the comment #51)

C37: Has a similar correspondence between inferred primary productivity and coccolith degree of calcification been seen in other cores? At the moment, no comparison with published coccolith mass or abundance data for the last deglaciation is made, although such data exist (e.g. the cores studies in Beaufort et al 2011).

A detailed comparison with published data would increase the impact of your paper.

R37: The only comparative deglacial down-core coccolith record that we are aware of, is represented by the coccolith assemblage record from ODP site 1233, located in the Pacific sector of the Subantarctic Zone (Saavedra-Pellitero et al., 2011) and has now been included within Fig. 2, and discussed within the paper (lines 126, and 148-153). Beaufort et al., (2011), discuss sensitivity of coccolithophore calcification patterns to sea surface $[\text{CO}_{2\text{aq}}]$ over the past 40 kyrs, using similar birefringence-based method than the present study (see Methods (lines 341-351 for more details). Based on 180-surface water samples and 5 coccolith downcore records, these authors encompassed an unprecedented wide spectrum of both present and past oceanic conditions. Only one core located in the SAZ (MD94-103), could have been potentially compared to core MD07-3088. However, only 5 points cover the last deglacial (9 points over the last 20 kyrs), which is insufficient for a comparison with our centennial coccolith record. We note however, that this record, albeit of insufficient temporal resolution, is not inconsistent with our observations.

As proposed by Reviewer #3 in comment #52, we also compare and discuss the ~50% increase of the Noelaerhabdacea mass obtained herein during HS1 and YD, to the ~50% coccolith mass increase previously observed at ODP site 1123 (southernmost Pacific) during TII (McClelland et al., 2016) (l.188-192 of the revised manuscript and see Reply R#52).

This point raises the novelty and uniqueness of our study and highlights the crucial need for further high-resolution micropaleontological studies from the Subantarctic Zone, in order to better document productivity patterns of the Southern Ocean, and to what extent these exerted control on atmospheric pCO₂.

C38: Line 91: by mass, do you mean cellular PIC:POC ratio?

R38: Yes. This point has been better developed and discussed in the revised text (lines 159-173, and see the detailed answer to the comment #51)

C39: Line 94: “coccolith abundance and mass” of the dominant Noel family... in samples from a marine sediment core...

R39: In order to better discuss the principal results of our study, as recommended by Reviewer #3, we moved the most important figures from the supplement to the main text of the revised manuscript. Figure 2 is thus no longer based on Noëlaerhabdacea coccolith abundance and mass only, but also documents the abundance of *C. leptoporus* and *H. carteri*.

C40: Line 103: “Noëlaerhabdacea coccolith abundance and by inference surface ocean fertility” This is a sweeping statement; the authors need to clearly explain what this inference is based on, citing supporting evidence, before making it.

R40: This comment has been taken into account. We now discuss the results in a clearer way, clearly outlining the arguments before making interpretations (lines 132-153).

C41: As pointed out in the introduction, carbonate (and therefore coccolith) accumulation in sediments can be affected by many processes. If you feel you have confidently ruled out dissolution/dilution influences (detailed in the SI?) then say so.

R41: Diagenetic influences on carbonate and coccolith are now thoroughly discussed in the text (lines 131-132) and the methods section (“Coccolith taxonomy and preservation” l. 396-414). Dilution effects on coccolith abundances (and therefore CaCO₃ concentrations) are detailed in Reply R44).

C42: Line 116: be more specific about what you mean by “large-scale ocean circulation changes”.

R42: Here, we make reference to the impact of SO upwelling on the macronutrient budget in the SSW, and by inference on the Noëlaerhabdacea coccolith abundance at site MD07-3088. We have revised the text accordingly (lines 134-153). To avoid any confusion with larger-scale ocean circulation changes, and to better suit the data, we have changed “large-scale ocean circulation changes” to “regional ocean circulation changes” (l. 147-148).

C43: Line 117: “the downcore primary production record mimics opal fluxes in the AZ at sites influenced by similar processes. This suggests that the inferred changes in productivity are neither limited to a specific phytoplankton group nor a specific area, but highlights a large-scale sensitivity of phytoplankton growth to ocean circulation and nutrient supply from below”

Be careful about calling your Noel abundance record a “primary production record”; at best, it represents coccolithophore export productivity.

R43: Fair point. « Primary production record » has been changed to « coccolithophore export productivity record» (l. 148).

C44: Do sedimentation rates also increase at times of inferred higher productivity? Do your coccolith abundance records have the same structure when expressed as MARs?

R44: The sedimentation rates inferred for the core MD07-3088 range from 438 cm/kyr during the Last ice age (around 19 kyrs) to 31 cm/kyr (around 12 kyrs), with averages of ~ 300 cm/kyrs during the LGM and ~ 60 cm/kyr during the deglaciation and the Holocene (Siani et al., 2013) (figure R1 below). Therefore, they do not significantly change over the last ~18 kyrs.

Figure R1: Age vs Depth of the core MD07-3088.

Figure R2 : Evolution of the sedimentation rates of core MD07-3088 over the last deglaciation as well as the number (/g of sediment) and flux (cm^2/Kyr) of the main coccoliths.

Besides, the main increasing/decreasing trends depicted by coccolith MARs, are highlighted by coccolith abundances (number/g of sediments) (Figure R2 above). Particularly, the three deglacial peaks in the abundance of *Noëlaerhabdacea* coccolith (/g of sediment) are accompanied by concomitant peaks in *Noëlaerhabdacea* coccolith MARs. Only during the LGM and to a lesser degree during the ACR, do significant increases in sedimentation rates (up to 6 times higher from 18.2 to 20 kyrs and 2 times higher around 14.5 kyrs) coincide with significant increases in coccolith fluxes relative to coccolith abundances (Figure R2). However, during the LGM, sedimentation rates increased at times of inferred higher productivity, thus corroborating our interpretations. During the ACR they increase at times of rather variable coccolith abundances (this is particularly the case for *Noëlaerhabdacea* coccoliths (Fig. 2)). As such, the increase in coccolith flux (and *Noëlaerhabdacea* coccolith one) obtained at a centennial scale during this time interval is likely robust, but might also be driven changing sediment accumulation, which ultimately depend on our radiocarbon data points obtained at a millennial scale. Therefore, assuming that coccolith abundance trends were likely not substantially affected by dilution processes (Figs. R1 and R2), and that coccolith fluxes may be biased by low-resolution sedimentation rate reconstructions around 14.5 kyrs, we deem it appropriate to present and discuss coccolith abundances rather than coccolith fluxes. Besides, MAR derived are not always accurate. Application of the ^{230}Th normalization method would indeed be preferable to estimate vertical particle (coccolith) fluxes (e.g. Francois et al., 2004).

However, the application of the method is not straightforward in marginal settings as ^{230}Th is preferentially scavenged in high particle flux environments (boundary scavenging) thereby introducing a substantial bias in the vertical flux determinations (Francois et al., 2004; Chase et al., 2014).

Furthermore, since our data are consistent with micropalaeontological and geochemical data covering large swaths of the Antarctic, Subantarctic and Subtropical zones, we remain confident that our records are primarily controlled by large-scale processes, rather than by local patterns.

C45: To say that the entire region and all phytoplankton groups are behaving the same in terms of primary productivity because you see covariation between your Noel abundance record from the Chilean Margin and an opal record from a site in the Atlantic sector of the PFZ is a gross oversimplification. It is true that coccolithophores and diatoms often increase together in upwelling regions and also in Fe fertilization experiments, and much has been published on this. However, the SAZ/AZ is an extremely complex region oceanographically, and there is much heterogeneity in terms of productivity in these (shifting) frontal regions (for example, see Hillenbrand and Cortese 2006 for a longer-term perspective).

R45: We have now been more prudent when comparing the pattern of *Noëlaerhabdacea* abundance at site MD07-3088 to the opal fluxes obtained within the AZ (Anderson et al., 2009). Actually, when discussing millennial scale variations, it seems that during the last ice age at times atmospheric pCO_2 were relatively low, Antarctic export production declined, while subantarctic phytoplankton community thrived, thus highlighting different modes of productivity changes (Kohfeld et al., 2005; Martinez-Garcia et al., 2009; Jaccard et al., 2013). Such patterns have been interpreted as the result of a northward shift in Southern Ocean Fronts (Mortlock et al., 1991), associated with the glacial expansion of sea-ice (Hillenbrand and Cortese, 2006). More recently however, greater supply of Fe-bearing dust to the glacial Subantarctic Zone has also been evoked to explain increased productivity pattern on glacial/interglacial timescales thus questioning the impact of frontal zonation shifts (Martinez-Garcia et al., 2011, 2014) (see "Material and site description within the Method section, lines 275-278 for core MD07-3088).

C46: Many records of opal (and other export productivity proxies) exist for your time interval and region and if you'd like to extrapolate your results to the whole region a more thorough analysis of published records is needed, in my opinion.

R46: As explained above, we have been more careful when discussing the role SO upwelling might have exerted on productivity the Southern Ocean. Accordingly, an exhaustive paragraph dedicated to the main deglacial export production data in the Subantarctic Zone has now been added in the introduction, in order to more accurately set up the "state of the art" context of our study (lines 66-70). We note however that export production data remain relatively sparse and largely concentrate on South Atlantic sedimentary archives (Kumar et al., 1995; Kohfeld et al., 2005; Martinez-Garcia et al., 2009; Lamy et al., 2014; Martinez-Garcia et al., 2014; Gottschalk et al., 2016 to mention a few studies), located downwind of Patagonia. As stated in the revised version of the manuscript (l. 73-75), "these records are not necessarily representative of the entire Southern Ocean", and "the comparison with South Pacific records in particular is not straightforward because of differing modes of iron supply (Boyd et al., 2007) and contrasting factors modulating productivity patterns". Indeed, only a handful deglacial productivity records exist within the Pacific sector of the Subantarctic Zone (Saavedra-Pellitero et al., 2011) outside of the present-day coastal upwelling off central Chile, located in the northernmost part of the Subantarctic Zone (Muratli et al., 2010; Lamy et al., 2014; Chase et al., 2014). More importantly, most of the studies present millennial-scale records, which are unfortunately not comparable to our decennial to centennial scale data. Therefore, apart from the Antarctic opal fluxes discussed previously (Anderson et al., 2009, Figure 2), the only comparative down-core productivity data that exist to the best of our knowledge are represented by the high-resolution coccolith assemblage record from ODP site 1233, located in the Pacific sector of the Subantarctic Zone (Saavedra-Pellitero

et al., 2011). As such, these coccolith data have now been included within Fig. 2, and discussed within the paper (lines 124-130 and 148-153). Millennial scale opal and centennial scale C_{org} data exist at ODP site 1233 (Muratli et al., 2010). However, as reported by the authors, changes in opal contents seem associated to the occurrence of ash layers. Changes in C_{org} , although interpreted by the authors as the consequence of dilution processes by terrigenous inputs, seem to depict similar long-term trends as our C_{org} record.

C47: Line 107: My understanding is that SAMW forms in the SAZ, subducts into the pycnocline and flows northwards at depth, providing nutrients to the low-latitudes (<30°) via advective mixing and upwelling. Ref 18 shows modelling evidence for SAMW fueling productivity between 30S and 30N – so I am confused as to whether you are saying that upwelling of SAMW occurs at your site (and is the dominant source of macronutrients) or not. Please clarify.

R47: Core MD07-3088 is located much further South than 30°S and no upwelling occurs today at the site as can be seen from the surface phosphate concentration map (Fig. 1). As stated above, we now pay particular attention in better describing the regional oceanographic setting, including the SWW, the AAIW and the SAMW (lines 33-40). Because the SWW are mainly fed by surface waters originating from the AZ, themselves resulting from deep waters upwelling at the Antarctic Divergence as a result of wind-driven (Ekman) upwelling, changes in productivity down-core MD07-3088 primarily reflect past changes in SSW chemistry (macronutrient and CO_{2aq} concentrations), that are paced by changes in SO upwelling dynamics. This process is better exposed in the revised manuscript (lines 134-148 and 159-188). Nutrients carried by AAIW and AMW may mix with surface layers throughout the SH oceans, however, they are not the main source of macronutrients within SSW.

Reviewer # 3 is certainly correct that when referencing to Palter et al. (2010) (ref 18 of the first manuscript), we oversimplified the mechanisms controlling sea-surface fertilization events at Site MD07-3088. We now make reference to Strub et al., (1998) and DiFiore et al., (2006) in the new text (Refs. 2 and 40, l. 137).

C48: Line 115: to use this as an argument, you need to remind the reader what the $\Delta\Delta^{14}C$ and $\Delta\delta^{13}C$ records mean.

R48: The sentence « that reflect increased rates of vertical mixing,» has been added to the new text, lines 146.

C49: Lines 114-125: I am not able to follow your arguments here. First you say that coccolithophore and diatom productivity appear to be going hand in hand throughout the region, and that all groups seem to be stimulated by upwelled nutrients, then you go on to say that your data support existing records showing a shift from diatom to coccolithophore productivity due to silica leakage. Very confusing.

R49: We agree with Reviewer #3. The last part of this chapter does not add much to the paper. We have deleted lines 120-125 of the previous version of the manuscript.

C50: Again (similar to the statement at line 103), the statement “mean coccolith mass, a proxy for sea-surface [CO_{2aq}]” is controversial, and not supported by arguments or references in the text where it appears. You need to be more careful with your wording, and order your arguments appropriately (i.e. We observe X and Y. Because of this and this, supported by these other things/existing studies, and because we can exclude a and b as potential drivers of coccolith mass in our core, we hypothesize that coccolith mass reflects degree of calcification and is responding to changes in [CO_{2aq}] driven by changes in upwelling intensity).

R50: Fair point. This remark has been taken into account, and this paragraph has now been rewritten. A thoroughly revised paragraph has been added at the beginning of the argumentation (l. 159-173). It particularly discusses changes in newly calculated size-normalized coccolith thickness indices (the Size Normalized index “SN” according to Bolton et al., 2016 and the coccolith Aspect Ratio “AR_L” according to McClelland et al., 2016, see the Methods section (lines 356-379) and Reply R51 below) demonstrating that changes in Noëlaerhabdacea mass reflect variations in the thickness of an individual coccolith and therefore represent the degree of cellular calcification (i.e. PIC/POC_{coccolith}) instead of coccolith size (or area) that could highlight changes in coccolithophore cell size. This chapter comes just before the chapter discussing the relationships that might exist between coccolithophore calcification rates and [CO_{2aq}] and the actual interpretation of our results.

C51: I would like to see the mean coccolith size data (for the same size classes as mass) to be convinced that changes in coccolith mass are independent of size, as stated in the text, and are “mainly an effect of increasing coccolithophore calcification” (does this mean increasing inferred cellular PIC:POC ratios?). Also, in my mind “increasing coccolithophore calcification” with no size changes equates to “more heavily calcifying morphotypes”, unless you are saying there is an increase in coccolith production rate only? In which

case the mean mass should stay the same when coccoliths/g increases. I don't understand how you can exclude the possibility that the dominant morphotype is changing.

R51: Please find the mean *Noëlaerhabdacea* coccolith mass and area (reflecting coccolith size) data, as well as the mean AR_L and SN indices newly calculated for the studied samples and presented over the last 20-3 kyrs, in Fig. 2 and Supplementary Fig. 3, below. As reported in the answer to question 50 above, and as explained lines 159-173 in the new manuscript, changes in coccolith mass reflect changes in coccolithophore calcification rates, i.e. PIC/POC ratio. This is supported by the positive correlation that exists between coccolith mass and these recently developed Size Normalized thickness indices (SN and AR_L see Supplementary Fig. 3). Therefore, increasing coccolithophore calcification rates highlight more heavily calcifying morphotypes, and not only an increase in coccolith production. In such a scenario, we cannot exclude any changes in the dominant morphotype. The sentence "The coccolith pattern cannot be explained by floral assemblage shift toward larger and heavier calcifying morphotypes, but mainly an effect of increasing coccolithophore calcification, as its affects all narrowly restricted size classes" that could be confusing has been removed from the new manuscript.

Age (yrs)	Mass (pg)	Area (cm ²)	ARL	SN (cm)	Age (yrs)	Mass (pg)	Area (cm ²)	ARL	SN (cm)
3220	2.13	8,04276	0,08998	0,25121	14478	2,67	9,08333	0,09918	0,28573
3612	2,61	8,08177	0,11829	0,31902	15127	2,44	9,27394	0,08621	0,25019
4012	2,44	8,36824	0,10054	0,28003	15698	3,01	8,55073	0,12334	0,34547
4483	2,23	8,03318	0,09788	0,26954	15902	2,62	8,72723	0,10035	0,28931
4954	2,08	8,30199	0,08528	0,23897	16213	2,84	8,57003	0,11311	0,32087
5425	2,73	7,81914	0,13136	0,34630	16496	2,71	8,83213	0,10446	0,29715
5891	2,33	8,14765	0,10355	0,27979	16661	2,49	8,94116	0,09246	0,26685
6256	2,53	8,35027	0,10348	0,28864	16816	2,48	9,04973	0,09214	0,26442
6621	2,53	9,32822	0,08548	0,24565	16976	2,59	8,67997	0,10086	0,28495
6986	2,54	8,42028	0,10615	0,29240	17159	2,42	8,77070	0,09367	0,26572
7351	2,20	8,17407	0,09416	0,26083	17341	2,09	8,38180	0,08303	0,23693
7534	2,68	8,02481	0,12128	0,32876	17685	2,24	8,48312	0,08559	0,24322
7716	2,28	8,25099	0,09919	0,27068	17762	2,10	8,73825	0,08473	0,23913
7899	2,83	9,30379	0,09519	0,27495	17840	2,29	8,20599	0,09668	0,26709
8081	2,60	8,38501	0,10965	0,30330	17917	2,48	8,13210	0,10765	0,29640
8264	2,29	8,31430	0,09686	0,26861	17995	2,74	8,36194	0,11395	0,31653
8448	2,71	8,10598	0,12137	0,32854	18124	2,49	8,69061	0,09581	0,27232
8640	2,50	8,27462	0,10749	0,29457	18154	2,39	9,50311	0,07544	0,22179
8832	2,79	8,04778	0,12718	0,34203	18179	2,23	8,50413	0,09012	0,25216
9213	2,66	8,42480	0,11227	0,30889	18206	2,41	8,64799	0,09500	0,26806
9389	2,65	8,26273	0,11645	0,31685	18244	2,58	9,84634	0,08008	0,23666
9564	2,30	8,43978	0,09368	0,26261	18281	2,75	8,74355	0,10950	0,30582
9739	2,83	8,72822	0,11316	0,31755	18318	2,30	8,39202	0,09357	0,26164
9915	2,50	8,83545	0,09692	0,27494	18392	3,25	8,53252	0,13003	0,36565
10090	2,71	8,65554	0,10850	0,30567	18429	2,48	8,53188	0,09937	0,27925
10610	2,75	8,45465	0,11592	0,32102	18467	2,48	8,53552	0,09901	0,27722
10780	3,02	8,76916	0,12812	0,35348	18541	2,33	8,66700	0,09050	0,25545
10950	2,81	8,71346	0,11223	0,31583	18578	2,03	8,31745	0,08392	0,23734
11120	2,66	8,76916	0,10619	0,29848	18615	2,29	8,62503	0,08957	0,25241
11290	2,51	8,48098	0,10459	0,29088	18653	2,64	8,66419	0,10481	0,29441
11512	2,72	8,48312	0,11302	0,31389	18689	2,22	9,41867	0,06952	0,20588
11943	2,31	9,52615	0,07677	0,22335	18783	2,25	8,58022	0,08744	0,24726
12159	2,39	8,89495	0,08965	0,25599	18908	2,28	8,61062	0,09071	0,25456
12767	2,16	8,44061	0,08617	0,24552	19034	2,58	8,40052	0,10478	0,29389
13248	2,24	8,76155	0,08536	0,24313	19159	2,49	8,60689	0,09976	0,27728
13488	2,36	8,66209	0,09212	0,26134	19285	2,26	8,34719	0,09203	0,25811
13817	2,17	8,56870	0,08561	0,24304	19410	2,21	8,39704	0,08652	0,24564
13932	2,44	10,00354	0,07500	0,21793	19598	2,38	8,35035	0,09638	0,26853
14046	2,46	8,38310	0,10145	0,28430	19786	2,26	9,50572	0,06999	0,20742

Table 1: *Noëlaerhabdacea* morphometric data

C52: Are the [CO_{2aq}] estimates suggested for the last deglaciation in your core anywhere near the range used in experiments (don't forget about the temp effect on [CO_{2aq}])? i.e. would you expect strong sensitivity? Same for the comparisons with fossil data (refs 29-32) – which of these represent a similar range of [CO_{2aq}] variability, and how does sensitivity of mass to [CO_{2aq}] compare?

R52: Unfortunately, we cannot provide [CO_{2aq}] estimates at site MD07-3088. However, it seems important to compare our coccolithophore calcification pattern to the estimated [CO_{2aq}] variations. Therefore, we compare our micropaleontological data with the available variations of [CO_{2aq}] for the Atlantic surface waters of the SAZ (core PS2498, Martinez-Boti et al. 2015). Indeed, surface waters at site PS2498 were likely also influenced by waters that were advected northwards via Ekman pumping. The Southern Ocean is zonal and these data represent the only one comparative down-core [CO_{2aq}] dataset that exist so far for the SAZ during the deglaciation. [CO_{2aq}] variations in the SAZ are up to 3 μmol/kg and 1.9 μmol/kg, during HS1 and YD respectively, i.e. when *Noëlaerhabdacea* coccolith mass, as well as "SN" values increase by about 50% (from ~2 to ~3 pg, and from ~0.2 to ~0.3 respectively) at site MD07-3088 (Supplementary Fig. 3, using a common age model for core PS2498-1 (Martinez-Boti et al., 2015) and core MD07-3088 (Supplementary Fig. 1).

In refs 29-32, Henderiks and Pagani (2007) and Hannisdal et al., (2012) present *Noëlaerhabdacea* morphological datasets (reticulofenestrads) that are not comparable with our record since both studies present changes in

coccolith sizes (length) (associated with changes in atmospheric $p\text{CO}_2$ or $[\text{CO}_{2\text{aq}}]$) that represent changes in cell size, and not in the degree of cell calcification, as reported by our changes in coccolith thickness. Besides, these Eocene-Miocene ancestors, as reported by Henderiks and Pagani (2007), appear to have ~50% larger cell diameters than modern Noëlaerhabdacea (*E. huxleyi*), producing thicker calcified coccoliths ($> 1 \mu\text{m}$).

Interestingly, our results are in the same order of magnitude than those observed at ODP site 1123 (southernmost Pacific) during TII (McClelland et al., 2016), and at site NGHP-01-01A in the northern part of the Indian Ocean, around 10 Myrs (10.3 – 9.3 Myrs) (Bolton et al., 2016). Indeed, ODP site 1123 and site NGHP-01-01A depict 50% increases in Noëlaerhabdacea coccolith mass (~2 to ~3 pg) and SN (~0.4 to ~0.6) when $[\text{CO}_{2\text{aq}}]$ rise of about 3 $\mu\text{mol/L}$ and 2 $\mu\text{mol/L}$ respectively. Since site NGHP-01-01A shows long-term *Reticulofenestra* and *Gephyrocapsa* coccolith data over the last 16 Myrs, i.e. different species from our study, but also coccolithophores that secrete thicker coccoliths (as exposed above), it seems correct to only compare our data with those of McClelland et al., (2016) (lines 188-192 in the new text).

C53: Please explain how increased upwelling intensity (of SAMW?) during HS1 and the YD leads to increased SST, when usually enhanced upwelling is associated with decreased SST.

R53: We made this argument more explicit in the revised text, and now refer to increased SO upwelling at the Antarctic Divergence (as opposed to local upwelling). These increased SO upwelling episodes occur during Southern Hemisphere deglacial warm phases and are possibly linked to a southward shift of the prevailing westerly winds (Toggweiler et al. 2006) and/or sea-ice retreat (Keeling and Stephens 2000, Ferrari et al. 2014).

C54: Has the ratio of Br/Ca ever been applied before to estimate the strength of the STP/CCP, or is your study the first application of this proxy?

R54: Actually, it is the first time to our knowledge that Br/Ca ratio is used to estimate the strength of the STP/CCP (i.e. the Biological Pump) in the past. This has now been mentioned within the introduction, lines 112-114.

C55: How is it better than using TOC/CaCO₃ to characterize the export Corg to CaCO₃ ratio (for ex. is the Br signal preserved when OC is not?)?

R55 : In sedimentary archives, Br is associated with marine organic matter (Mayer et al., 2007) and is affected by diagenetic processes the same way TOC is. As explained in detail in Reply #R4, the use of Br/Ca to reflect downcore changes in the TOC/CaCO₃ ratios is mainly justified based on technical aspects. Indeed, Br/Ca ratio data have been obtained by X-Ray Fluorescence on ~ 1800 samples in less than 10 days, allowing to document past changes in the Biological Pump at a decadal timescale. Such resolution cannot realistically be achieved for discrete CaCO₃ and TOC measurements using the vacuum-gasometric technique and the Elementary Analyzer used herein (see methods). Besides, as demonstrated in the article (lines 201-221), Br/Ca and TOC/CaCO₃ both robustly reflect changes in the sedimentary content of TOC relative to CaCO₃.

C56 : I don't fully understand how you derive STP strength (is this the same as "biological pump efficiency" mentioned on Fig. 2?), so some clarification is needed here.

R56: A similar point has also been raised by Reviewer # 1, who pointed out a certain confusion in the terminology used in the initial version of our manuscript. As detailed in our comment Reply R8, we use now clearer definitions of the STP, CCP and BCP. In the revised manuscript, the biological carbon pump (BCP) refers to the sequestration of atmospheric CO₂ within subsurface waters. It is composed of two components: the soft-tissue pump (STP) which is a sink of CO₂ through the settling of organic matter (and CO₂), and the carbonate counter pump (CCP) which is a source of atmospheric CO₂ through the settling of CaCO₃ (and alkalinity). The net sequestration of atmospheric CO₂ by the BCP is therefore a balance between the STP and CCP components (see changes lines 23, 28, 30, 77, 97, 119, 221, 241, 244, 247, 251, and 257).

C57: Again, the set-up of the arguments needs work. For example, starting a sentence with "this pattern could potentially be explained by lower TOC preservation" immediately puts doubt in the reader's mind, even though you then go on to explain why this is unlikely.

R57 : This sentence has been removed from the revised manuscript.

C58: For the last ~6 kyr, Br/Ca is high, but coccolith abundance is low. What is your interpretation of this Holocene interval?

R58: The increasing trends of Br/Ca over the last 3-6 kyrs mirror the increasing content of TOC in the sediments (Fig. S4) as it has also been observed further north at sites 1233 (41°S, 74°W) and 1234 (36°S, 73°W) (Muratli et al., 2009). Since no specific increase in coccolith size (and probably coccolithophore cell size) is observed at that time (Supplementary Fig. 3), this trend is most likely associated to higher abundances of other non-calcareous phytoplanktonic organisms (diatoms or picoplankton).

C59: Is 1/rho standard notation for POC/PIC ratio? I thought that rho is usually used for density.

R59 : Yes. « ρ » is used to document PIC/POC ratio (see McClelland et al. (2016) as an example).

C60: Figure 3, as it is currently discussed in the text, doesn't really add much to the paper. Most of the important info pertaining to this figure seems to be in the methods and caption.

R60: We agree with the reviewer and the figure (now Fig.4) is presented in a clearer way. It is unfortunately yet not possible to quantitatively assess the influence of CCP on atmospheric pCO₂. However, we indicate and now, better discuss, possible changes of the POC:PIC ratio (lines 238-256).

References

- Abelmann et al., *Paleoceanography* 21, PA1013, doi:10.1029/2005PA001199 (2006).
- Anderson, R., et al., *Science* 323, 1443-1448 (2009).
- Beaufort et al., *G3* 8 (8), doi:10.1029/2006GC001493 (2007).
- Beaufort, L., et al., *Nature* 476, 80-83, doi:10.1038/nature10295 (2011).
- Bolton, C.T. and Stoll, H.M. *Nature* 500, 558-562, doi:10.1038/nature12448 (2013).
- Bolton, C.T., et al., *Nature Communications* DOI: 10.1038/ncomms10284 (2016).
- Boyd, P.W. et al., *Science* 315, 612-617 (2007).
- Cartapanis O., et al., *Palaeoceanography* 26, PA4208, doi:10.1029/2011PA002126 (2011).
- Chase, Z., et al., *Quaternary Science Reviews* 99, 135-145 (2014).
- DiFiore, P.J., et al., *Journal of Geophysical Research* 11, C08016, doi:10.1029/2005JC003216 (2006).
- Engel, A., et al., *Lino. Oceanogr.* 50 (2), 493-507 (2005).
- Francois, R., et al., *Paleoceanography* 19, PA1018, doi:10.1029/2003PA000939 (2004).
- Flores, J.-A., et al., *Marine Micropaleontology* 40, 377-402 (2000).
- Flores, J.-A. et al., *Palaeo3* 196, 409-426 (2003).
- Flores, J.-A., et al., *Frontiers in Microbiology* 3, doi:10.3389/fmicb.2012.00233 (2012).
- Gottschalk, J., et al., *Nature Communications*, doi:10.1038/ncomms11539 (2016).
- Haddam, N., PhD manuscript, University Paris Saclay, pp. 262 (2016)
- Haddam, N. A., et al., Submitted to *Quaternary Science Reviews*.
- Hannisdal, B., et al., *Global Change Biol.* 18, 3504–3516 (2012). □
- Hebbeln, D., et al., *Marine Geology* 164, 119-137 (2000).
- Henderiks, J., *Marine Micropaleontology* 67, 143-154 (2008).
- Henderiks, J., and Renaud, S., *J. Nannoplankton Res.* 26 (1), 1-12 (2004).
- Henderiks, J. & Pagani, M. *Paleoceanography* 22, PA3202, doi:10.1029/2006PA001399 (2007).
- Hillenbrand, C.D., and Cortese, G., *Palaeo3* 242, 240-252 (2006).
- Hodell, D.A. et al., *Global and Planetary Change* 24, 7-26 (2000).
- Howard, W.R., Prell, W.L., *Paleoceanography* 9 (3), 453-482 (1994).
- Jaccard, S., et al., *Science* 339, 1419-1422 (2013).
- Jacot des Combes, H., et al., *Paleoceanography* 23, PA4209, doi:10.1029/2008PA001589 (2008).
- Kohfeld, K.E., et al., *Science* 308, 74-78 (2005).
- Kumar, N., et al., *Nature* 378, 675 – 680 (1995).
- Lamy et al., *Science* 343, 403-407, doi: 10.1126/science.1245424 (2014).
- Martin, J.-H., et al., *Nature* 345, 156-158 (1990).
- Martinez –Boti, M.A., et al., *Nature* 518, 219-222 (2015).
- Martinez-Garcia, A., et al., *Paleoceanography* 24, PA1207, doi : 10.1029/2008PA001657 (2009).
- Martinez-Garcia, A., et al., *Nature* 000, doi: 10.1038/nature10310 (2011).
- Martinez Garcia, A., et al., *Science* 343, 1347-1350 (2014).
- Mayer, L.M., et al., *Marine Chemistry* 107, 244-254 (2007).
- McClelland, H.L.O., et al., *Scientific Reports*, 6:34263- DOI: 10.1038/srep34263 (2016).
54. Meier, K., et al., *Marine Micropaleontology* 112, 1-12 (2014).
- Mahowald, N.M., et al., *Journal of Geophysical Research* 11, doi:10.1029/2005JD006653 (2006).
- Martinez Garcia, A., et al., *Science* 343, 1347-1350 (2014).
- Milliman, J.D. et al., *Deep-Sea Research I* 46, 1653-1669 (1999).
- Morrison, A., Frölicher, T., Sarmiento, J., *Upwelling in the Southern Ocean*, *Physics today*, 27-32 (2015).
- Mortlock et al., *Nature* 351, 220-223 (1991).
- Muratli et al., *Nature Geoscience* 3, doi:10.1038/NGEO715 (2009).
- Muratli, J.M., et al., *Nature Geoscience* 3, 23-26, doi: 10.1038/NGEO715 (2010).

Orsi, et al., *Deep-Sea Res.* 42 (5), 641–673 (1995).
Palter, J. B., et al., *Biogeosciences*, 7, 3549–3568 (2010).
Pollard, R.T. et al., *Nature* 457 n(7229), 577-581 (2009).
Rembauville, M., et al., *Deep-Sea Research I* 115, 22-35 (2016).
Saavedra-Pellitero, M., et al., *Paleoceanography* 26, PA1201, doi:10.1029/2009PA001824 (2011).
Saavedra-Pellitero, M., et al., *Quaternary Science Reviews* 158, 1-14 (2017).
Salter, I., et al., *Limnology and Oceanography* 55 (5), 2207-2218 (2010).
Salter, I., et al., *Global Biogeochemical Cycles*, doi: 10.1029/2010GB003977 (2012)
Salter, I. et al., *Nature Geoscience*, DOI: 10.1038/NGEO2285 (2014).
Siani, G., et al., *Nature Communication* 4, 2758. DOI: 10.1038/ncomms3758 (2013).
Smetacek, V., et al., *Antarctic Science* 16 (4), 541-558 (2004).
Smetacek, V., et al., *Nature* 487, 313-319 (2012).
Stephens, B.B., Keeling, R.F., *Nature* 404, 171-174, doi:10.1038/35004556 (2000).
Strub, P. et al., *The Global Coastal Ocean, Regional Studies and Syntheses*. 273–315 (Wiley, 1998).
Thomas, A.C. et al., *Journal of Geophysical Research* 99, 7355-7370 (1994).
Toggweiler, J.R., et al., *Paleoceanography* 21 PA2005doi:10.1029/2005PA001154 (2006).

Reviewers' comments:

Reviewer #1 (Remarks to the Author):

The authors have provided a detailed and carefully weighted response to all of the comments raised during the review process. I just have a few minor points I would like to see addressed prior to publication:

C1: I still have some concerns about the location of the core, regarding its proximity to the continental margin. In their response the authors mention the site is characterised by a zone of high precipitation that results in high fluvial sediment supplies to the South Pacific Ocean. This results in a high sedimentation rate that facilitates decadal resolution during the last glacial termination. I would therefore still have some concerns about delineating vertical sedimentation from the upper-ocean and lateral fluvial input from the continent.

The authors argue in their reply and revised manuscript that our current understanding outside of the Atlantic sector is limited "little is known about deglacial export production patterns and STP strength in the Subantarctic Zone (SAZ) outside of the Atlantic sector." (Lines 63-64). This may indeed be true, but it was my understanding that the reason for focusing on the Atlantic sector is that this is the area where the iron-rich dust plumes from Patagonia are likely to have exerted maximal influence on productivity changes by alleviating iron limitation. Indeed the authors offer this rationale themselves: "However, these records remain sparse and largely concentrate on South Atlantic sedimentary cores located downwind of Patagonia, the most prominent dust source region to the Southern Ocean" (Lines 70-72).

The authors provide much more information in the revised version regarding the location of the core and clearly make distinctions about its characteristics and separating it from the Atlantic sector of the SAZ. I believe they now provide adequate information to the reader concerning these details about core location and the prevailing conditions, and make a good argument for considering productivity patterns at different locations.

C2: Concerning the response of the authors to my original comment C6. I am aware that it is common in paleoceanographic studies to use sedimentary TOC values as a proxy for export productivity. I agree with their argument that as a first order principle one could expect higher productivity to result in higher sedimentary TOC values. From a qualitative point of view one therefore might be able to argue that these relationships are larger than those caused by selective degradation processes. The challenge here is that the paper attempts to use TOC as a quantitative estimate of POC export below the ventilation depth in comparison to calcite export to estimate the effect on the CCP. The authors now assume minimal calcite dissolution and TOC sedimentary values are 20-30% of ventilation depth export. Defining remineralization length scales of carbon in the BCP is an active field of research in contemporary oceanography. The authors claim that remineralization increases as a function of depth. This is correct, but it does not proceed in a linear fashion that would allow one to argue 1500m is an intermediate depth and thus reflects surface productivity. POC remineralization length scales typically follow a power-law relationship, reaching an asymptote in the bathypelagic ocean where typically <10% of surface-export reaches. These power-law curves are parameterized by an exponent b , which describes most of the attenuation in the surface ocean and has been shown to strongly depend on ecosystem structure. It therefore remains challenging to argue that changes in sedimentary TOC primarily reflect the strength of the biological carbon pump (i.e. POC flux out of the mixed layer) and not transfer efficiency (difference between POC flux at the mixed layer and into bathypelagic ocean (>1000m), the latter strongly related to ecosystem structure of export (plankton functional groups, aggregates, fecal pellets, etc). This is relevant if one attempts to use sedimentary TOC and calcite values to reconstruct CO₂ exchange in the surface ocean. The authors should also not confuse good preservation of material arriving at the seafloor with how representative what arrives at the seafloor is with respect to

surface export patterns, which is of course of most relevance concerning the BCP. I accept the arguments that minimal test dissolution and few fragments is a reasonable basis to assume the calcite fluxes are representative of the ventilation depth. However, the same argument cannot be made for TOC, and since one must compare TOC and calcite to deduce the strength of the CCP, this needs to be made very clear. I realize it is challenging for the authors to achieve much more with their present data, but that it is not common for paleoceanography studies to account for these mechanistic processes is a poor argument for understating their significance. I would encourage the authors to be more explicit in the manuscript concerning the limitations of linking TOC to surface POC export and the impact that has on the quantitative reconstructions.

C3: It seems to me from Figure 3 that foraminifer calcite is quantitatively more important than calcite biomass by a factor of around 3 (compare Figures 3b and 3c). It is therefore important to understand how much calcifying benthic foraminifer might contribute to this. The authors have responded by stating that they weighed a few discrete fractions of benthic foraminifer and it ranges from 0.3 to 1 mg/g. It would be useful to add here how this compares with total planktonic calcite biomass (forams + coccoliths). It seems from rapidly consulting Figure 3b that this is similar to the total calcite coccolith biomass (0.4 – 2 mg/g).

C4: Reviewer 2 comment C16

The reviewer raises a point here about the effect of differing atmospheric CO₂ on coccolith mass. The authors have engaged in a sufficient and detailed rebuttal. However, I wonder about foraminifera, which appear to be the dominant component of calcite mass (according to Figures 3b and 3c). It has been shown previously that *G. bulloides* test weights are strongly impacted by CO₂ water chemistry over glacial cycles (Moy et al. 2009, Nature Geoscience doi :10.1038/ngeo460). Does that play a role here?

Reviewer #2 (Remarks to the Author):

Duchamp-Alphonse et al. submitted a sufficiently revised manuscript. Hence, I suggest, to accept it for publication.

However, the only thing that has to be changed before, is (still) the nomenclature of the benthic forams used. In lines 722 and 737 you (incorrectly) refer to *C. wuellerstorfi*. The original description was done by Schwager (1866) as *Anomalina wuellerstorfi*. You might argue to call it *Cibicoides*, *Cibicides* or *Fontbotia*, however *wuellerstorfi* stays *wuellerstorfi*.

Schwager, C. (1866). Fossile Foraminiferen von Kar Nikobar. Reise der Österreichischen Fregatte Novara um die Erde in den Jahren 1857, 1858, 1859 unter den Befehlen des Commodore B. von Wüllerstorff-Urbair. Geologischer Theil (Zweite Abtheilung, Paläontologische Mittheilungen) 2(2): 187-268

Reviewer #3 (Remarks to the Author):

Overall the manuscript is much improved and I am impressed by how well and thoroughly you have addressed all the reviewers' (non-trivial) comments. The new version of the paper definitely comes across as more robust, and the introduction and set-up is much clearer. Well done. It's really nice to see the agreement between your study and those of McClelland and Saavedra-Pellitero. Below I point out a couple of major points that still need addressing before publication, and some suggestions for a number of minor corrections and edits.

Important comments:

New figure S3 is great. I am now fully on board with your interpretation of increased degree of calcification when average Noel coccolith mass increases. Personally, I think one of the curves (SN thickness or aspect ratio), or the entire figure, should be included in the main manuscript, so that readers are immediately aware that the mass results reflect an increased degree of calcification of coccoliths rather than a shift to larger coccoliths.

But be careful with wording (in S3 caption and throughout): (1) "Therefore, changes in coccolith mass document changes in the degree of cell calcification (i.e. PIC/POC ratio)." No! Changes in coccolith mass document changes in the degree of COCCOLITH calcification, which (based on results from living/cultured coccolithophores – cite refs) is thought to reflect an increase in the average PIC/POC ratio of cells.

(2) Stop using the term "calcification rates" (definition: CaCO₃ precipitated by an organism or a community per unit of time) – you cannot measure calcification rate using fossil coccoliths, living growing cells are needed!

In Fig S3: there is something wrong with the CO₂aq data – how can the concentration values be negative? The values cited in the text (line 189) do not match up with the values on the axis. Also the peaks in CO₂aq don't coincide at all with the peaks in coccolith mass/SN thickness, which is a bit problematic for your interpretation of what is driving an increase in degree of calcification, no??

Your "Obviously..." statement at the end of the Fig S3 caption is not obvious. There is no peak in coccolith mass during the later (YD) phase of enhanced SO upwelling as defined by your yellow bar, and you do not and cannot calculate "cell calcification rates".

During the Younger Dryas, 12-13 ka (yellow band on all figures), there is no peak in coccolith mass or abundance, like there is during H1. The increase in coccolith mass rather occurs immediately after the YD (as defined by the yellow bar), between 9-11.5 kyr. So throughout the manuscript, you need to stop referring to the two yellow-shaded intervals collectively as having "highest abundances/mass/calcification" (e.g. lines 134, 189, Figure 3 caption: "Yellow shading marks periods of reinvigorated SO upwelling during the last deglaciation, in conjunction to higher CCP strength and enhanced Biological Pump efficiency, at times of increased atmospheric pCO₂"). And you certainly need to address this mismatch in timing between inferred increased SO upwelling during the YD (indicated by your yellow band) and the peak in coccolith abundance/degree of calcification that occurs immediately afterwards (which is roughly in agreement with the vertical mixing proxies from the same core and your Br/Ca record). It's a bit confusing when you keep referring to a YD peak in coccolith mass/abundance that isn't there... maybe the answer is as simple as moving the yellow bar and not calling it YD? But I am not 100% sure which "SO upwelling" proxy records the yellow bar placements are based on.

The mass contribution of forams to the sediment CaCO₃ is more than twice that of all coccoliths, and more than 10 times that of the Noel group of coccoliths, which you say are the driver of changes in the CCP. This is interesting, and warrants discussion. Presumably forams also have a big impact on the CPP (or do you assume that PIC/POC changes in forams are constant – an unlikely scenario given assemblage changes and mass differences between species). Is the total coccolith mass based on counts from automatically processed images likely accurate or an underestimate?

I am quite excited to see that you have full planktonic foram assemblage data (I missed this the first time around). Are these published elsewhere? Can this also provide information on upwelling? e.g. deep versus shallow dweller abundance?

I suggest that integrating your Response 58 into the manuscript is a good idea, to explain the Holocene pattern.

Suggested edits:

Line 41: fixes

Line 42: delete comma

Line 43/44: does this mean the post-industrial S. Ocean is able to? Confusing sentence that starts in the past and ends in the present... Rewrite. Was unable (not has been)

Line 46: geometry is not the appropriate term in English... configuration? Pathways?

Line 48: why is Stratification in speech marks?

Line 49: macronutrient uptake

Line 52: in the ocean abyss

Sentence line 59-62: doesn't make sense. Delete comma after both.

Line 66: rephrase to $\delta^{15}N$ measurements and oxygenation proxies

Line 70: and mainly come from S. Atlantic sedimentary cores...

Line 82: replace yet with so far

Line 84: add "by calcifying plankton (primarily foraminifera, coccolithophores, pteropods)"

Line 86: clarify: offset the carbon removal from the surface ocean/atmosphere associated with the STP.

Line 88/91: it IS crucial! Coccolithophores ARE relevant

Line 91: group of single-celled phytoplankton

Sentence line 95 doesn't make sense.

Noëlaerhabdaceae is still spelt incorrectly throughout the paper.

Line 112: buried material. Seeing as you can't quantify what was remineralized at depth... This is important. You can't and don't reconstruct PIC:POC of sinking material.

Also: qualitatively document

Line 117: the release of

Line 160: "Indeed, variations in coccolith mass are usually thought to reflect variations in the thickness of an individual coccolith" – this is ONLY true when coccolith size/shape changes have been ruled out. So you can't put this sentence before the next one in its current form... change "are usually thought to " to "can under some circumstances".

Line 167: Change Besides to "We show/find that" or similar

Line 169: see comment on Fig S3 caption text.

Line 172: SN thickness does not "document molar PIC/POCcoccolith ratio of Noëlaerhabdaceae coccolithophores"

Sentence Line 176 does not flow very well.

Line 180: I think you mean $LOW CO_2aq$??

Line 193: inferred major upwelling phases

Line 195: and calcification (by changing morphotype dominance?)

Warming is as much in phase with the coccolith proxies as any of the other proxies that you say are of primary importance.

Line 205: origin

Line 221: burial ratio not rain ratio...

Line 233: associated with

Line 254: delete bio

Line 260: the response of calcifying plankton

LINE 266: was retrieved

Line 302: dutertrei

Reviewer #1 (Remarks to the Author):

The authors have provided a detailed and carefully weighted response to all of the comments raised during the review process. I just have a few minor points I would like to see addressed prior to publication:

Comment #1 (C1):

a) I still have some concerns about the location of the core, regarding its proximity to the continental margin. In their response the authors mention the site is characterised by a zone of high precipitation that results in high fluvial sediment supplies to the South Pacific Ocean. This results in a high sedimentation rate that facilitates decadal resolution during the last glacial termination. I would therefore still have some concerns about delineating vertical sedimentation from the upper-ocean and lateral fluvial input from the continent.

b) The authors argue in their reply and revised manuscript that our current understanding outside of the Atlantic sector is limited "little is known about deglacial export production patterns and STP strength in the Subantarctic Zone (SAZ) outside of the Atlantic sector." (Lines 63-64). This may indeed be true, but it was my understanding that the reason for focusing on the Atlantic sector is that this is the area where the iron-rich dust plumes from Patagonia are likely to have exerted maximal influence on productivity changes by alleviating iron limitation. Indeed the authors offer this rationale themselves: "However, these records remain sparse and largely concentrate on South Atlantic sedimentary cores located downwind of Patagonia, the most prominent dust source region to the Southern Ocean" (Lines 70-72).

The authors provide much more information in the revised version regarding the location of the core and clearly make distinctions about its characteristics and separating it from the Atlantic sector of the SAZ. I believe they now provide adequate information to the reader concerning these details about core location and the prevailing conditions, and make a good argument for considering productivity patterns at different locations.

Reply #1 (R1):

a) We thank Reviewer #1 for pointing out the confusion that remains regarding the possibility of lateral fluvial inputs in affecting our open-ocean micropalaeontological and geochemical signals. Previous mineralogical (Lamy et al., 1998) and geochemical (Hebbeln et al., 2000a) studies, as well as observations based on sediment traps deployed along the Chilean continental slope (Hebbeln et al., 2000b) argue against significant lateral transport of sediments in this area, and there are ample recent studies from the literature documenting sediment cores retrieved within the Chilean continental slope, as suitable archives to reconstruct oceanic patterns (Klump et al., 2001 ; Kaiser et al., 2005 ; Romero et al., 2006 ; Lamy et al., 2010 ; Caniupan et al., 2011 ; Saavedra-Pellitero et al., 2011 ; Kilian and Lamy, 2012 ; Lamy and Pol-Holz et al., 2013 ; Montade et al., 2013 ; Verleye et al., 2013 ; Chase et al., 2014 as examples). Besides, core MD07-3088 is constituted of homogeneous fine grained material, devoid of any turbidite (or other sediment reworkings) (Kissel, 2007). Most importantly, ^{24}C dating have been obtained for this core and exclude any significant sediment remobilisation (i.e. no age reversals) (Siani et al., 2013). This is confirmed by the summer SSTs record based on 158 foraminifera samples using the Modern

Analogue Technique (Siani et al., 2013 ; Haddam et al., 2016), and the vegetation record based on 137 pollen data (Montade et al., 2013) at site MD07-3088, that are well in phase with the main climatic events that characterize the Southern Ocean (Monnin et al., 2001 ; Marcott et al., 2014). At last, as mentioned in our previous reply, we are convinced that the overall good correlation of our micropaleontological and geochemical data with micropaleontological (Saavedra-Pellitero et al., 2011) and geochemical data from a wide range of location within the SO and the EEP (Anderson et al., 2009 ; Muratli et al., 2009; Martinez-Boti et al., 2015), as well as geochemical data from EDC ice core (Marcott et al., 2014), highlight regional open-ocean processes rather than lateral fluvial input. We are thus convinced that the sediment core primarily records changes in vertical particle flux and is only marginally affected by lateral advection of fluvial particles. We clearly emphasize it now in the « methods » section (lines 275-279).

b) Variations in eolian Fe supply to the Southern Ocean are frequently invoked to have affected glacial-interglacial atmospheric CO₂ variability, as Fe availability can modulate southern ocean phytoplankton primary and export productivity. This is particularly the case within the Atlantic sector (as highlighted by Reviewer #1). However, as mentioned in the manuscript : « the comparison with South Pacific and Indian Ocean records in particular is not straightforward because of differing modes of iron supply and contrasting factors modulating productivity patterns » (l. 74-76). Indeed, sedimentary (Morre and Braucher, 2008, Tagliabue et al., 2009) and hydrothermal (Tagliabue et al., 2010) Fe sources have also been highlighted, and dust may not be the only source of iron to the Southern Ocean (Tagliabue et al., 2014). Particularly, hydrothermal Fe sources seem to play an important role in the Pacific and Indian sectors of the Southern Ocean (Tagliabue et al., 2014). We now specify these aspects in the manuscript (lines 76-77). Besides, at site MD07-3088, as pointed out lines 134-141, phytoplankton growth is typically modulated by the supply of macronutrient via SO upwelling since river runoff and glacier erosion provide additional sources of micronutrients alleviating the limitation Fe is imposing on phytoplankton growth in the open Southern Ocean.

C2: Concerning the response of the authors to my original comment C6. I am aware that it is common in paleoceanographic studies to use sedimentary TOC values as a proxy for export productivity. I agree with their argument that as a first order principle one could expect higher productivity to result in higher sedimentary TOC values. From a qualitative point of view one therefore might be able to argue that these relationships are larger than those caused by selective degradation processes. The challenge here is that the paper attempts to use TOC as a quantitative estimate of POC export below the ventilation depth in comparison to calcite export to estimate the effect on the CCP. The authors now assume minimal calcite dissolution and TOC sedimentary values are 20-30% of ventilation depth export. Defining remineralization length scales of carbon in the BCP is an active field of research in contemporary oceanography. The authors claim that remineralization increases as a function of depth. This is correct, but it does not proceed in a linear fashion that would allow one to argue 1500m is an intermediate depth and thus reflects surface productivity. POC remineralization length scales typically follow a power-law relationship, reaching an asymptote in the bathypelagic ocean where typically <10% of surface-export reaches. These power-law curves are parameterized by an exponent *b*, which describes most of the attenuation in the surface ocean and has been shown to strongly depend on ecosystem structure. It therefore remains challenging to argue that changes in sedimentary TOC primarily reflect the strength of the biological carbon pump (i.e. POC flux out of the mixed layer) and not transfer efficiency (difference between POC flux at the mixed layer and into bathypelagic ocean (>1000m), the latter strongly related to ecosystem structure of export (plankton functional groups, aggregates, fecal pellets, etc). This is relevant if one attempts to use sedimentary TOC and calcite values to reconstruct CO₂ exchange in the surface ocean. The authors should also not confuse good preservation of material arriving at the seafloor with how representative what arrives at the seafloor is with respect to surface export patterns, which is of course of most relevance concerning the BCP. I accept the arguments that minimal test dissolution and few fragments is a reasonable basis to assume the calcite fluxes are representative of the ventilation depth. However, the same argument cannot be made for TOC, and since one must compare TOC and calcite to deduce the strength of the CCP, this needs to be made very clear. I realize it is challenging for the authors to achieve much more with their present data, but that it is not common for paleoceanography studies to account for these mechanistic processes is a poor argument for understating their significance. I would encourage the authors to be more explicit in the manuscript concerning the limitations of linking TOC to surface POC export and the impact that has on the quantitative reconstructions.

R2 : We certainly agree with Reviewer #1 that the relationship between sedimentary TOC content and the flux of POC is not straightforward. Indeed, complex remineralization processes occur within both the water column and the sediments. However, as already mentioned in our previous response, it appears that i) the latitudinal distribution pattern of TOC in surface sediments along the Chilean margin (Hebbeln et al., 2000a) closely resembles satellite-derived chlorophyll concentrations (Thomas et al., 1994), and ii) downcore trends in TOC contents recorded at site MD07-3088 mirror those from sites 1233 and 1234, located in the central Chilean margin (Muratli et al., 2009), indicating that sedimentary TOC concentrations can be used as a robust first-order proxy reflecting POC export, at least qualitatively. We now clarify these aspects in the method section "Sedimentary POC:PIC ratio vs POC:PIC rain ratio (1/ρ) (l. 459-487), and particularly lines 476-480).

As recommended, we now clearly separate the discussions related to the remineralization processes taking place within the water column and those occurring within the sediments (diagenetic processes) (l. 460-487).

As mentioned by Reviewer #1, the water column witnesses biological consumption and remineralization processes that reduce the downward flux of POC, and thus the efficiency of carbon sequestration and burial. However, depending on the ecosystem structure (pelagic food web structure, proportion of fecal pellets versus phytoplankton aggregates, proportion of lithogenic material, fraction of export associated with ballasting minerals and their sinking rates, water temperature, carbon demand of the mesopelagic bacteria, zooplankton communities ...), there is a large variability in POC transfer efficiency, globally (T_{eff} : ratio of exported POC flux (within the water column, and usually within the twilight zone) to primary productivity (within the surface mixed waters)) (Buesseler, 1998 ; Dunne et al., 2007 ; Buesseler et al., 2007, 2008 ; Buesseler and Boyd, 2009 ; Smetacek et al., 2012 ; Turner, 2015). Since the 80's, there have been numerous attempts to parametrize the depth-dependent particle flux attenuation (e.g. Martin et al., 1987 ; François et al., 2002 ; Armstrong et al., 2002 ; Buesseler and Boyd, 2009). We certainly concur that bacterial degradation does not proceed linearly (as we previously assumed), but rather follows power-law or exponential relationships. However, the global relevance of these relationships remains questionable, and while observations often report T_{eff} of POC <10%, there are numerous examples for which T_{eff} reaches substantially higher values (Dunne et al., 2007), and may attain T_{eff} values as high as 50% at 500 m (Buesseler et al., 2008), and below the twilight zone (i.e. below 1000 m (Smetacek et al., 2012) and even below 3000 m water depth (Buesseler, 1998)). Furthermore, i) high-latitude ecosystems, particularly under iron-replete conditions are often characterized by efficient POC export (Buesseler, 1998 ; Dunne et al., 2007 ; Smetacek et al., 2012); ii) bacterial degradation is temperature-dependent and as such remineralization rates are substantially slower in (sub)polar environments, and iii) near-shelf ecosystems characterized by high lithogenic fluxes acting to protect organic matter from bacterial degradation (Dunne et al., 2007), are typical environments that favour efficient OM export and low subsurface flux attenuation. These ecosystems are typically characterized by T_{eff} zone averaging 10 to 50%. Besides, in such environments, a substantial portion of POC reaches the sea floor. Therefore, upon consideration of all these aspects, we infer that sedimentary TOC concentration at site MD07-3088 must record OM export with minimal bias (l. 480-482). Therefore, to consider wide range of plausible T_{eff} (see references above), we now test the impact of BCP for HS1 and ACR, in cases for which 10 to 50% of the exported POC, is preserved within the sediments (instead of the previous values of 20-30% as highlighted by Reviewer #1) (Fig. 5 and lines 235-239 and 748-757) of the new manuscript). It is not possible to quantitatively constrain possible changes in T_{eff} patterns over the last 3-20 kyrs. However we infer that the wide range of T_{eff} values considered in our sensitive study, provides a conservative approach that may be valid in the past as well.

C3: It seems to me from Figure 3 that foraminifer calcite is quantitatively more important than calcite biomass by a factor of around 3 (compare Figures 3b and 3c). It is therefore important to understand how much calcifying benthic foraminifer might contribute to this. The authors have responded by stating that they weighed a few discrete fractions of benthic foraminifer and it ranges from 0.3 to 1 mg/g. It would be useful to add here how this compares with total planktonic calcite biomass (forams + coccoliths). It seems from rapidly consulting Figure 3b that this is similar to the total calcite coccolith biomass (0.4 – 2 mg/g).

R3 : We apologize for not having correctly mentioned that Fig. 3c (now Fig. 4c) does not represent calcite mass of the overall foraminifera fraction but only relates to planktonic foraminifera, based on estimates of planktonic foraminifera abundance and mean weight as detailed in the Methods l. 400-433). Therefore, the calcite mass of benthic foraminifera is not included within these estimates, and the term « planktonic » has been added within new caption of Figure 4c.

Indeed, as pointed out by Reviewer #1, the planktonic foraminifera calcite flux is higher than the coccolith flux during the deglaciation and it is the opposite during the Holocene. It is worth noting that the proportions of calcite produced by both planktonic foraminifera and coccolithophores are underestimated, as emphasized in the revised manuscript lines 331-334 and 420-422). Indeed, our estimates do not take into account <150 μ m planktonic foraminifera species such as *G. uvula*, partly *G. quinqueloba*, and juvenile specimens, while SYRACO might underestimate the number of coccolith per g of sediments, which has implications for the estimates of the coccolith calcite masses presented in Figures 3a and 3b (now Fig. 4a and 4b) (ans see R10). However, these estimations do not have any repercussions in the trends of foraminifera and coccolith calcite masses that remain relevant.

Below is the distribution of the benthic foraminifera calcite mass over the last 20-3 kyrs, ranging from 0.1 to 1.0 mg/g, with an average of 0.29 mg/g. It is usually in lower proportions than the estimates of average coccolith calcite mass (ranging from 0.22 to 2.59 mg/g, with an average of 0.71 mg/g), but appear to be in the same order of magnitude as indicated by the Reviewer #1.

However, as the calcite mass of benthic foraminifera is not of interest for CCP efficiency (and is significantly lower than coccoliths and planktic foraminifera calcite masses), we think that those data do not add anything much to the paper and removed the lines dealing with benthic foraminifera from the method section (previous lines 440-443).

C4: Reviewer 2 comment C16
 The reviewer raises a point here about the effect of differing atmospheric CO₂ on coccolith mass. The authors have engaged in a sufficient and detailed rebuttal. However, I wonder about foraminifera, which appear to be the dominant component of calcite mass (according to Figures 3b and 3c). It has been shown previously that *G. bulloides* test weights are strongly impacted by CO₂ water chemistry over glacial cycles (Moy et al. 2009, Nature Geoscience doi :10.1038/ngeo460). Does that play a role here?

R4 : Following reviewer #1's recommendation, we have now computed the weights of 6 to 60 specimens of *G. bulloides* (the most abundant planktonic specie at site MD07-3088), covering a wide size range (150-200 µm, 200-250 µm, 250-315 µm, 315-355 µm, 355-400 µm and 400-450 µm) for 16 depth intervals (including the LGM, the HS1, the ACR, the YD and the Holocene). We report mean weights decreasing by about 20% for the different sizes, from the LGM to the Holocene, and of about 7% and 18% during HS1 and YD respectively. If similar weight decreases are observed within the other planktonic species, the magnitude of the changes in the overall weight (~20%) would be not sufficient enough to significantly change the estimated planktonic foraminifera mass flux. Indeed, because of the drastic increases within the planktonic foraminifera abundance during these time intervals (more than one order of magnitude), fluctuations in the planktonic foraminifera weights would imply changes in the flux of planktonic foraminifera calcite mass that remain within the error bars. We now specify this lines 422-433.

Reviewer #2 (Remarks to the Author)

Duchamp-Alphonse et al. submitted a sufficiently revised manuscript. Hence, I suggest, to accept it for publication.

We thank reviewer #2 for her/his positive and supportive assessment of our work.

C5 : However, the only thing that has to be changed before, is (still) the nomenclature of the benthic forams used. In lines 722 and 737 you (incorrectly) refer to *C. wuellerstorfi*. The original description was done by Schwager (1866) as *Anomalina wuellerstorfi*. You might argue to call it *Cibicidoides*, *Cibicides* or *Fontbotia*, however *wuellerstorfi* stays *wuellerstorfi*.

Schwager, C. (1866). Fossile Foraminiferen von Kar Nikobar. Reise der Österreichischen Fregatte Novara um die Erde in den Jahren 1857, 1858, 1859 unter den Befehlen des Commodore B. von Wüllerstorff-Urbair. Geologischer Theil (Zweite Abtheilung, Paläontologische Mittheilungen) 2(2): 187-268

R5 : amended (lines 711 and 740 of the new manuscript).

Reviewer #3 (Remarks to the Author):

Overall the manuscript is much improved and I am impressed by how well and thoroughly you have addressed all the reviewers' (non-trivial) comments. The new version of the paper definitely comes across as more robust, and the introduction and set-up is much clearer. Well done. It's really nice to see the agreement between your study and those of McClelland and Saavedra-Pellitero. Below I point out a couple of major points that still need addressing before publication, and some suggestions for a number of minor corrections and edits.

We thank reviewer #3 for her/his positive comments and are glad s/he feels that our manuscript has improved substantially.

Important comments:

C6 : New figure S3 is great. I am now fully on board with your interpretation of increased degree of calcification when average Noel coccolith mass increases. Personally, I think one of the curves (SN thickness or aspect ratio), or the entire figure, should be included in the main manuscript, so that readers are immediately aware that the mass results reflect an increased degree of calcification of coccoliths rather than a shift to larger coccoliths.

C6 : We agree with Reviewer # 3 that including Figure S3 to the main manuscript helps clarifying our argumentation. Indeed, the figure directly documents the wide coccolith dataset we obtained and highlights the usefulness of coccolith mass changes in reflecting the degree of coccolith calcification (comparison with SN thickness and aspect ratio). Supplementary Figure 3 is now Figure 3.

C7 : But be careful with wording (in S3 caption and throughout): (1) "Therefore, changes in coccolith mass document changes in the degree of cell calcification (i.e. PIC/POC ratio)." No! Changes in coccolith mass document changes in the degree of COCCOLITH calcification, which (based on results from living/cultured coccolithophores – cite refs) is thought to reflect an increase in the average PIC/POC ratio of cells. (2) Stop using the term "calcification rates" (definition: CaCO₃ precipitated by an organism or a community per unit of time) – you cannot measure calcification rate using fossil coccoliths, living growing cells are needed!

R7 : We agree with Reviewer #3 and the terms « calcification rates » as well as « cell calcification (i.e. PIC/POC ratio) » have been removed from the revised manuscript as recommended (text and Figure captions). We now refer to « degree of coccolith calcification » or « coccosphere calcite quota » that are more appropriated terminologies when dealing with fossil coccolith records (Henderiks and Pagani, 2008 ; Beaufort et al., 2011 ; O'Dea et al., 2014) (lines 158, 161, 163-164, 167, 170-171 in the text, and line 731 in the caption of Fig. 3 (previous Fig. S3).

C8 : In Fig S3: there is something wrong with the CO_{2aq} data – how can the concentration values be negative? The values cited in the text (line 189) do not match up with the values on the axis. Also the peaks in CO_{2aq} don't coincide at all with the peaks in coccolith mass/SN thickness, which is a bit problematic for your interpretation of what is driving an increase in degree of calcification, no??

R8 : Fair point. We apologize for omitting to mention that CO₂ concentrations values we previously had plotted on the y-axis of previous Supplementary Figure 3 (now Fig. 3, see C6), represent [CO_{2aq}] anomalies, i.e. changes in local [CO_{2aq}] with respect to modern conditions (thus explaining negative values). The values cited in the text refer to the magnitude of the rising [CO_{2aq}] anomalies, and actually match the trends observed in the curve. Estimations of [CO_{2aq}] at site MD07-3088 were required by Reviewer # 3 in the previous review, in order to better compare our coccolith calcification trends to deglacial coccolith and [CO_{2aq}] patterns from the literature (previous

Comment C52). However, since we could not provide $[CO_{2aq}]$ at our site, we have compared our micropaleontological data with the only surface water $[CO_{2aq}]$ that exist in the SAZ so far, and that have previously been obtained in the Atlantic sector of the Southern Ocean (site PS2498-1 ; Martinez-Boti et al., 2015). As mentioned by the authors in the method section (Martinez-Boti et al., 2015), local constant reservoir corrections were used at site PS2498-1 and different ΔR values may affect the chronology by up to 0.9 kyr for the LGM and the early deglaciation. Indeed, it is probable that the small mismatch highlighted by Reviewer # 3 (particularly during the early deglaciation) between coccolith calcification and $[CO_{2aq}]$ data reflect small (yet significant) age model uncertainties. Since « these potential inherent age model uncertainties do not impact our general conclusions unduly » (as mentioned by Martinez-Boti et al., 2015), it still remains interesting to compare long-term trends of $[CO_{2aq}]$ to our coccolith calcification pattern. Indeed, the increase in coccolith calcification and $[CO_{2aq}]$ observed at/around HS1 and YD are in the same order of magnitude than those recorded for the penultimate deglaciation (McClelland et al., 2016) (l. 186-189). However, because of the age model uncertainties, it seems more accurate to remove the $[CO_{2aq}]$ curve from our manuscript, and only refer to Martinez-Boti et al. (2015) (and McClelland et al., 2016) within the text.

C9 : Your “Obviously...” statement at the end of the Fig S3 caption is not obvious. There is no peak in coccolith mass during the later (YD) phase of enhanced SO upwelling as defined by your yellow bar, and you do not and cannot calculate “cell calcification rates”.

During the Younger Dryas, 12-13 ka (yellow band on all figures), there is no peak in coccolith mass or abundance, like there is during H1. The increase in coccolith mass rather occurs immediately after the YD (as defined by the yellow bar), between 9-11.5 kyr. So throughout the manuscript, you need to stop referring to the two yellow-shaded intervals collectively as having “highest abundances/mass/calcification” (e.g. lines 134, 189, Figure 3 caption: “Yellow shading marks periods of reinvigorated SO upwelling during the last deglaciation, in conjunction to higher CCP strength and enhanced Biological Pump efficiency, at times of increased atmospheric pCO_2 ”).

And you certainly need to address this mismatch in timing between inferred increased SO upwelling during the YD (indicated by your yellow band) and the peak in coccolith abundance/ degree of calcification that occurs immediately afterwards (which is roughly in agreement with the vertical mixing proxies from the same core and your Br/Ca record). It's a bit confusing when you keep referring to a YD peak in coccolith mass/abundance that isn't there... maybe the answer is as simple as moving the yellow bar and not calling it YD? But I am not 100% sure which “SO upwelling” proxy records the yellow bar placements are based on.

R9 : We do not totally agree with Reviewer # 3. The yellow shading that highlights increased upwelling rates characteristic of the YD, is based on $\Delta^{14}C$ B-P previously reported by Siani et al., (2013) (see Figure 2) between 12.7 and 11.3 kyrs. This time interval is associated with substantially higher values in coccolith abundances/mass/calcification (Figures 2- 4, as well as Supplementary Figure 2 of the new manuscript). Our results are entirely consistent with higher surface ocean fertility and $[CO_2 aq]$ associated with SO upwelling. However, as noted by Reviewer #3, the peak values do not fall within the YD, within the age model uncertainties. As the $\Delta\delta^{13}C$ values have been collected at a much higher temporal resolution than $\Delta\Delta^{14}C$ (Siani et al., 2013), it seems relevant to delineate the upwelling phases according to both, the constraints provided by $\Delta\Delta^{14}C$ and $\Delta\delta^{13}C$ rather than $\Delta\Delta^{14}C$ alone. This is particularly true for (and around) the YD where the upwelling phase is lasting well into the early Holocene (which is like moving the left limit of the yellow bar as suggested Reviewer # 3). In that way, the yellow bar associated to the YD now includes both, the increasing trends and the major peaks in coccolith abundances/masses/calcification. As mentioned by Reviewer # 3, this new representation of the « yellow bar » helps highlighting the strong relationship that exists between the coccolith and the high resolution vertical mixing proxy $\Delta\delta^{13}C$ (and $\Delta\Delta^{14}C$) data at site MD07-3088 (See Figure 2)

Remark: the term "cell calcification rates" has been removed from the new manuscript (see R7 for the detail answer).

C10 : The mass contribution of forams to the sediment $CaCO_3$ is more than twice that of all coccoliths, and more than 10 times that of the Noel group of coccoliths, which you say are the driver of changes in the CCP. This is interesting, and warrants discussion. Presumably forams also have a big impact on the CPP (or do you assume that PIC/POC changes in forams are constant – an unlikely scenario given assemblage changes and mass differences between species). Is the total coccolith mass based on counts from automatically processed images likely accurate or an underestimate?

R10 : As also pointed out by Reviewer # 1, foraminifera calcite mass is quantitatively more important than coccolith calcite mass by a factor of 2-3. We certainly agree that planktonic foraminifera affect the strength of the CCP, and accordingly we systematically considered both foraminifera and coccolith calcite masses when discussing the strength of the CCP (lines 212-214, 225-229, 244-246 and 250-252).

As correctly pointed out by reviewer # 3, SYRACO tends to underestimate the number of coccoliths per gram of sediments and thus the calcite mass produced by coccolithophores. In fact the software has difficulties in taking into consideration unfocussed images. We now emphasize this potential shortcoming in the revised version of the manuscript, presenting SYRACO results more cautiously (l 331-334). However, while this possible limitation may contribute to reduce the total number of coccolith as well as the total amount of calcite associated to coccoliths, in no way, it impacts the trends/patterns of coccolith abundances/masses/calcification and our estimates are conservative.

C11 : I am quite excited to see that you have full planktonic foram assemblage data (I missed this the first time around). Are these published elsewhere? Can this also provide information on upwelling? e.g. deep versus shallow dweller abundance?

R11 : Actually, the curves documenting the number of planktic foraminifera per g of sediment, and the SST (based on Modern analog Technique) are published (Siani et al., 2013; Haddam et al., 2016); the data related to the calcite mass are new. As changes in foraminifera at site MD07-3088 do not document local (or coastal) upwelling but rather increased availability of nutrients derived from enhanced upwelling at the Antarctic divergence, we are not aware that planktic assemblages could clearly provide information on such process. Furthermore, site MD07-3088 is located within the subantarctic area and we do not know a good indicator of low/high nutrient content in such a context, as it is usually documented by the *G.ruber*/*G. bulloides* ratio in tropical settings.

C12 : I suggest that integrating your Response 58 into the manuscript is a good idea, to explain the Holocene pattern.

R12 : According to the micropalaeontological and geochemical data we have collected so far, the best way to explain the increase in Br/Ca ratio over the last 3-6 kyrs is to hypothesize an increase in non-calcareous phytoplankton abundances such as diatoms or picoplankton. However, without any diatoms (or picoplankton) data, it does not seem relevant to discuss these aspects within the manuscript.

Suggested edits:

Almost all the comments raised below by Reviewer # 3 have been positively considered and changes have been done (see the new line numbers that are notified to the right of the required changes). Detailed replies to comments appear in italics when needed.

Line 41: fixes (l. 41).

Line 42: delete comma (l. 42).

Line 43/44: does this mean the post-industrial S. Ocean is able to?

Yes, indeed. Due to the modern atmospheric concentrations of CO₂ (averaging 400 ppm), the air-sea flux of CO₂ within the Southern Ocean has been changed, and this overall area now represents a sink rather than a source for atmospheric CO₂ (Morrison et al., 2015).

Confusing sentence that starts in the past and ends in the present... Rewrite. Was unable (not has been)
We have rewritten the sentence in the past. Now « was » replaces « has been ». (l. 42-45).

Line 46: geometry is not the appropriate term in English... configuration? Pathways?

We've changed to « Pathways ». (l. 46).

Line 48: why is Stratification in speech marks?

Speech marks have been removed. (l. 48).

Line 49: macronutrient uptake (l. 49).

Line 52: in the ocean abyss (l. 51).

Sentence line 59-62: doesn't make sense. Delete comma after both. (l. 59-62).

Line 66: rephrase to d15N measurements and oxygenation proxies (l. 66).

Line 70: and mainly come from S. Atlantic sedimentary cores...(l. 70).

Line 82: replace yet with so far.

« Yet » has been replaced with « thus far » (l. 84)

Line 84: add "by calcifying plankton (primarily foraminifera, coccolithophores, pteropods)".

We've added " by calcifying plankton (primarily foraminifera and coccolithophores)" (l. 86).

Line 86: clarify: offset the carbon removal from the surface ocean/atmosphere associated with the STP.

In this sentence, « offset » means « compensate », « balance ». We prefer to leave the sentence as it is. (l. 89).

Line 88/91: it IS crucial! Coccolithophores ARE relevant (l. 90 and 93).

Line 91: group of single-celled phytoplankton (l. 93-94).

Sentence line 95 doesn't make sense.

We removed this sentence.

Noëlaerhabdaceae is still spelt incorrectly throughout the paper.

« Noëlaerhabdacea » has been changed to « Noëlaerhabdaceae » l. 101 and elsewhere.

Line 112: buried material. Seeing as you can't quantify what was remineralized at depth... This is important. You can't and don't reconstruct PIC:POC of sinking material. (l. 111).

Also: qualitatively document (l. 112).

Fair point. As required by Reviewer # 1, we now better discuss the relationship that exists between sedimentary TOC and the flux of POC that exits the surface mixed waters (See « methods l. 459-482). The sentences are now « We compare these micropalaeontological data with reconstructed past changes in the buried POC:PIC ratio, suggested to reflect the C-rain ratio. These records are the first, to our knowledge, to qualitatively document the relative contribution of both the STP and CCP to the deglacial rise in atmospheric CO₂ at a decadal scale. (l. 110-113).

Line 117: the release of (l. 117).

Line 160: "Indeed, variations in coccolith mass are usually thought to reflect variations in the thickness of an individual coccolith" – this is ONLY true when coccolith size/shape changes have been ruled out. So you can't put this sentence before the next one in its current form... change "are usually thought to " to "can under some circumstances". (l. 159-160).

Line 167: Change Besides to "We show/find that" or similar

Besides has been replaced by « We find that » (l. 165).

Line 169: see comment on Fig S3 caption text.

« Degree of cell calcification » has been changed to « coccolithophore calcite quota » in the new caption of Fig.3.

Line 172: SN thickness does not "document molar PIC/POCcoccolith ratio of Noëlaerhabdaceae coccolithophores"

Indeed. the sentence now reads: « document the degree of Noëlaerhabdaceae coccolith calcification ». (l. 170-171).

Sentence Line 176 does not flow very well.

This sentence has been lightened to : « However, in the geological record, i.e. when general selection for growth strategies²⁷ and phenotypic plasticity naturally occurred and regulated the carbon acquisition within the cell⁴⁷, heavier calcified coccoliths were always associated with increased atmospheric carbon dioxide concentrations^{26,27,48,49}. (l. 174-177).

Line 180: I think you mean LOW CO_{2aq}??

No. We mean High pCO₂ (Bolton and Stoll, 2013). (l. 177).

Line 193: inferred major upwelling phases

Increased temperatures may occur during the major upwelling phases. However, increased temperatures are not infer to major upwellings. As already detailed in our reply to previous comments (R53), temperatures and upwelling are not directly connected.

Line 195: and calcification (by changing morphotype dominance?)

No. Our data show that increased temperatures may have partially contributed in promoting coccolith production and calcification, through increasing coccolith abundance and mass within the same morphotypes (particularly within the Noëlaerhabdaceae) (see the chapter "Increase in subantarctic surface water fertility and [CO_{2aq}] during Southern Ocean upwelling (l. 122-193) and Supplementary Figure 2.

Warming is as much in phase with the coccolith proxies as any of the other proxies that you say are of primary importance.

We do not agree with that (see Figures 2a-2e).

Line 205: origin (l. 201).

Line 221: burial ratio not rain ratio...

Considering the comments C2 of Reviewer # 1, we now clearly distinguish the buried POC:PIC ratio to the POC:PIC rain ratio. We have thus rewritten this sentence as follow :
« Since the sedimentary POC:PIC ratio changes eventually document the changes of C rain-ratio (POC:PIC) (see Methods), the downcore Br/Ca ratio is used to provide an estimate of the strength of the STP relative to the CCP, which serves as a robust tool to reconstruct decadal changes in the BCP efficiency ». (l. 214-218).

Line 233: associated with

The increase in the CCP is not only associated with rising macronutrient availability, but is associated to rising macronutrient availability. This relationship is well documented lines 132-193 for coccolithophores, since increased rates in coccolith abundance and mass are recorded due to increased fertility and [CO_{2aq}] conditions (associated to the major upwelling phases).

Line 254: delete bio (l. 250).

Line 260: the response of calcifying plankton (l. 256).

LINE 266: was retrieved (l. 261).

Line 302: dutertrei (l. 289).

References

- Anderson, R., et al., Science 323, 1443-1448 (2009).
- Armstrong, R.A., et al., Deep Sea Research II, 49, 219-236 (2002).
- Beaufort, L., et al., Nature 476, 80-83, doi:10.1038/nature10295 (2011).
- Bolton, C. and Stoll, H., Nature 500, 558-562, doi:10.1038/nature12448 (2013).
- Buesseler, K.O., Global Biogeochemical Cycles 12 (2), 297-310 (1998).
- Buesseler, K.O. et al., Science 316, 567-570, (2007).
- Buesseler, K.O., et al., Deep-Sea Research II 55, 1522-1539 (2008).
- Buesseler, K.O., Boyd, P.W., Limnol. Oceanogr. 54 (4) 1210-1232 (2009).
- Caniupan, M., et al., Paleoceanography 26, PA3221, doi:10.1029/2010PA002049 (2011).
- Chase, Z., et al., Quaternary Science Reviews 99, 135-145 (2014).
- Dunne, J.P., et al., Global Biogeochemical Cycles 21, GB4006, doi:10.1029/2006GB002907 (2007).
- François, R. et al., Global Biogeochemical Cycles 16, 1087, doi: 10.1029/2001GB001722 (2002).
- Haddam, N.A., et al., Paleoceanography 31, 822-837, doi: 10.1002/2016PA002946 (2016).
- Hebbeln, D., et al., Marine Geology 164, 119-137 (2000a).
- Hebbeln, D., Deep-Sea Research II (47), 2101-2128 (2000b).
- Henderiks, J., and Pagani, M., Earth and Planetary Science Letters 269, 575-583
- Kaiser, J., et al., Paleoceanography 20, PA4009, doi:10.1029/2005PA001146 (2005).
- Kilian, R. and Lamy, F., Quaternary Science Reviews 53, 1-23 (2012).
- Kissel, C., MD159-PACHIDERME IMAGES XV Cruise and Data Reports (2007).
- Klump, J., et al., Marine Geology 177, 1-11 (2001).
- Lamy, F., et al., Geologische Rundschau 87, 477-494, (1998).
- Lamy, F., et al., Nature Geoscience 3, doi :10.1038/NGEO959 (2010).
- Lamy, F., and Pol-Holz, R., Encyclopedia of Quaternary Science 3, pp73-85 (2013).
- Marcott, S.A., et al., Nature 514, 616-619, doi :10.1038/nature13799 (2014).
- Martin, J.H., et al., Deep-Sea Research A 34, 267-285 (1987).
- Martinez-Boti, M.A., et al., Nature 518, 219-222 (2015).
- McClelland, H.L.O., et al., Scientific Reports, 6:34263, DOI: 10.1038/srep34263 (2016).
- Monnin, E., et al., Science 291, 112-114 (2001).
- Montade, V., et al., Palaeogeography, Palaeoclimatology, Palaeoecology 369, 335-348 (2013).
- Moore, J.K., Baucher, O., Biogeosciences 5 (3), 631-656, doi : 10.5194/bg-5-631-2008 (2008).
- Morrison, A., Frölicher, T., Sarmiento, J., Upwelling in the Southern Ocean, Physics today, 27-32 (2015).
- Moy et al., Nature Geoscience doi :10.1038/ngeo460 (2009).
- Muratli, J.M., et al., Nature Geoscience 3, 23-26, doi: 10.1038/NGEO715 (2009).
- O'Dea, S.A., et al., Nature Communications 5:5363, doi: 10.1038/ncomms6363 (2014).
- Romero, O.E., et al., Quaternary Research 65, 519-525 (2006).
- Saavedra-Pellitero, M., et al., Paleoceanography 26, PA1201, doi:10.1029/2009PA001824 (2011).
- Siani, G., et al., Nature Communications 4/2758, doi: 10.1038/ncomms3758 (2013).
- Smetacek, V., et al., Nature 487, 313-319, doi:10.1038/nature11229 (2012).
- Tagliabue, A., et al., Geophysical Research Letters 36, L13601, doi: 10.1029/2009GL038914 (2009).
- Tagliabue, A., et al., Nature Geosciences 3, 252-256, doi: 10.1038/ngeo818 (2010).
- Tagliabue, A., et al., Geophysical Research Letters, 41, 920-926, doi : 10.1002/2013GL059059 (2014).
- Thomas, A.C., et al., Journal of Geophysical Research 99 (C4), 7355- 7370 (1994).
- Turner, J.T., Progress in Oceanography 130, 205-248 (2015).
- Verleye, T.J., et al., Quaternary Research 80 (3), 495-501 (2013).

Reviewers' comments:

Reviewer #1 (Remarks to the Author):

The authors should be commended for responding to all reviewers comments in a comprehensive manner. I am now happy for this paper to proceed to publication.

Reviewer #3 (Remarks to the Author):

Overall the revised paper is better than previous versions. I have a few final comments to improve clarity and robustness of interpretations.

Comment on the authors' response to Reviewer 1 on TOC content and its relationship to POC export.

As a paleoceanographer, I find some of the responses of the authors to these important comments throughout the review process rather disheartening. Indeed, it is not common in our discipline to interpret TOC MAR or percent as a quantitative export productivity proxy, and generally this parameter is considered in combination with other independent indicators of export productivity to provide a qualitative picture of paleo export productivity. I therefore agree with Reviewer 1 that the authors need to acknowledge very explicitly the limitations of their organic carbon pump reconstructions. By not doing so, they won't do themselves any favours because, if the relationship between TOC accumulation and POC export is not discussed in detail, readers will question their understanding of the system and therefore the conclusions drawn. In the latest version, the authors discuss these limitations in more detail, which is an improvement. The authors state that "downcore trends in TOC contents recorded at site MD07-3088 mirror those from sites 1233 and 1234, located in the central Chilean margin (Muratli et al., 2009), indicating that sedimentary TOC concentrations can be used as a robust first-order proxy reflecting POC export, at least qualitatively" – First, I don't think they really mean "mirror" (opposite patterns), secondly, I don't see why similar patterns in three locations means that all (or any) of them consequently must represent export productivity changes.

R8: You point out potential differences between your age model and that of Martinez Boti used for the EEP [CO₂aq] curve as the likely explanation for the difference in timing between EEP [CO₂aq] changes and calcification changes in your core, in the SE Pacific. Another possibility is that the age model match is OK, and upwelling/CO₂aq changes were not exactly synchronous between the two sites. Maybe it is worth mentioning this. Or mention that your preferred interpretation is that the mismatch between peaks results from age model offsets of up to xx kyr during the deglaciation, as written in your response.

R12: are there no opal MAR records, or even smear slide descriptions, from your core that may shed light on this hypothesis? It seems a shame to not discuss this Holocene "mismatch" compared to your interpretation of your records in the rest of the core, to ignore this interval rather degrades your overall interpretations of proxy records.

I understand that, because of a lack of air-sea equilibrium wrt CO₂ in your study region, you cannot use your core's SST record to reconstruct local [CO₂aq] using ice-core pCO₂. However, because your local [CO₂aq] will be very sensitive to local temperature changes and local increases in upwelling (cooling or increased upwelling -> higher [CO₂aq]), is the record of Martinez-Boti really likely to be representative of what coccolithophores are experiencing at your site? This warrants a brief sentence or two of discussion.

Line 233 (and other places): associated with

"The increase in the CCP is not only associated with rising macronutrient availability, but is associated to rising macronutrient availability. This relationship is well documented lines 132-193

for coccolithophores, since increased rates in coccolith abundance and mass are recorded due to increased fertility and [CO₂aq] conditions (associated to the major upwelling phases)."
This is simply a matter of correct English usage. "associated to" is not grammatically correct. Maybe you prefer to say "is associated with and likely driven by" increased macronutrient availability?

Line 24: coccolith abundance

Line 96: "accounts for a significant proportion of the global marine export production" reference needed.

Line 116 and paper title: I am not entirely sure what you mean by "enhancing ocean-atmosphere carbon partitioning", you need to be more explicit. I am not sure if you are trying to say "reducing exchange between surficial and deep ocean carbon reservoirs (e.g. by increased stratification between surface and deep ocean)", "enhancing ocean carbon uptake from the atmosphere" or "enhancing C uptake by the surface ocean and export towards the deep ocean"... or something else. I have the same problem with the paper's title, and I strongly suggest you change it to something clearer.

Line 127: Add Site 1233 to the map.

Line 144: have an affinity

Line 177: more heavily calcified coccoliths

Line 188: ~ 2-3 µm/L rises in SSW [CO₂aq] in the EEP

A few sentences explaining why summer SST increases occur during periods of enhanced upwelling would be appropriate – usually increased upwelling is associated with cooling, which would effectively enhance your [CO₂aq] increase, rather than counteract it because CO₂ is more soluble in colder water. Does upwelling likely occur mainly in winter?

Line 214: calcifiers

Line 217: rephrase. E.g., We suggest that, in our core, POC:PIC ratio changes in the sediments likely reflect changes in the C rain ratio (POC:PIC) (See methods). Therefore, downcore Br/Ca....

Line 433: decrease by

Reviewer #1 (Remarks to the Author):

Comment #1 (C1):

The authors should be commended for responding to all reviewers comments in a comprehensive manner. I am now happy for this paper to proceed to publication.

Reply #1 (R1):

We sincerely thank Reviewer #1 for all of his/her constructive comments, which we feel really contributed to improve our manuscript.

Reviewer #3 (Remarks to the Author)

C2: Overall the revised paper is better than previous versions.

R2: We certainly agree with Reviewer # 3 and feel that we provided a rather complete and thoroughly revised manuscript. Indeed, all of the issues raised by the three reviewers have been carefully taken into account, thus resulting over the past 7 months, in really exciting and constructive discussions. The remarks really helped improving the manuscript, and we did our best to integrate the required changes in the most comprehensive way possible (as pointed by Reviewer #1).

I have a few final comments to improve clarity and robustness of interpretations.

C3: Comment on the authors' response to Reviewer 1 on TOC content and its relationship to POC export. As a paleoceanographer, I find some of the responses of the authors to these important comments throughout the review process rather disheartening. Indeed, it is not common in our discipline to interpret TOC MAR or percent as a quantitative export productivity proxy, and generally this parameter is considered in combination with other independent indicators of export productivity to provide a qualitative picture of paleo export productivity. I therefore agree with Reviewer 1 that the authors need to acknowledge very explicitly the limitations of their organic carbon pump reconstructions. By not doing so, they won't do themselves any favours because, if the relationship between TOC accumulation and POC export is not discussed in detail, readers will question their understanding of the system and therefore the conclusions drawn. In the latest version, the authors discuss these limitations in more detail, which is an improvement.

R3: As pointed by Reviewer #3, the use of TOC MAR as a quantitative export productivity proxy is rather uncommon and often export production reconstructions are based on specific organic-(e.g. alkenones) and/or inorganic components (such as biogenic SiO₂ and CaCO₃) that are hypothesized to reflect marine TOC MAR and thus POC export (e.g. Kohfeld et al., 2005; Kumar et al., 1995; Lamy et al., 2014; Gottschalk et al., 2016; Martinez-Garcia et al., 2009, 2014; Muratli et al., 2010). In all cases, remineralization processes and diagenetic alterations occur and must be critically discussed and ruled out before interpreting the downcore records.

Here, we provide a decadal record of the marine POC preserved within the sediments, based on a combination of TOC, Br_{XRF}, δ¹³C_{org} and C/N data. We note that all proxies provide an internally consistent picture and thus, we are confident that TOC MARs can be used as a first-order proxy to reconstruct past changes in POC flux. Moreover, TOC MARs certainly need to be considered with a grain of salt in slowly-accumulating, well-oxygenated pelagic sediments. However, potentially biasing effects are minimized in rapidly accumulating sediments in marginal settings, where the oxygen exposure is reduced.

While we certainly agree that the first revised version of our manuscript lacked a detailed discussion related to organic matter degradation we are now confident that the last version of our manuscript clarifies these aspects (as described in our previous reply R2, and as observed by Reviewer # 1 above). As mentioned in our previous reply R2, we now have considered a wide range of plausible T_{eff} values at site MD07-3088. These aspects are clearly outlined in both the main part of the text (Fig. 5 and lines 214-257) and the supplementary material (l. 459-487) and we feel that the reader is openly informed about the assumptions underlying our multi-proxy approach.

We present what we feel is the most complete decadal-resolved micropalaeontological dataset (abundance and morphometric analyses (length, width, area, mass) of individual coccoliths to the best of our knowledge. Therefore, while our geochemical and micropaleontological proxies could have been compared with further complementary independent indicators of export productivity we are convinced that we present a very comprehensive and robust record of paleo-STP and CCP changes.

C4: The authors state that “downcore trends in TOC contents recorded at site MD07-3088 mirror those from sites 1233 and 1234, located in the central Chilean margin (Muratli et al., 2009), indicating that sedimentary TOC concentrations can be used as a robust first-order proxy reflecting POC export, at least qualitatively” – First, I don't think they really mean “mirror” (opposite

patterns), secondly, I don't see why similar patterns in three locations means that all (or any) of them consequently must represent export productivity changes.

R4: Fair point. "Mirror" should be replaced by "Mimic".

Similar patterns at three locations show that our TOC record most probably document a regional feature, and is not significantly biased by remineralization processes, that are expected to have different patterns depending on the location of the studied cores. Because the latitudinal distribution pattern of TOC in surface sediments along the Chilean margin closely resembles satellite-derived chlorophyll concentrations in the area, then the sedimentary TOC concentrations can be used as a POC export proxy. See our entire previous reply R2 as pointed below: *"Indeed complex remineralization processes occur within both the water column and the sediments. However, as already mentioned in our previous response, it appears that i) the latitudinal distribution pattern of TOC in surface sediments along the Chilean margin (Hebbeln et al., 2000a) closely resembles satellite-derived chlorophyll concentrations (Thomas et al., 1994), and ii) downcore trends in TOC contents recorded at site MD07-3088 mirror those from sites 1233 and 1234, located in the central Chilean margin (Muratli et al., 2009), indicating that sedimentary TOC concentrations can be used as a robust first-order proxy reflecting POC export, at least qualitatively."*

We apologize for not mentioning that sedimentary TOC concentrations can be used as a robust first-order proxy reflecting regional export, at least qualitatively.

C5: R8: You point out potential differences between your age model and that of Martinez Boti used for the EEP [CO₂aq] curve as the likely explanation for the difference in timing between EEP [CO₂aq] changes and calcification changes in your core, in the SE Pacific. Another possibility is that the age model match is OK, and upwelling/CO₂aq changes were not exactly synchronous between the two sites. Maybe it is worth mentioning this. Or mention that you preferred interpretation is that the mismatch between peaks results from age model offsets of up to xx kyr during the deglaciation, as written in your response.

R5: We infer potential discrepancies between the SAZ (and not the EEP) record presented by Martinez-Boti. These differences are inherent to the different (independent) approaches used to establish the respective stratigraphic frameworks. Our age model is based on reservoir age corrections ranging from 1300 to 800 years since the LGM (see Table S1 of Siani et al., 2013), while the age model established by Martinez-Boti et al., (2015) is based on constant reservoir ages (see our previous reply R8).

Furthermore, Ekman-driven upwelling in the Southern Ocean, draws CO₂ enriched deep water up to the surface. In modern settings, this upwelling branch synchronously influences surface waters of the entire Antarctic and Subantarctic Zones without any apparent differences between Pacific, Indian or Atlantic sectors (Marshall and Speer, 2012). We hypothesis that the situation remained unchanged for the past.

C6: R12: are there no opal MAR records, or even smear slide descriptions, from your core that may shed light on this hypothesis? It seems a shame to not discuss this Holocene "mismatch" compared to your interpretation of your records in the rest of the core, to ignore this interval rather degrades your overall interpretations of proxy records.

R6: As mentioned in our previous reply R12, we unfortunately do not have any biogenic opal or diatom data at our site. We feel that without such data, the discussion related to the Holocene would be quite speculative, and beyond the scope of the present study.

C7: C8: I understand that, because of a lack of air-sea equilibrium wrt CO₂ in your study region, you cannot use your core's SST record to reconstruct local [CO₂aq] using ice-core pCO₂. However, because your local [CO₂aq] will be very sensitive to local temperature changes and local increases in upwelling (cooling or increased upwelling -> higher [CO₂aq]), is the record of Martinez-Boti really likely to be representative of what coccolithophores are experiencing at your site? This warrants a brief sentence or two of discussion.

R7: As explained several times in our previous replies (e.g. R30, R31, R36, R42, 547, R53 of our initial point-by-point response), and in our reply R5 above, there is no local upwelling at site MD07-3088. Coccolithophore patterns observed at our site are mainly driven by changes in the Southern Ocean upwelling that impact the surface water chemistry of the entire SAZ, including the Atlantic sector (i.e. site PS2498-1).

C8: Line 233 (and other places): associated with "The increase in the CCP is not only associated with rising macronutrient availability, but is associated to rising macronutrient availability. This relationship is well documented lines 132-193 for coccolithophores, since increased rates in coccolith abundance and mass are recorded due to increased fertility and [CO₂aq] conditions (associated to the major upwelling phases)."

This is simply a matter of correct English usage. "associated to" is not grammatically correct. Maybe you prefer to say "is associated with and likely driven by" increased macronutrient availability?

R8: amended (line 230)

C9: Line 24: coccolith abundance
R9: Amended

C10: Line 96: “accounts for a significant proportion of the global marine export production” reference needed.
R10: “Winter and Siesser (1994)” (now ref. 25) has been added to the text.

C11: Line 116 and paper title: I am not entirely sure what you mean by “enhancing ocean-atmosphere carbon partitioning”, you need to be more explicit. I am not sure if you are trying to say “reducing exchange between surficial and deep ocean carbon reservoirs (e.g. by increased stratification between surface and deep ocean)”, “enhancing ocean carbon uptake from the atmosphere” or “enhancing C uptake by the surface ocean and export towards the deep ocean”... or something else. I have the same problem with the paper’s title, and I strongly suggest you change it to something clearer.

R11: By enhanced partitioning we mean that deglacial biological export production contributed to the release of respired carbon from the deep ocean reservoir to the surface, and thus to the atmosphere.
We’ve made it clearer within the manuscript: See lines 114:117: “Our study highlights that changes in biological export production in high southern latitudes operated synergistically with physical mechanisms thereby enhancing the transfer of carbon from the ocean to the atmosphere during the last deglacial”.

C12: Line 127: Add Site 1233 to the map.
R12: Done – changes have also been made within the figure caption (line 702).

C13: Line 144: have an affinity
R13: Done (l. 143)

C14: Line 177: more heavily calcified coccoliths
R14: Done (l. 176)

C15: Line 188: ~ 2-3 $\mu\text{m/L}$ rises in SSW [CO_2aq] in the EEP
R15: In the SAZ has been added to the text (l. 187) (see our reply R5).

C16: A few sentences explaining why summer SST increases occur during periods of enhanced upwelling would be appropriate – usually increased upwelling is associated with cooling, which would effectively enhance your [CO_2aq] increase, rather than counteract it because CO_2 is more soluble in colder water. Does upwelling likely occur mainly in winter?

R16: We’ve already answered this specific point in our initial reply: see the previous comment/reply 53 below:
C53: Please explain how increased upwelling intensity (of SAMW?) during HS1 and the YD leads to increased SST, when usually enhanced upwelling is associated with decreased SST.
R53: We made this argument more explicit in the revised text, and now refer to increased SO upwelling at the Antarctic Divergence (as opposed to local upwelling). These increased SO upwelling episodes occur during Southern Hemisphere deglacial warm phases and are possibly linked to a southward shift of the prevailing westerly winds (Toggweiler et al. 2006) and/or sea-ice retreat (Keeling and Stephens 2000, Ferrari et al. 2014).

Therefore, the mechanisms and relationships between upwelling and SST highlighted above by Reviewer #3 may be applied to local/coastal upwelling systems. This is not the case here (See also R7 above).

C17: Line 214: calcifiers
R17: Done (l. 212)

C18: Line 217: rephrase. E.g., We suggest that, in our core, POC:PIC ratio changes in the sediments likely reflect changes in the C rain ratio (POC:PIC) (See methods). Therefore, downcore Br/Ca....
R18: Done (l. 214-216)

C19: Line 433: decrease by
R19: We are somewhat confused here, as the suggested modification does not match the sentence l. 433.

References

- Gottschalk, J., et al., Nature Communications, doi:10.1038/ncomms11539 (2016).
- Kohfeld, K.E. et al., Science 308, 74-78 (2005).
- Kumar, N., et al., Nature 378, 675 – 680 (1995).
- Lamy, F., et al., Science 343, 403-407, doi: 10.1126/science.1245424 (2014).
- Marshall, J., Speer, K. Nat. Geoscience 5, 171–180 (2012).
- Martinez-Garcia, A. et al., Paleoceanography 24, PA1207, doi : 10.1029/2008PA001657 (2009).
- Martinez Garcia, A., et al., Science 343, 1347-1350 (2014).
- Muratli, J.M., et al., Nature Geoscience 3, 23-26, doi: 10.1038/NGEO715 (2010).